# G²RPO: Geometric GRPO
# Escaping LLMs' Reasoning Ruts to Break the Accuracy–Entropy Trade-off

**Ali Rad** [1]   **Khashayar Filom** [1]   **Darioush Keivan** [1]   **Peyman Mohajerin Esfahani** [2]   **Ehsan Kamalinejad** [1]

## Abstract

Reinforcement learning with verifiable rewards (RLVR) is a cornerstone of post-training for large reasoning models, yet widely used algorithms such as Group Relative Policy Optimization (GRPO) often exhibit diversity collapse. We provide a geometric diagnosis by formalizing GRPO as a dynamical flow on the probability simplex. Under a mode-based coarse-graining of rollouts, GRPO induces a collision field over correct modes, monotonically pushing probability mass toward simplex vertices and producing a winner-take-all regime. To address this, we introduce G²RPO (Geometric GRPO), which performs vector-field editing by adding advantage-level granularity bonuses inversely proportional to mode probabilities, encouraging underrepresented correct modes while mitigating performance side effects. In experiments with 7B and 14B models trained on math reasoning and evaluated on AIME 2024/2025, G²RPO prevents the late-stage entropy crash, increases active correct-mode coverage over the base models by 172%–205%, and improves pass@1 by +1.4 to +7.9 points relative to GRPO. Overall, diversity is not merely a regularizer but a geometric property that must be controlled to improve reasoning models without trapping them in a single dominant strategy.

**Code:** https://github.com/cognichip/G2RPO.

## 1. Introduction

Reinforcement learning with verifiable rewards (RLVR) is a standard post-training stage for LLMs on verifiable tasks

[1]Cognichip AI [2]University of Toronto. Correspondence to: Ali Rad <ali@cognichip.ai>.

*Proceedings of the 43rd International Conference on Machine Learning*, Seoul, South Korea. PMLR 306, 2026. Copyright 2026 by the author(s).

(e.g., math, coding), typically via PPO/GRPO-style policy gradients (Schulman et al., 2017; Shao et al., 2024). While effective, RLVR often rapidly reduces "uncertainty": in GRPO this appears as token-entropy collapse (Yu et al., 2025) or an accuracy–entropy trade-off (Cui et al., 2025). Yet token entropy and `pass@k` can miss family-level concentration; the relevant observable is (coarse-grained) sequence-level entropy over solution families. More generally, RLVR often increases `pass@1` while leaving `pass@k` nearly unchanged (Cobbe et al., 2021; Yue et al., 2025), entering a winner-take-all regime where one correct family suppresses others. We call this **diversity collapse**. Crucially, it can *hide in plain sight*: token entropy and `pass@k` may remain high while the solution-family distribution concentrates on a single correct family. We call this a *reasoning rut*: the model may become stronger on familiar patterns but narrower in the range of solution families it uses.

This paper asks: *Is diversity collapse in GRPO a predictable consequence of the underlying learning dynamics, and can a principled geometric edit mitigate it in practice?*

Here and throughout, the theoretical claim is made within a mean-field mode-policy abstraction with finite good–bad coarse-graining, a binary verifier, and KL-free GRPO dynamics. Finite rollouts, discrete optimizer steps, and changing mode sets introduce approximation error, which we probe empirically in Sections A and 5.

**Contributions.**   Here are our main contributions:

- **A mean-field model for GRPO.** We adopt a prompt-level RLVR lens that treats the policy as a *mode policy* over recurring reasoning modes, and in the mean-field regime derive the GRPO ODE system (2) governing (i) normalized probabilities over good/bad modes and (ii) the total bad mass (the complement of accuracy), formalizing the heuristic that *correct modes compete like species while incorrect modes diffuse like noise.*

- **A diversity observable beyond token entropy.** We highlight that token-level entropy and `pass@k` can fail to detect diversity collapse at the level of solution families. We therefore focus on sequence-level (mode) entropy of the completion distribution, or its family-level coarse-

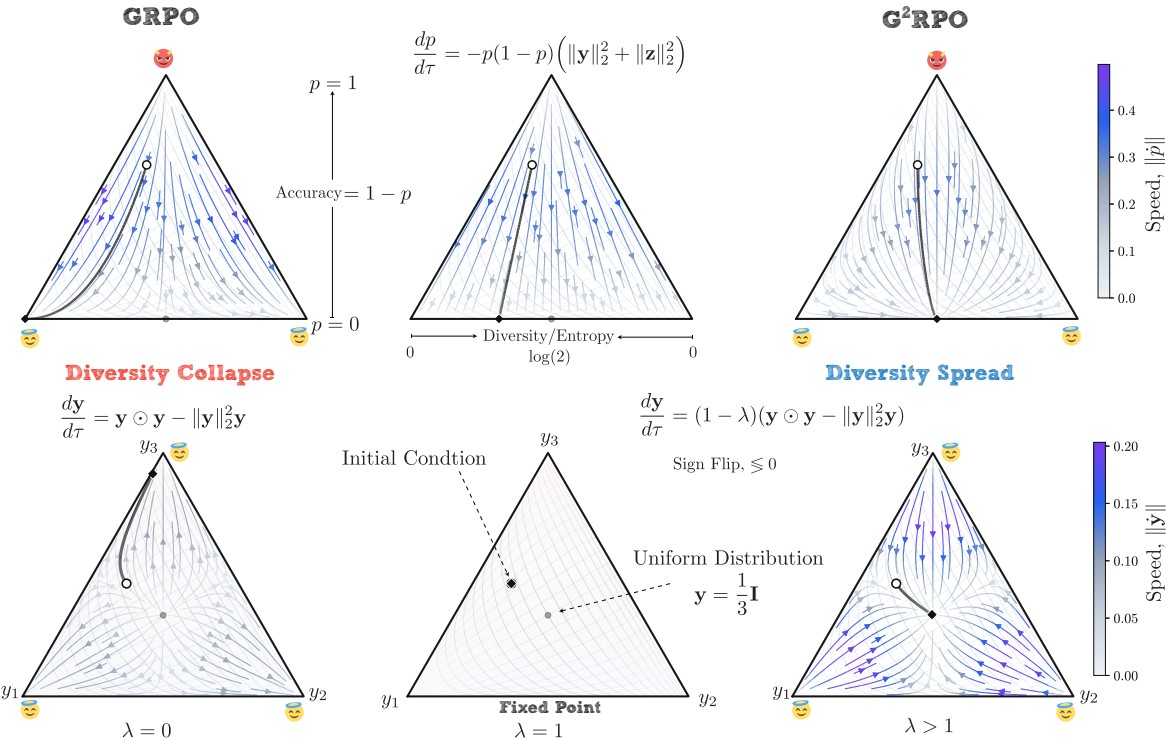

*Figure 1.* **GRPO is a collision field; G²RPO edits it into anti-collision.** Mean-field phase portraits on the probability simplex under the $(p, y, z)$ decomposition (see Eq. 2, and the natural time-rescaling $\tau$ following it). GRPO concentrates probability among correct modes (collision). Our base G²RPO uses an inverse-probability granularity bonus to cancel (fixed point) or flip (anti-collision) the within-good drift.

graining, as the relevant observable, and we operationalize it empirically via clustering-based estimates of mode distributions.

- **A theoretical justification of GRPO's brittleness.** We show that this competition among good modes yields a meaningful *winner-take-all* outcome: generically, the within-good distribution converges to a simplex vertex because GRPO induces a *collision field* (cf. (3)) on the simplex; see Lemma 4.1.

- **A principled modification to GRPO.** To resolve diversity collapse, we introduce **G²RPO** (Algorithm 1), obtained by adding a per-sample advantage bonus proportional to inverse mode probability (a choice unique in a precise sense). This reshapes the within-good dynamics, causing the entropy on the good simplex to increase; see Section 4.

- **Neutrality.** We show this can be implemented at the advantage level without altering the ODE governing the total bad mass, and thus without qualitatively changing learning-speed/accuracy dynamics.

- **Experimental validation.** Despite the mean-field approximation and the per-prompt multi-armed bandit abstraction, we find that the resulting algorithm improves

diversity in practice: using a Sentence-Transformers embedding model to cluster rollouts into modes, experiments with 7B- and 14B-parameter models show that G²RPO improves accuracy while, unlike vanilla GRPO, sustaining mode diversity; see Section 5.

**Conflict of Interest Disclosure.** The authors declare no financial conflicts of interest related to this work.

## 2. Related Work

**RLVR and Group-Relative Policy Optimization.** Reinforcement learning with verifiable rewards (RLVR) is widely used for reasoning tasks where correctness can be checked automatically (Wen et al., 2025; Su et al., 2025; Cai et al., 2025). Group-normalized variants of policy gradient (e.g., GRPO/RLOO-style updates) are attractive because they remove the need for a learned value function while stabilizing variance through within-group normalization (Ahmadian et al., 2024; Yu et al., 2025). Our work studies the dynamics induced by these group-relative updates in the binary-verifier regime.

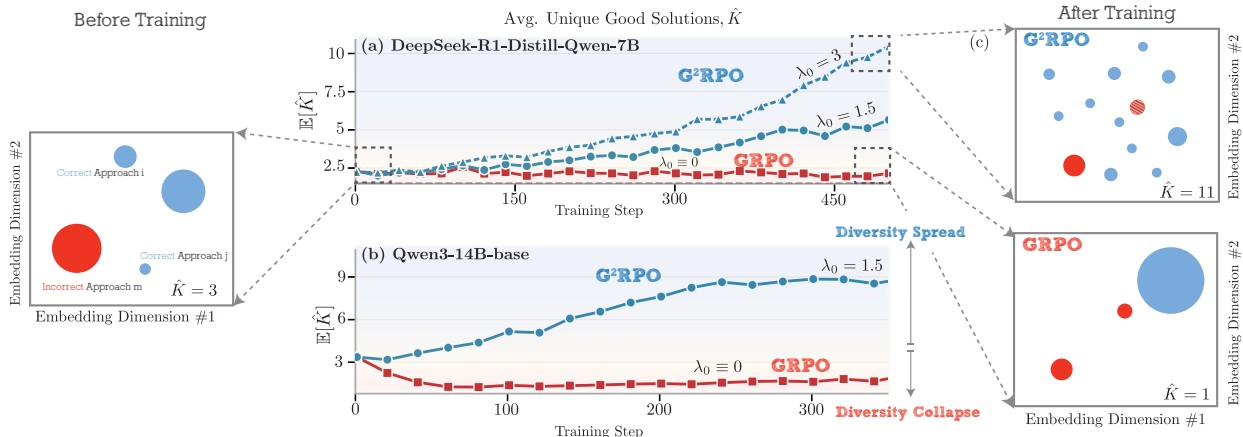

**Figure 2. Mode coverage increases under G²RPO.** Avg number of unique good solutions $\hat{K}$ over training (center) and an embedding-space visualization of clusters before/after training (left/right). GRPO collapses to a single dominant correct cluster; G²RPO spreads mass across many correct clusters.

**Entropy Collapse and Diversity in RL.** Policy optimization often exhibits an "entropy crash" where the policy concentrates sharply late in training. In RLVR for LLMs, this manifests as homogenized reasoning traces and weaker gains in `pass@k` compared to `pass@1` (Cui et al., 2025; Yue et al., 2025). Common mitigations include KL regularization to a reference model (as in RLHF) and explicit entropy bonuses, but these typically trade off stability, compute, and final accuracy (Ouyang et al., 2022; Lightman et al., 2023). G²RPO instead targets the *within-good* concentration mechanism directly.

**Mode-Level Abstraction.** Rollout-level reward allows us to model the policy as a multi-arm bandit for each prompt (Kreutzer et al., 2017; Nguyen et al., 2017). For us, arms correspond to *solution families/modes*.

**Simplex Geometry and Replicator Dynamics.** The mean-field ODE (cf. (Rad et al., 2026)) is closely related to classical replicator dynamics on the probability simplex and its Shahshahani (information-geometric) interpretation (Shahshahani, 1979; Hofbauer & Sigmund, 1998). Our contribution is to connect this geometry to GRPO's Jacobian-squared channel in RLVR, and to provide a minimal "vector-field edit" that flips the winner-take-all drift while preserving the baseline accuracy-learning trajectory.

## 3. GRPO as a Simplex Flow Over Reasoning Modes

For a fixed prompt $x$, an LLM with parameters $w$ induces a distribution $\pi_w(\cdot \mid x)$ over completions. Despite the astronomical size of the completion space, rollouts in practice cluster into a small number of modes. In Section 5, we estimate these modes using semantic embeddings and clustering

(cf. (Zhou et al., 2025)).

We capture this structure with a prompt-wise bandit abstraction by *coarsening* completions into finitely many *evaluation modes* $\{h_1, \ldots, h_{K+M}\}$ using a task-specific equivalence rule (e.g., verifier- or rubric-equivalence). The induced mode-level policy is a categorical distribution $\mathbf{p}(t) \in \Delta^{K+M-1}$, which we parameterize by

$$\mathbf{p}(t) = \big(p_1(t), \ldots, p_K(t), p_{b_1}(t), \ldots, p_{b_M}(t)\big) \in \Delta^{K+M-1},$$
$$\mathbf{p}(t) = \text{softmax}\big(\theta(t)\big), \qquad \theta(t) \in \mathbb{R}^{K+M}.$$

Crucially, $\theta$ is *not* the model parameter vector $w$: it is a low-dimensional, prompt-dependent summary of mode masses (defined up to additive shifts) that evolves implicitly as GRPO updates $w$. See Appendix B for more details. In the RLVR setting, the verifier assigns each mode a correctness label, inducing a good–bad split into $K$ **good** modes (indexed $1:K$) and $M$ **bad** modes (indexed $b_1:b_M$). This abstraction is the minimal setup needed to make diversity collapse a statement about how probability mass moves across modes.

**A $(p, y, z)$ Decomposition.** Define the total bad mass $p \in [0, 1]$ and the within-block compositions

$$p := \sum_{m=1}^{M} p_{b_m}, \qquad y_j := \frac{p_j}{1-p} \quad (j = 1, \ldots, K), \tag{1}$$
$$z_m := \frac{p_{b_m}}{p} \quad (m = 1, \ldots, M).$$

Then $y \in \Delta^{K-1}$ and $z \in \Delta^{M-1}$ (where $\Delta^{d-1} := \{u \in \mathbb{R}^d : u_i \geq 0, \sum_{i=1}^{d} u_i = 1\}$ is the probability simplex), and the policy factors as $\mathbf{p} = ((1-p)\,y, \; p\,z)$. The scalar $p$ tracks accuracy progress, while $y$ and $z$ capture mode diversity within the good/bad blocks, respectively.

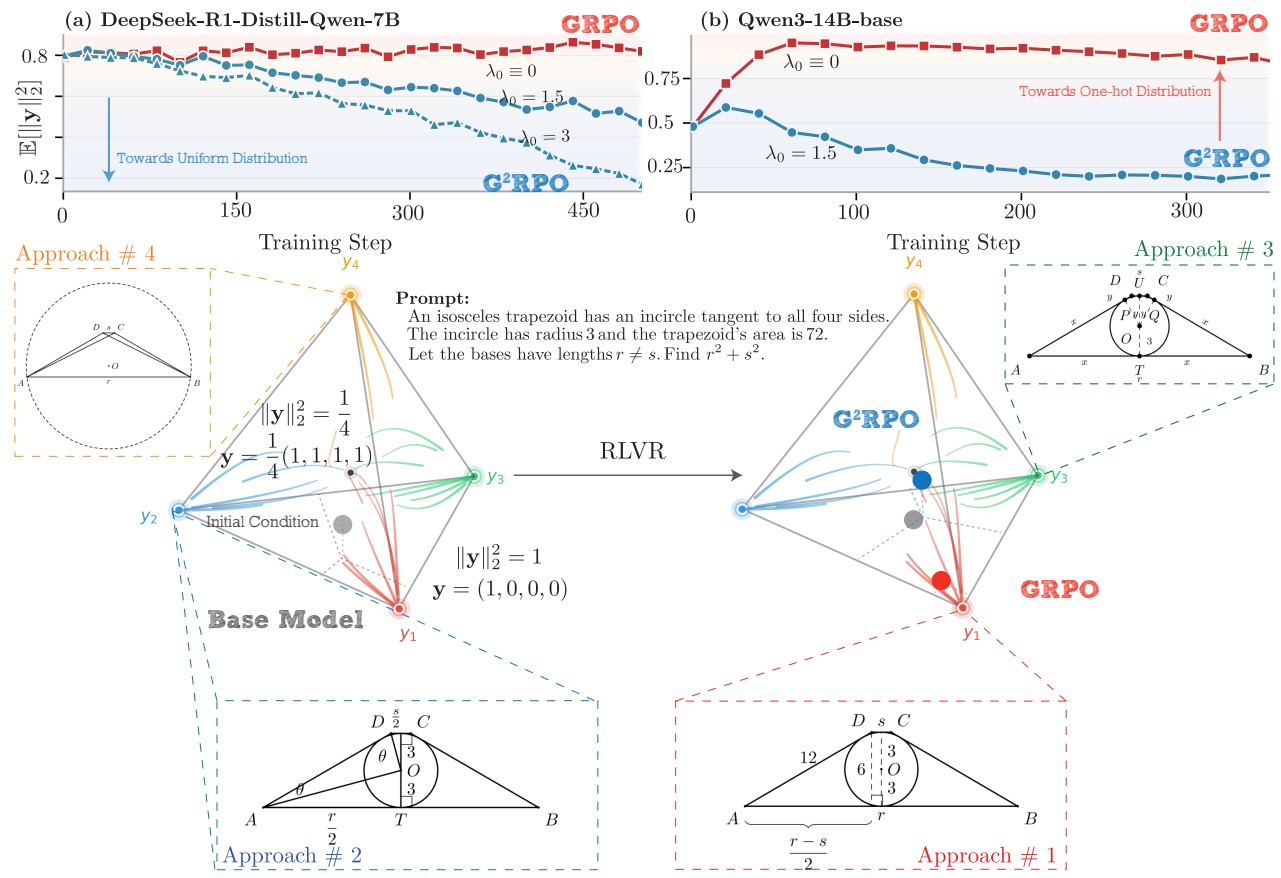

*Figure 3.* **Within-good concentration $L_y$ during training.** GRPO drives $L_y \uparrow 1$ (winner-take-all), while G²RPO lowers $L_y$ toward a more uniform mixture over good modes. Bottom: simplex trajectories before/after training for a representative prompt (vertices are distinct good solution modes, see Appendix. G); GRPO collapses to one mode (reasoning rut), whereas G²RPO remains multi-modal, and the entropy of the probability distribution on the good simplex increases.

**Mean-Field GRPO Dynamics.** Under mean-field assumptions for group-normalized RLVR (Rad et al., 2026), GRPO induces an ODE in $(p, y, z)$. In the noiseless binary-verifier setting (used in our experiments), the interior dynamics ($p \in (0, 1)$) take the form

$$\begin{aligned} \dot{y} &= \kappa(p)\, V(y), \qquad \dot{z} = -\kappa(p)\, V(z), \\ \dot{p} &= -\kappa(p)\, p(1-p)\left(\|y\|_2^2 + \|z\|_2^2\right), \end{aligned} \quad (2)$$

where $\kappa(p) = \eta\sqrt{p(1-p)}$ and

$$V(u) := u \odot u - \|u\|_2^2\, u. \quad (3)$$

where $\odot$ denotes the Hadamard product. Introducing the internal time $\tau$ via $d\tau/dt = \kappa(p(t))$, the within-block flows become $\frac{dy}{d\tau} = V(y)$ and $\frac{dz}{d\tau} = -V(z)$. A self-contained derivation of these ODEs is provided in Appendix B.

**Accuracy–Diversity Coupling.** The bad-mass channel in Equation (2) depends on the *concentrations* $\|y\|_2^2$ and $\|z\|_2^2$. Intuitively, when the policy concentrates on a small set of modes, the group-normalized update becomes higher signal-to-noise and drains bad mass faster. This coupling explains why naive diversity bonuses can create an apparent accuracy–entropy trade-off: reducing concentration (good for diversity) can slow $\dot{p}$ (bad for accuracy) unless corrected. G²RPO's neutrality mechanism in Section 4 is designed to keep $\dot{p}$ close to the baseline $\dot{p}$ while editing only the within-good flow.

## 4. G²RPO: Vector-Field Editing via Granularity Bonuses

**Collision Geometry and Diversity Collapse.** In internal time, the good-block evolves under the collision flow $\frac{dy}{d\tau} = V(y)$ on the simplex, which progressively concentrates probability mass onto fewer coordinates. A convenient diversity diagnostic is the within-good $\ell_2$ concentration

$$L_y := \|y\|_2^2 \in [1/K, 1], \qquad K_{\text{eff}} \approx \frac{1}{L_y},$$

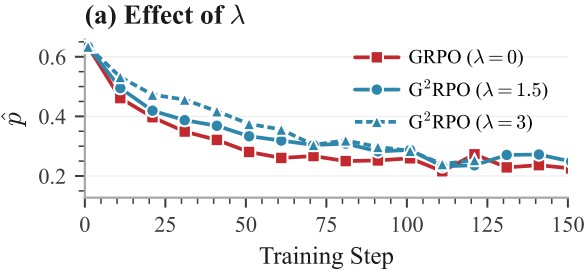

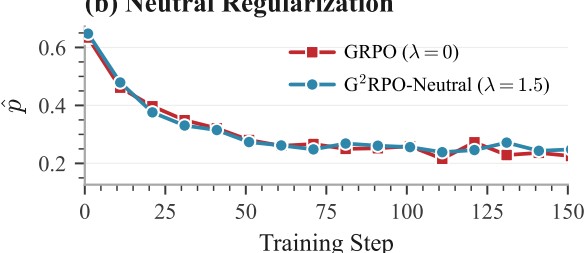

*Figure 4.* **Neutrality stabilizes the bad-mass channel.** Left: increasing $\lambda$ slows mid-run decay of $\hat{p}$. Right: adding neutrality largely restores the GRPO $\hat{p}$ trajectory.

where $K_{\text{eff}}$ is the effective number of active good modes (larger $K_{\text{eff}}$ indicates broader mode coverage). A key consequence is *monotone concentration*: along GRPO trajectories, $L_y$ increases unless $y$ is uniform on its support.

**Lemma 4.1** (Monotone concentration of correct modes). *Assume $\kappa(p) \geq 0$. Along solutions of $\dot{y} = \kappa(p)\, V(y)$,*

$$\frac{d}{dt}\, \|y\|_2^2 = 2\kappa(p)\Big(\sum_{j=1}^{K} y_j^3 - \|y\|_2^4\Big) \; \geq \; 0, \qquad (4)$$

*with equality iff $y$ is uniform on its support.*

Moreover, under a mild genericity condition (no exact ties), the identity of the largest coordinate of $y$ is preserved and the flow converges to a vertex (winner-take-all), i.e., diversity collapse among correct modes. Formal statements and proofs are in Section B.3.

GRPO's diversity collapse is driven by the collision geometry of $V(\cdot)$ acting on the within-block compositions. Rather than adding ad-hoc exploration heuristics, we cast RLVR shaping as a *vector-field editing* problem: we edit the induced mean-field vector field to address diversity collapse within the good block, promoting broader coverage over correct modes, while keeping the bad-mass evolution law (the $p$-channel) close to the baseline functional form.

**Advantage Shaping.** G²RPO keeps the GRPO/PPO objective and optimizer unchanged, and modifies only the per-sample advantage. Let $\widehat{A}^{\text{base}}$ denote the standard group-normalized outcome advantage. We use

$$\widehat{A}^{\text{G}^2} = \widehat{A}^{\text{base}} + B(p, y, z; \lambda(t)),$$

where $B$ is a gain-controlled *granularity bonus* computed from prompt-wise mode statistics. For clarity, we decompose it into a good-mode term and a bad-mode term,

$$B = \big(B^+(y), \; B^-(p, y, z)\big),$$

where $B^+$ is designed to counteract within-good collision (diversity collapse), and $B^-$ enforces a neutrality condition so that the bad-mass channel $p$ remains in the same functional form.

**Good-Mode Anti-Collision Bonus.** Within the good block we use a mode-wise bonus $B^+(y) = (B_j^+(y))_{j=1}^K$ that upweights rarer correct modes:

$$B_j^+(y) = \lambda(t)\Big(\frac{1}{Ky_j} - 1\Big), \qquad j = 1, \ldots, K.$$

It is $y$-centered ($\langle y, B^+(y)\rangle = 0$) and diverges as $y_j \to 0$, discouraging collapse of low-mass good modes.

**Why Inverse-Probability? (Collinearity & Essential Uniqueness).** This reciprocal form is not ad hoc: in mean-field, bonuses enter GRPO through a Jacobian-squared filter, which induces a within-block drift of the form $C_s(u) = s^{\odot 2} \odot u - s\langle s^{\odot 2}, u\rangle$. To edit collapse *without introducing a new tangent direction*, we impose the collinearity constraint $C_s(B) = \alpha(s)\, V(s)$, where $V(s) = s^{\odot 2} - \|s\|_2^2 s$ is the native collision field. Restricting to permutation-equivariant scalar bonuses $B_i(s) = f(s_i)$, this condition yields $f(s) = c/s + d$, so the centered reciprocal family above is (up to scaling and shifts) the unique symmetric choice that remains collinear and provides a single gain that can slow, cancel, or flip the collision drift (Appendix. C.2).

**Effect in Mean-Field: A Clean Sign Flip.** The reciprocal bonus yields the internal-time good-block ODE

$$\frac{dy}{d\tau} = \big(1 - \tilde{\lambda}(t)\big) V(y), \qquad (5)$$

where $\tilde{\lambda}(t)$ is an effective dimensionless gain determined by $(p, \lambda(t))$. Thus $\tilde{\lambda} = 1$ cancels collision and $\tilde{\lambda} > 1$ flips it into anti-collision, making $y = \frac{1}{K}\mathbf{1}$ stable.

**Neutrality: Preserving the Bad-Mass Learning Channel.** Because $p$ couples to within-block concentrations in Equation (2), a good-only bonus can inadvertently perturb $\dot{p}$. We therefore enforce a neutrality principle:

$$\dot{p}\ (\text{with bonus}) \approx \dot{p}\ (\text{baseline GRPO}).$$

In the $(K+M)$-mode model, neutrality is achieved by adding a block-uniform offset to all incorrect modes. One

**Algorithm 1** G$^2$RPO update for one prompt group

---

**Require:** Prompt $x$, policy $\pi_\theta$, verifier $V(\cdot)$, group size $G$, gain $\lambda(t)$
1: Sample rollouts $\{o_i\}_{i=1}^G \sim \pi_{\theta_{\mathrm{old}}}(\cdot \mid x)$ and verify rewards $r_i \leftarrow V(o_i) \in \{0,1\}$.
2: Split indices $\mathcal{G} = \{i : r_i = 1\}$ (good) and $\mathcal{B} = \{i : r_i = 0\}$ (bad); set $\hat{p} \leftarrow |\mathcal{B}|/G$.
3: Cluster $\{o_i\}_{i \in \mathcal{G}}$ to obtain $K$ clusters and masses $\hat{y} \in \Delta^{K-1}$.
4: (Optional) cluster $\{o_i\}_{i \in \mathcal{B}}$ to estimate $\hat{z}$; otherwise use the empirical bad histogram.
5: **for** $i = 1$ to $G$ **do**
6:    **if** $i \in \mathcal{G}$ **then**
7:       Let $j = \mathrm{cluster}(o_i)$ and set $B_i \leftarrow \lambda(t)\left(\frac{1}{K\hat{y}_j} - 1\right)$.
8:    **else**
9:       Set $B_i \leftarrow B^-(\hat{p}, \hat{y}, \hat{z})$ (neutralizer; Equation (6)) or 0.
10:    **end if**
11: **end for**
12: Form per-sample advantages $\widehat{A}_i \leftarrow$ NormalizeGroup($r_i$) $+ B_i$ and broadcast to response tokens.
13: Update $\theta$ with the standard clipped GRPO/PPO loss (unchanged) using $\widehat{A}$.

---

closed form (applied per bad mode) is

$$B^-(p, y, z) = \lambda(t) \frac{\frac{1}{K} - \|y\|_2^2}{p\left(\|y\|_2^2 + \|z\|_2^2\right)}. \tag{6}$$

(Full derivation and stable gain schedules appear in Appendix. Section E.)

**From Theory to Practice: Estimating Modes from Rollouts.** In training, we observe only $G$ rollouts per prompt. Based on Algorithm. 1, we estimate $(\hat{p}, \hat{y}, \hat{z})$ by verifying rollouts and clustering the correct (and optionally incorrect) samples in embedding space. Let $n_j$ be the size of correct cluster $j$ among $G_{\mathrm{good}}$ correct samples; then $\hat{y}_j = n_j/G_{\mathrm{good}}$. G$^2$RPO assigns each correct rollout in cluster $j$ the bonus $\lambda(t)(\frac{1}{K\hat{y}_j} - 1)$ and assigns each incorrect rollout the neutralizer in Equation (6) (with plug-in estimates). Implementation is a one-line modification: add the per-sample bonus to GRPO advantages and broadcast over response tokens (see Appendix. H).

**Practical Note: Finite Rollouts and Safeguards.** With finite groups, we estimate mode masses from cluster counts rather than per-sequence probabilities. If no correct mode is found, we set $B_i = 0$, recovering GRPO; if $\hat{p} = 0$, we apply the good-mode bonus but omit the neutralizer. In our $G = 16$ runs, training was stable and the neutralizer was never clipped: any nonzero $\hat{p}$ is at least $1/16 = 0.0625$,

so the safety threshold was not triggered. Flooring, bonus clipping/ramping, and neutralizer clipping/disable are kept only as safeguards for smaller or noisier groups.

## 5. Experiments

**Training Setup.** We experiment with two models: a reasoning ("thinking") model, `DeepSeek-R1-Distill-Qwen-7B`, and a non-reasoning baseline, `Qwen3-14B-Base`. We train on `DAPO-17K` (Yu et al., 2025) for 8 epochs with global batch size 256 and group size $G$=16 rollouts/prompt. Verification uses exact match on extracted final answers. We disable KL regularization ($\beta$=0) to isolate the effect of the bonus. Unless otherwise stated, we use a constant gain $\lambda_0 = 1.5$ with the schedule and enable neutrality. Hyperparameters are in Appendix. H.

**Mode Discovery.** To estimate recurring reasoning modes, each prompt's rollouts are embedded (all-MiniLM-L6-v2) and clustered with DBSCAN; we treat each cluster as a reasoning mode and use cluster frequencies to estimate $\hat{y}$.

Each correct cluster corresponds to a coarse *good mode*; the number of clusters estimates $K/M$ and cluster frequencies estimate $\hat{y}, \hat{k}$. We report diversity metrics aggregated across prompts.

We treat $K$ and $M$ as dynamic, prompt-level empirical quantities estimated from the current rollout group. Because the absolute number of clusters depends on the embedding scale and DBSCAN radius, we compare methods under a fixed operational mode proxy and interpret $\hat{K}$ together with concentration/entropy and downstream pass@k. Appendix A reports encoder and DBSCAN-scale sensitivity checks.

**Metrics.** We report pass@1 accuracy on AIME 2024/2025. For diversity, we log: (i) Avg $K$ (average number of distinct correct clusters), (ii) sequence-level entropy—defined as the entropy of the observed good mode distribution—and token-level entropy (see Appendix. H), and (iii) the within-good concentration $L_y = \|y\|_2^2$ (lower is more diverse; $K_{\mathrm{eff}} \approx 1/L_y$).

To connect mode-level diversity to sampling success, Appendix A.1 additionally reports pass@k for $k \in \{2, 4, 16, 30\}$ under the same evaluation protocol; G$^2$RPO improves over GRPO for every reported $k$ in both backbones.

**Main Results.** Table 1 summarizes the central pattern across both backbones and both AIME 2024/2025 evaluations. GRPO increases pass@1 but *collapses* mode-level diversity below the base model: Avg $\hat{K}$ drops by **16.2%** (7B) and **56.8%** (14B), while the good-mode concentra-

*Table 1.* GRPO and G$^2$RPO compared to the Base model. For diversity proxies (Avg $K$, entropies), GRPO shifts below Base (red) while G$^2$RPO shifts above Base (green). $\Delta_{\text{B}\to X} = X - \text{Base}$.

| Metric | Base | GRPO | | G$^2$RPO | | G$^2$RPO$-$GRPO |
|---|---|---|---|---|---|---|
| | | Value | $\Delta_{\text{B}\to\text{GRPO}}$ | Value | $\Delta_{\text{B}\to G^2\text{RPO}}$ | |
| **DeepSeek-R1-Distill-Qwen-7B** | | | | | | |
| *AIME accuracy (%)* | | | | | | |
| AIME'25 | 38.7 | 46.7 | 8.0 | **48.9** | 11.2 | +2.2 |
| AIME'24 | 52.1 | 60.4 | 8.3 | **61.8** | 9.7 | +1.4 |
| *Diversity proxies* | | | | | | |
| Avg $K$ | 2.35 | 1.97 | -0.38 (-16.2%) | **6.40** | +4.05 (+172.3%) | ↑ +4.43 (+224.8%) |
| Sequence-level entropy | 0.33 | 0.26 | -0.07 (-21.2%) | **0.71** | +0.38 (+115.2%) | ↑ +0.45 (+173.0%) |
| Token-level entropy | 0.76 | 0.49 | -0.27 (-35.5%) | **1.01** | +0.25 (+32.9%) | ↑ +0.52 (+ 106.1%) |
| $L_{2y}$ (lower = more diverse) | 0.80 | 0.84 | +0.04 (+5.0%) | **0.46** | -0.34 (-42.5%) | -0.38 (- 45.2%) |
| **Qwen3-14B-Base** | | | | | | |
| *AIME accuracy (%)* | | | | | | |
| AIME'25 | 9.2 | 37.5 | 28.3 | **40.3** | 31.1 | +2.8 |
| AIME'24 | 13.8 | 45.9 | 32.1 | **53.8** | 40.0 | +7.9 |
| *Diversity proxies* | | | | | | |
| Avg $K$ | 3.36 | 1.45 | -1.91 (-56.8%) | **10.25** | +6.89 (+205.1%) | ↑ +8.80 (+606.9%) |
| Sequence-level entropy | 0.78 | 0.46 | -0.32 (-41.0%) | **0.95** | +0.17 (+21.8%) | ↑ +0.49 (+106.5%) |
| Token-level entropy | 0.87 | 0.18 | -0.69 (-79.3%) | **1.90** | +1.03 (+118.4%) | ↑ +1.72 (+955.6%) |
| $L_{2y}$ (lower = more diverse) | 0.47 | 0.81 | +0.34 (+72.3%) | **0.15** | -0.32 (-68.1%) | -0.66 (-81.5%) |

tion $L_y = \|y\|_2^2$ increases by **5.0%** and **72.3%**, respectively. G$^2$RPO breaks this tension. It improves `pass@1` over GRPO by **+1.4 to +7.9** points while *simultaneously* expanding active correct-mode coverage by **+225%** (7B) and **+607%** (14B) relative to GRPO, with consistently lower $L_y$ and higher mode coverage. Token- and sequence-level entropies follow the same qualitative trend as secondary sanity checks (not primary proxies); see Figures 2, 3, 5 and 6.

**Training Dynamics: GRPO Collapses; G$^2$RPO Spreads.**
The mean-field theory predicts a winner-take-all drift among correct modes under GRPO. Empirically, GRPO steadily drives $\hat{K}(t)$ downward and $L_y(t)$ upward, indicating progressive concentration onto a single dominant correct mode. G$^2$RPO produces the opposite signature: $\hat{K}(t)$ increases and $L_y(t)$ decreases throughout training, moving toward the uniform-spread benchmark $L_y^\star = 1/G$ (here $G = 16$, so $L_y^\star \simeq 0.06$). By the end of training, $L_y$ reaches **0.46** (7B) and **0.15** (14B), reflecting substantial deconcentration despite finite-rollout and clustering noise. Consistent with the mode-level diagnostics, GRPO often exhibits a late-stage entropy crash, whereas G$^2$RPO maintains higher entropy and avoids collapse; see Figures 2, 3 and 6.

**Ablations: Gain Strength and Neutrality.** Increasing the gain $\lambda$ monotonically strengthens anti-collision in the good block: larger $\lambda$ yields higher $\hat{K}$ and lower $L_y$ (Figures 2 and 3). In our setting, $\lambda = 3$ reaches up to **+224%** more discovered good modes than GRPO (and **+92%** over

$\lambda = 1.5$), matching the prediction that $\lambda$ acts as a single, interpretable control on within-good spreading. However, without correction, good-only shaping can perturb the bad-mass channel and slow the early decrease of the incorrect fraction $\hat{p}$, with larger deviations for larger $\lambda$. The neutrality correction restores separation of roles: it keeps $\hat{p}$ close to the GRPO trajectory while retaining the diversity gains in $(\hat{K}, L_y)$; see Figure 4.

**Additional Checks.** Appendix A stress-tests whether the effect is an artifact of the mode proxy or finite-budget setup. On AIME-2025, G$^2$RPO improves over GRPO for every reported `pass@k`: the gains range from **+1.7** to **+4.3** points for Qwen3-14B and from **+2.0** to **+4.9** points for DeepSeek-7B. Encoder and DBSCAN-scale swaps leave the final DeepSeek `pass@30` unchanged at **78%** in the short robustness check, even though the absolute values of $\hat{K}$ and entropy shift as expected. Neutrality mainly improves early training stability, moderate KL dampens but does not remove the diversity advantage, and the math-RLVR checkpoints outperform GRPO on **5/6** zero-shot transfer pairs. Together, these checks support the main conclusion while keeping the strongest empirical claims tied to the controlled AIME setting.

## 6. Conclusion and Limitations

Diversity collapse in GRPO is not merely a tuning artifact: in our mean-field abstraction, GRPO induces a collision

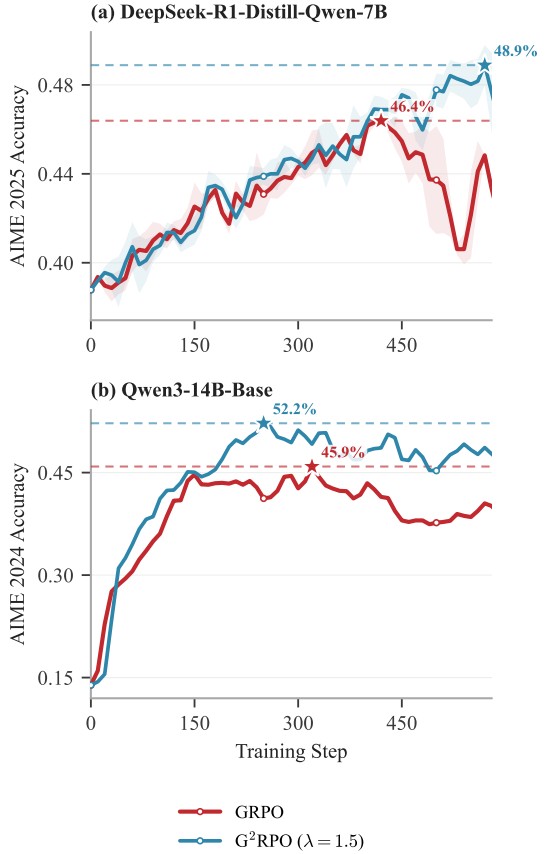

**(a) DeepSeek-R1-Distill-Qwen-7B**

**(b) Qwen3-14B-Base**

GRPO
G$^2$RPO ($\lambda = 1.5$)

*Figure 5.* **Accuracy improves without an "exploration tax."** AIME pass@1 during training. G$^2$RPO (blue) matches or exceeds GRPO (red) while maintaining higher diversity (see Figures 2 and 6).

field that generically drives winner-take-all among correct reasoning modes. G$^2$RPO addresses this by editing the underlying vector field with an inverse-probability granularity bonus, together with a neutrality correction that preserves the accuracy-learning channel. Empirically, this yields broader mode coverage *and* higher accuracy on AIME.

We close by noting limitations and open directions.

**Finite-Rollout Observability.** G$^2$RPO can only reweight modes that appear in a finite group of $G$ rollouts. Very long-tail correct modes may require larger $G$, better sampling, or longer training.

**Clustering as a Proxy.** Mode identification relies on embedding-based clustering of sampled outputs rather than an intrinsic notion of reasoning mode. While our results are robust to reasonable clustering settings, improved discovery (e.g., verifier-aware clustering or trajectory-level features) is a promising direction.

**Mean-Field Approximation.** Our analysis studies a mean-field flow; finite step sizes, clipping, and nonstationarity introduce deviations. Nonetheless, the qualitative signatures (GRPO concentration vs. G$^2$RPO spreading) align with the theoretical picture.

**Evaluation Scope and Mode Proxy.** Our main experiments focus on math RLVR with two models and AIME-style evaluation. The embedding+DBSCAN mode proxy is operational rather than intrinsic: our sensitivity checks suggest the qualitative trend is stable, but absolute mode counts are scale-dependent.

## Impact Statement

This work studies the training dynamics of reasoning models under reinforcement learning with verifiable rewards (RLVR). By promoting diversity among correct reasoning strategies, we aim to reduce the risk that capable models become narrowly optimized around a single solution template. Potential benefits include more robust reasoning in education, programming assistance, and scientific workflows, where alternative solution paths can help users detect brittle or flawed patterns. At the same time, improving reasoning capability can increase the effectiveness of systems used in high-stakes or dual-use contexts, so deployment should include domain-appropriate verification and human oversight. Our method does not remove risks from incorrect rewards, biased benchmarks, or misuse; it targets one class of RLVR mode-collapse failures.

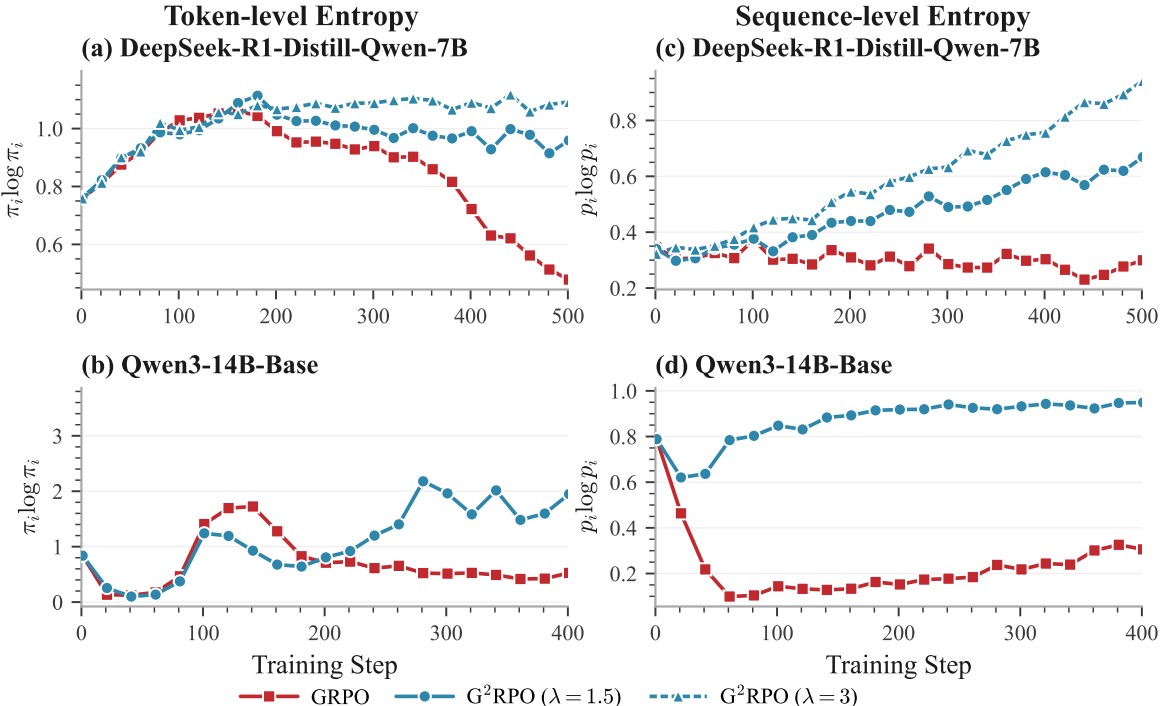

*Figure 6.* **G$^2$RPO reverses entropy collapse.** Token- and sequence-level entropies during training. GRPO exhibits an entropy crash; G$^2$RPO sustains (and for larger $\lambda$ increases) entropy while still improving accuracy.

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

# A. Additional Experiments and Evaluations

The main text focuses on the central AIME `pass@1` results and the mode-level diagnostics that directly test the predicted GRPO collision dynamics. This appendix collects the additional checks used to stress-test that conclusion. We organize them around five questions: whether diversity gains translate into improved sampling success (Section A.1), whether the result depends on a particular clustering proxy (Section A.2), how the method behaves under neutralization and KL changes (Sections A.3 and A.4), whether simpler diversity-oriented alternatives or rollout-group size explain the effect (Section A.5), and how stable and transferable the resulting checkpoints are (Sections A.6 and A.7). These experiments are intended as supplementary evidence rather than exhaustive hyperparameter sweeps; their role is to check whether the qualitative picture in the main paper survives common alternative explanations.

## A.1. Sampling Success Beyond `pass@1`

The main experiments report `pass@1` because it is the most direct single-sample accuracy metric. A natural follow-up is whether broader mode coverage also improves multi-sample success. Table 2 reports AIME-2025 `pass@k` under the same evaluation protocol. G$^2$RPO outperforms GRPO at every reported $k$ for both backbones: the Qwen3-14B gains are $+4.3$, $+2.4$, $+2.8$, and $+1.7$ points for $k \in \{2, 4, 16, 30\}$, while the DeepSeek-7B gains are $+3.7$, $+2.6$, $+2.0$, and $+4.9$ points. Thus the additional mode coverage does not come at the expense of sampling performance. We nevertheless treat `pass@k` as complementary to the mode diagnostics: a high `pass@k` can still be achieved by repeatedly sampling superficial variants of a small number of correct solution families.

*Table 2.* AIME-2025 `pass@k` evaluation. Values are means with uncertainty over evaluation runs.

| Qwen3-14B-Base | pass@2 | pass@4 | pass@16 | pass@30 |
|---|---|---|---|---|
| G$^2$RPO | $64.1\% \pm 14.4\%$ | $70.7\% \pm 11.4\%$ | $77.5\% \pm 9.5\%$ | $81.3\% \pm 5.6\%$ |
| GRPO | $59.8\% \pm 15.8\%$ | $68.3\% \pm 16.6\%$ | $74.7\% \pm 10.4\%$ | $79.6\% \pm 2.2\%$ |
| Base | $22.4\% \pm 15.1\%$ | $30.1\% \pm 22.0\%$ | $49.0\% \pm 7.9\%$ | $56.4\% \pm 12.1\%$ |
| **DeepSeek-R1-Distill-Qwen-7B** | pass@2 | pass@4 | pass@16 | pass@30 |
| G$^2$RPO | $55.4\% \pm 15.3\%$ | $63.0\% \pm 13.5\%$ | $68.8\% \pm 6.3\%$ | $75.1\% \pm 9.2\%$ |
| GRPO | $51.7\% \pm 16.1\%$ | $60.4\% \pm 11.9\%$ | $66.8\% \pm 7.2\%$ | $70.2\% \pm 7.2\%$ |
| Base | $47.3\% \pm 17.2\%$ | $54.2\% \pm 12.1\%$ | $66.1\% \pm 6.2\%$ | $67.7\% \pm 5.0\%$ |

## A.2. Robustness of the Mode Proxy

Our implementation identifies reasoning modes by embedding rollouts and clustering them with DBSCAN. This is an operational proxy: absolute values of $\hat{K}$ and normalized entropy can change with the encoder or clustering radius. The important question is therefore not whether the exact cluster count is invariant, but whether the observed trend—G$^2$RPO increasing correct-mode coverage while maintaining sampling performance—is stable under reasonable changes to the proxy.

Table 3 swaps the clustering encoder inside G$^2$RPO from `all-MiniLM-L6-v2` (Sentence Transformers, 2021; Reimers & Gurevych, 2019) to `bge-small-en-v1.5` (Beijing Academy of Artificial Intelligence, 2023; Xiao et al., 2024), keeping the remaining training configuration fixed for the short run. Both encoders show increasing mode coverage and reach the same final `pass@30` of 78%. Table 4 repeats the check by halving the DBSCAN radius. As expected, the smaller radius reports larger absolute cluster counts, but the qualitative trajectory and final `pass@30` again match the default setting. These results support using $\hat{K}$ and entropy as relative diagnostics under a fixed proxy, while avoiding claims that their absolute values are intrinsic properties of the model.

*Table 3.* Encoder-swap robustness on DeepSeek-R1-Distill-Qwen-7B / DAPO-17K. Each cell reports $(\hat{K}, \hat{H}, \texttt{pass@30})$.

| Encoder | Step 0 | Step 60 | Step 130 |
|---|---|---|---|
| `all-MiniLM-L6-v2` | $(2.2, 0.36, 66\%)$ | $(2.5, 0.35, 76\%)$ | $(2.7, 0.37, 78\%)$ |
| `bge-small-en-v1.5` | $(2.3, 0.39, 66\%)$ | $(2.7, 0.40, 75\%)$ | $(3.2, 0.44, 78\%)$ |

*Table 4.* DBSCAN radius sensitivity using `all-MiniLM-L6-v2`. Each cell reports ($\hat{K}$, $\hat{H}$, `pass@30`).

| Setting | Step 0 | Step 60 | Step 130 |
|---|---|---|---|
| Default $\epsilon$ | $(2.2, 0.36, 66\%)$ | $(2.5, 0.35, 76\%)$ | $(2.7, 0.37, 78\%)$ |
| $\epsilon/2$ | $(2.9, 0.41, 67\%)$ | $(3.4, 0.43, 74\%)$ | $(3.9, 0.47, 78\%)$ |

### A.3. Neutrality Ablation

The neutralizer is designed to preserve the baseline bad-mass learning channel while the granularity bonus reshapes the within-good distribution. Table 5 shows that removing neutrality does not eliminate the eventual diversity effect: by epoch 8, both variants reach similar accuracy and mode coverage. The main difference is early training stability. Without neutrality, epoch 1 accuracy drops from $40\%$ to $36\%$, while the diversity diagnostics remain nearly unchanged. We therefore view neutrality as a finite-budget stabilizer: it reduces the early accuracy cost of mode spreading without being strictly necessary for eventual deconcentration.

*Table 5.* Neutrality ablation on DeepSeek-R1-Distill-Qwen-7B / DAPO-17K. Each cell reports (accuracy, Avg $\hat{K}$, normalized $\hat{H}$).

| Method | Epoch 1 | Epoch 5 | Epoch 8 |
|---|---|---|---|
| G$^2$RPO with neutrality | $(40\%, 2.2, 0.3)$ | $(46\%, 4.1, 0.5)$ | $(49\%, 5.9, 0.6)$ |
| G$^2$RPO without neutrality | $(36\%, 2.1, 0.3)$ | $(45\%, 4.2, 0.5)$ | $(49\%, 6.0, 0.6)$ |

### A.4. Effect of KL Regularization

The main experiments set the KL coefficient to zero in order to isolate the effect of the proposed vector-field edit. Table 6 gives a small controlled sweep with nonzero KL. Moderate regularization ($\beta = 0.04$) dampens the growth in $\hat{K}$ but preserves a diversity advantage for G$^2$RPO over GRPO at epoch 2. A stronger coefficient ($\beta = 0.10$) largely suppresses diversity growth for both methods. This behavior is consistent with the mean-field picture: KL regularization contracts the policy toward the reference distribution and can therefore counteract the anti-collision effect.

*Table 6.* KL sweep on DeepSeek-R1-Distill-Qwen-7B. Each cell reports (token entropy, Avg $\hat{K}$, normalized $\hat{H}$).

| Method | Epoch 1 | Epoch 2 |
|---|---|---|
| G$^2$RPO | $(1.21, 2.05, 0.41)$ | $(1.28, 2.71, 0.49)$ |
| G$^2$RPO + KL ($\beta = 0.04$) | $(1.20, 1.76, 0.40)$ | $(1.26, 2.12, 0.46)$ |
| G$^2$RPO + KL ($\beta = 0.10$) | $(1.17, 1.70, 0.39)$ | $(1.17, 1.68, 0.39)$ |
| GRPO | $(1.10, 1.66, 0.35)$ | $(1.12, 1.67, 0.30)$ |
| GRPO + KL ($\beta = 0.04$) | $(1.06, 1.65, 0.37)$ | $(1.05, 1.66, 0.36)$ |
| GRPO + KL ($\beta = 0.10$) | $(1.03, 1.60, 0.31)$ | $(1.03, 1.60, 0.32)$ |

### A.5. Diversity-Oriented Baselines and Rollout-Group Size

Table 7 compares G$^2$RPO with several diversity-oriented alternatives in the same short-run setting. These rows should be interpreted as preliminary controls rather than a full benchmark: the alternatives may benefit from longer training or method-specific tuning. Within this matched setup, however, they do not reproduce the same mode growth as G$^2$RPO by epoch 2. In particular, a PPO-style entropy bonus keeps the token distribution less sharp but does not substantially increase the discovered correct-mode count, which is consistent with our distinction between token entropy and mode-level diversity.

Table 8 checks whether the improvement is merely a byproduct of sampling more rollouts per prompt. G$^2$RPO with $G = 8$ reaches higher mode coverage after one epoch than GRPO with $G = 16$, suggesting that the effect is not explained only by rollout-group size.

### A.6. Finite-Group Stability and Component Overhead

Because G$^2$RPO estimates mode masses from a finite rollout group, small $G$ can make the empirical mode histogram noisy. Table 9 reports a Monte Carlo subsampling study based on the Appendix F prompt. As expected, the estimated cluster count

*Table 7.* Preliminary diversity-oriented controls on DeepSeek-R1-Distill-Qwen-7B / DAPO-17K. Each cell reports (accuracy, Avg $\hat{K}$, normalized $\hat{H}$).

| Method | Epoch 1 | Epoch 2 |
|---|---|---|
| G$^2$RPO | $(40\%, 2.2, 0.3)$ | $(46\%, 4.1, 0.5)$ |
| GRPO + PPO-style entropy bonus | $(40\%, 1.8, 0.3)$ | $(44\%, 1.8, 0.2)$ |
| GTPO (Simoni et al., 2025) | $(39\%, 2.0, 0.3)$ | $(42\%, 2.3, 0.3)$ |
| QAE (Wu et al., 2025) | $(39\%, 1.82, 0.3)$ | $(43\%, 2.4, 0.3)$ |

*Table 8.* Initial rollout-group-size check. Each cell reports (Avg $\hat{K}$, normalized $\hat{H}$).

| Method | Step 0 | Step 67 |
|---|---|---|
| GRPO with $G = 16$ | $(1.50, 0.29)$ | $(1.66, 0.35)$ |
| G$^2$RPO with $G = 8$ | $(1.44, 0.28)$ | $(2.05, 0.41)$ |

and cluster purity stabilize as $G$ grows, while the noise rate decreases. The training setting $G = 16$ lies in a moderate-noise regime: it is not as stable as larger groups, but it already reduces the noise rate substantially relative to $G = 4$ and $G = 8$.

Table 10 reports the component time of the added embedding-and-clustering module on the development CPU. These measurements are not end-to-end training overhead, since the table excludes model forward/backward passes and system-level overlap. They show that the additional cost is dominated by embedding inference rather than DBSCAN, and they motivate future engineering work such as batching, GPU embedding, or cheaper mode features.

*Table 9.* Finite-group mode-estimation stability under Monte Carlo subsampling.

| $G$ | Estimated clusters | Cluster purity | Noise rate |
|---|---|---|---|
| 4 | $0.9 \pm 0.4$ | $0.76 \pm 0.34$ | 36% |
| 8 | $4.6 \pm 0.6$ | $0.71 \pm 0.21$ | 20% |
| 16 | $6.7 \pm 0.8$ | $0.74 \pm 0.07$ | 10% |
| 32 | $7.2 \pm 0.7$ | $0.79 \pm 0.03$ | 4% |
| 64 | $8.7 \pm 0.5$ | $0.80 \pm 0.01$ | 1% |

### A.7. Transfer Beyond AIME

Finally, we evaluate whether the math-RLVR checkpoints retain useful behavior outside AIME without task-specific retraining. Table 11 reports zero-shot checks on LiveCodeBench v6, GPQA, and MMLU-Pro. G$^2$RPO improves over GRPO on five of the six model–task pairs, with the exception of DeepSeek-7B on MMLU-Pro. This provides encouraging transfer evidence, especially on LiveCodeBench and GPQA, but we do not interpret it as a broad cross-domain superiority claim. The checkpoints were trained on math RLVR, and the transfer results should be read as a robustness check rather than the primary evaluation target.

## B. LLM as Multi-Armed Bandit and Its Mean Field Dynamics

**Overview.** This appendix summarizes a mean-field viewpoint for GRPO-style updates, using a finite multi-armed bandit abstraction that groups prompt-conditioned completions into discrete response modes, as explored in related work (Rad et al., 2026). We begin by describing how a fixed prompt and an evaluation-based clustering of completions induce a categorical distribution over modes, represented conveniently in logit coordinates. We then introduce a good–bad partition and a block decomposition that isolates the total probability mass on incorrect modes and the normalized within-block distributions over correct and incorrect modes. With this parameterization in hand, we record the expected GRPO updates in logit space, push them forward to probability space, and state the corresponding mean-field (ODE) limits. We conclude by instantiating the conditional advantages under a standard noisy binary reward model with group-wise normalization, which yields closed-form drift expressions used later in the paper.

**From token sequences to a finite mode policy.** Fix a prompt $x$. The base LLM with parameters $\omega$ induces an autoregressive distribution over token sequences $y \in \mathcal{Y}$, denoted $\pi_\omega(y \mid x)$. Let $\mathcal{V}$ be the token vocabulary, so length-$\ell$ completions

*Table 10.* Component timings for the added embedding-and-clustering module on the development CPU.

| Texts / step | Measured component time |
|---|---|
| 1,024 | 66.1 s |
| 4,096 | 243.3 s |

*Table 11.* Zero-shot transfer checks using the math-RLVR checkpoints.

| DeepSeek-R1-Distill-Qwen-7B | LCB v6 | GPQA | MMLU-Pro |
|---|---|---|---|
| G$^2$RPO | 22.9% | 54.0% | 49.7% |
| GRPO | 20.0% | 48.5% | 53.8% |
| Base | 18.9% | 50.5% | 51.7% |

| Qwen3-14B-Base | LCB v6 | GPQA | MMLU-Pro |
|---|---|---|---|
| G$^2$RPO | 26.9% | 54.0% | 68.7% |
| GRPO | 22.0% | 48.5% | 63.8% |
| Base | 25.1% | 38.9% | 51.6% |

lie in $\mathcal{V}^\ell$. Decoding stops either at an end-of-sequence token $\langle \mathrm{eos} \rangle$ or at a maximum length $L_{\max}$. We therefore restrict attention to the truncated completion space

$$\mathcal{Y}_{\leq L_{\max}} := \bigcup_{\ell=1}^{L_{\max}} \mathcal{V}^\ell, \qquad \pi_\omega^{(L)}(y \mid x) := \frac{\pi_\omega(y \mid x) \, \mathbf{1}\{y \in \mathcal{Y}_{\leq L_{\max}}\}}{\sum_{y' \in \mathcal{Y}_{\leq L_{\max}}} \pi_\omega(y' \mid x)}.$$

To obtain a prompt-level abstraction, we *coarsen* completions into finitely many evaluation modes $\mathcal{H} = \{h_1, \ldots, h_{K+M}\}$ via a surjection $\phi : \mathcal{Y}_{\leq L_{\max}} \to \mathcal{H}$, where each $h \in \mathcal{H}$ represents an evaluation-equivalence class (e.g., rubric equivalence, test-suite equivalence, or logical equivalence). This induces a categorical distribution over modes (the pushforward of $\pi_\omega^{(L)}$):

$$P_\omega(h \mid x) := \sum_{y \in \mathcal{Y}_{\leq L_{\max}} : \phi(y)=h} \pi_\omega^{(L)}(y \mid x), \qquad P_\omega(\cdot \mid x) \in \Delta^{K+M-1}.$$

For analysis we introduce *effective logits* $\theta \in \mathbb{R}^{K+M}$ such that

$$P_\omega(h_i \mid x) = \mathrm{softmax}(\theta)_i = \frac{\exp(\theta_i)}{\sum_{j=1}^{K+M} \exp(\theta_j)}.$$

The vector $\theta$ is a low-dimensional prompt-dependent coordinate that summarizes only the induced mode masses; it is defined only up to constant shifts $\theta \mapsto \theta + c\mathbf{1}$.

## B.1. A $(K+M)$-arm decomposition

**Block coordinates and diversity statistics.**  Fix $x$ and write the induced mode policy as

$$\mathbf{p} = \big( p_1, \ldots, p_K, \, p_{b_1}, \ldots, p_{b_M} \big) \in \Delta^{K+M-1}, \qquad \mathbf{p} = \mathrm{softmax}(\theta).$$

We interpret $1{:}K$ as *good* modes and $b_1{:}b_M$ as *bad* modes. Define the total bad mass

$$p := \sum_{m=1}^{M} p_{b_m} \in (0,1),$$

and the within-block normalized distributions

$$y_j := \frac{p_j}{1-p} \quad (j=1,\ldots,K), \qquad z_m := \frac{p_{b_m}}{p} \quad (m=1,\ldots,M),$$

so $y \in \Delta^{K-1}$ and $z \in \Delta^{M-1}$. Equivalently,

$$\mathbf{p} = \big( (1-p)\, y, \ p\, z \big). \tag{7}$$

We quantify within-block concentration via

$$L_y := \|y\|_2^2 = \sum_{j=1}^{K} y_j^2 \in [1/K, 1], \qquad L_z := \|z\|_2^2 = \sum_{m=1}^{M} z_m^2 \in [1/M, 1].$$

The aggregated-bad model is the special case $M = 1$ (so $z \equiv (1)$ and $L_z = 1$); keeping $M$ explicit lets us track how bad-mode concentration affects both the decay of total bad mass $p$ and later neutrality terms.

### B.2. Baseline GRPO drift in the $(K+M)$ block model

**Group-normalized rewards (notation and noise model).** Following (Rad et al., 2026), each rollout receives a Bernoulli reward $r \in \{0, 1\}$, generated from a latent correctness label good/bad. Let

$$\delta_{\mathrm{FN}} := \Pr(r = 0 \mid \text{good}), \qquad \delta_{\mathrm{FP}} := \Pr(r = 1 \mid \text{bad}), \qquad J := 1 - \delta_{\mathrm{FN}} - \delta_{\mathrm{FP}} \in [-1, 1].$$

With bad mass $p = \Pr(\text{bad})$ (so $\Pr(\text{good}) = 1 - p$), the Bernoulli mean is

$$q(p) := \mathbb{E}[r] = (1 - p)(1 - \delta_{\mathrm{FN}}) + p\, \delta_{\mathrm{FP}} = (1 - \delta_{\mathrm{FN}}) - Jp,$$

and the standard deviation is

$$\sigma(p) := \sqrt{\mathrm{Var}(r)} = \sqrt{q(p)\big(1 - q(p)\big)}.$$

GRPO normalizes rewards within each prompt-group using a $z$-score, so we work with the normalized pseudo-reward

$$\tilde{r} := \frac{r - q(p)}{\sigma(p)}.$$

This normalization motivates defining arm-wise conditional advantages $A_i := \mathbb{E}[\tilde{r} \mid I = i]$, which determin the expected GRPO drift.

**Block-symmetric conditional advantages.** For each arm $i$, define the conditional advantage

$$A_i := \mathbb{E}[\tilde{r} \mid I = i], \qquad \mathbf{A} = (A_1, \ldots, A_K, A_{b_1}, \ldots, A_{b_M}), \qquad \bar{A} := \langle \mathbf{p}, \mathbf{A} \rangle.$$

We assume *block symmetry*: conditional advantages are constant within each block,

$$A_j = a_{\mathrm{g}}(p) \quad (j \le K), \qquad A_{b_m} = a_{\mathrm{b}}(p) \quad (m \le M), \qquad \Delta r(p) := a_{\mathrm{b}}(p) - a_{\mathrm{g}}(p).$$

Writing $\alpha := 1 - p$, the mean and centered block advantages become

$$\bar{A} = \alpha\, a_{\mathrm{g}}(p) + p\, a_{\mathrm{b}}(p), \qquad a_{\mathrm{g}}(p) - \bar{A} = -p\, \Delta r(p), \qquad a_{\mathrm{b}}(p) - \bar{A} = \alpha\, \Delta r(p).$$

**Logit drift and its block form.** The logit-space expectation calculation in (Rad et al., 2026) (Proposition C.1) carries over verbatim and yields

$$\mathbb{E}[\Delta\boldsymbol{\theta} \mid \mathbf{p}] = \eta\, \mathfrak{J}(\mathbf{p})\, \mathbf{A}, \qquad \mathfrak{J}(\mathbf{p}) = \mathrm{Diag}(\mathbf{p}) - \mathbf{p}\mathbf{p}^\top.$$

Equivalently, using $\mathbb{E}[\Delta\theta_i] = \eta\, p_i(A_i - \bar{A})$ and block symmetry,

$$\mathbb{E}[\Delta\theta_j] = \eta\, p_j\big(a_{\mathrm{g}}(p) - \bar{A}\big) = -\eta\, p(1 - p)\, \Delta r(p)\, y_j, \qquad j = 1, \ldots, K, \tag{8}$$

$$\mathbb{E}[\Delta\theta_{b_m}] = \eta\, p_{b_m}\big(a_{\mathrm{b}}(p) - \bar{A}\big) = \eta\, p(1 - p)\, \Delta r(p)\, z_m, \qquad m = 1, \ldots, M. \tag{9}$$

In block form,

$$\mathbb{E}[\Delta\boldsymbol{\theta}] = \eta\, p(1 - p)\, \Delta r(p) \begin{bmatrix} -y \\ z \end{bmatrix}. \tag{10}$$

**Probability drift, total bad-mass drift, and within-block drift.** By Corollary C.3 in (Rad et al., 2026), for each arm $i$,

$$\Delta p_i = p_i(\Delta\theta_i - \mu), \qquad \mu := \langle \mathbf{p}, \Delta\boldsymbol{\theta} \rangle.$$

Combining (10) with the factorization (7) gives

$$\mu = \eta\, p(1-p)\, \Delta r(p) \left( p\, \|z\|_2^2 - (1-p)\, \|y\|_2^2 \right). \tag{11}$$

Substituting into $\Delta p_i = p_i(\Delta\theta_i - \mu)$ yields

$$\mathbb{E}[\Delta p_j] = -\eta\, p(1-p)^2\, \Delta r(p)\, y_j \left( y_j + p\|z\|_2^2 - (1-p)\|y\|_2^2 \right), \qquad j = 1, \ldots, K, \tag{12}$$

$$\mathbb{E}[\Delta p_{b_m}] = \eta\, p^2(1-p)\, \Delta r(p)\, z_m \left( z_m + (1-p)\|y\|_2^2 - p\|z\|_2^2 \right), \qquad m = 1, \ldots, M. \tag{13}$$

Summing (13) over $m$ (using $\sum_m z_m = 1$) gives the drift of the total bad mass:

$$\mathbb{E}[\Delta p] := \sum_{m=1}^{M} \mathbb{E}[\Delta p_{b_m}] = \eta\, [p(1-p)]^2\, \Delta r(p) \left( \|y\|_2^2 + \|z\|_2^2 \right). \tag{14}$$

To extract the within-block dynamics, note that $\alpha = 1 - p$ and $\Delta\alpha = -\Delta p$. Since $y_j = p_j/\alpha$ and $z_m = p_{b_m}/p$, the quotient identities are

$$\Delta y_j = \frac{1}{\alpha}\left( \Delta p_j - y_j\, \Delta\alpha \right) = \frac{1}{1-p}\left( \Delta p_j + y_j\, \Delta p \right), \qquad \Delta z_m = \frac{1}{p}\left( \Delta p_{b_m} - z_m\, \Delta p \right). \tag{15}$$

Substituting (12) and (14) yields

$$\mathbb{E}[\Delta y_j] = -\eta\, p(1-p)\, \Delta r(p)\, y_j \left( y_j - \|y\|_2^2 \right), \qquad j = 1, \ldots, K, \tag{16}$$

$$\mathbb{E}[\Delta z_m] = \eta\, p(1-p)\, \Delta r(p)\, z_m \left( z_m - \|z\|_2^2 \right), \qquad m = 1, \ldots, M. \tag{17}$$

Equivalently,

$$\mathbb{E}[\Delta y] = -\eta\, p(1-p)\, \Delta r(p) \left( y \odot y - \|y\|_2^2\, y \right), \qquad \mathbb{E}[\Delta z] = \eta\, p(1-p)\, \Delta r(p) \left( z \odot z - \|z\|_2^2\, z \right). \tag{18}$$

**Noisy GRPO specialization.** For the group-normalized noisy Bernoulli model summarized above (and derived in (Rad et al., 2026)),

$$a_{\mathrm{g}}(p) = \frac{J\, p}{\sigma(p)}, \qquad a_{\mathrm{b}}(p) = -\frac{J(1-p)}{\sigma(p)}, \qquad \Delta r(p) = -\frac{J}{\sigma(p)}.$$

Plugging into (14) and (18) gives

$$\mathbb{E}[\Delta p] = -\eta\, \frac{J}{\sigma(p)}\, [p(1-p)]^2 \left( \|y\|_2^2 + \|z\|_2^2 \right), \tag{19}$$

$$\mathbb{E}[\Delta y] = \eta\, \frac{J}{\sigma(p)}\, p(1-p) \left( y \odot y - \|y\|_2^2\, y \right), \tag{20}$$

$$\mathbb{E}[\Delta z] = -\eta\, \frac{J}{\sigma(p)}\, p(1-p) \left( z \odot z - \|z\|_2^2\, z \right). \tag{21}$$

**Continuous-time ODE form.** Under the mean-field scaling $t = n$, the drift ODEs become

$$\dot{y} = \kappa(p)\left( y \odot y - \|y\|_2^2\, y \right), \qquad \dot{z} = -\kappa(p)\left( z \odot z - \|z\|_2^2\, z \right), \qquad \dot{p} = -\eta\frac{J}{\sigma(p)}\, [p(1-p)]^2 \left( \|y\|_2^2 + \|z\|_2^2 \right), \tag{22}$$

where

$$\kappa(p) := \eta\, \frac{J}{\sigma(p)}\, p(1-p). \tag{23}$$

## B.3. Within-block collision dynamics in the $(K, M)$-arm model

The mean-field ODEs (22) imply that the within-block compositions evolve according to the *collision* vector field

$$V(s) := s \odot s - \|s\|_2^2 \, s,$$

with opposite signs on the good and bad blocks. The scalar factor $\kappa(p) = \eta \frac{J}{\sigma(p)} \, p(1 - p)$ only rescales time along these trajectories.

To remove this time-rescaling, introduce the *internal time*

$$\tau(t) := \int_0^t \kappa\big(p(s)\big) \, ds, \qquad \frac{d\tau}{dt} = \kappa\big(p(t)\big). \tag{24}$$

In $\tau$-time, the within-block dynamics take the form

$$\frac{dy}{d\tau} = V(y) = y \odot y - \|y\|_2^2 \, y, \tag{25}$$

$$\frac{dz}{d\tau} = -V(z) = -\big(z \odot z - \|z\|_2^2 \, z\big). \tag{26}$$

The outer mass evolves as

$$\frac{dp}{d\tau} = -\, p(1 - p)\Big(\|y\|_2^2 + \|z\|_2^2\Big). \tag{27}$$

We therefore focus on the collision flows (25)–(26); the outer mass $p$ enters only through the time change (27). We now record their basic structural properties.

**Block coordinates and softmax decoupling**  The next lemmas make precise a simple decoupling: the within-good composition $y$ depends only on the good logits, while the within-bad composition $z$ depends only on the bad logits.

**Lemma B.1** (Within-block softmax cancellation). *Let* $\mathbf{p} = \mathrm{softmax}(\theta) \in \Delta^{K+M-1}$ *with* $\theta \in \mathbb{R}^{K+M}$, *and define* $p = \sum_{m=1}^M p_{b_m}$, $y_j = p_j/(1 - p)$ *for* $j \in [K]$, *and* $z_m = p_{b_m}/p$ *for* $m \in [M]$. *Then*

$$y = \mathrm{softmax}(\theta_{\mathrm{good}}), \qquad y_j = \frac{e^{\theta_j}}{\sum_{k=1}^K e^{\theta_k}},$$

*and likewise*

$$z = \mathrm{softmax}(\theta_{\mathrm{bad}}), \qquad z_m = \frac{e^{\theta_{b_m}}}{\sum_{\ell=1}^M e^{\theta_{b_\ell}}}.$$

**Lemma B.2** (Pushforward map: logits → within-block composition). *Let* $s = \mathrm{softmax}(u) \in \Delta^{d-1}$. *For any increment* $\Delta u \in \mathbb{R}^d$,

$$\Delta s = \big(\mathrm{Diag}(s) - ss^\top\big)\Delta u = s \odot \Big(\Delta u - \langle s, \Delta u\rangle \mathbf{1}\Big).$$

*In particular, by Lemma B.1,* $\partial y/\partial \theta_{b_m} = 0$ *for all* $m$ *and* $\partial z/\partial \theta_j = 0$ *for all* $j$.

*Proof sketch.* Differentiate $s_i = e^{u_i - \log \sum_k e^{u_k}}$; this is the standard Jacobian of softmax. □

### B.3.1. GEOMETRY OF THE GOOD-BLOCK COLLISION FLOW

We now focus on collision of the good-block

$$\frac{dy}{d\tau} = y \odot y - L_y \, y, \qquad L_y = \|y\|_2^2. \tag{28}$$

Define also $S_3(y) := \sum_{j=1}^K y_j^3$.

**Lemma B.3** (Simplex invariance and Lyapunov potential). *The simplex $\Delta^{K-1}$ is forward invariant for* (28). *Moreover, the function*

$$\Phi(y) := \frac{1}{3} \sum_{j=1}^{K} y_j^3 - \frac{1}{4} \Big( \sum_{j=1}^{K} y_j^2 \Big)^2 = \frac{1}{3} S_3(y) - \frac{1}{4} L_y^2$$

*satisfies $\nabla \Phi(y) = (y_j^2 - L_y y_j)_j$, hence*

$$\frac{dy}{d\tau} = \nabla \Phi(y), \qquad \frac{d}{d\tau} \Phi\big(y(\tau)\big) = \|\nabla \Phi\big(y(\tau)\big)\|_2^2 \geq 0.$$

*Proof sketch.* Using (28), we have

$$\frac{d}{d\tau} \sum_{j=1}^{K} y_j = \sum_{j=1}^{K} \frac{dy_j}{d\tau} = \sum_{j=1}^{K} y_j(y_j - L_y) = \Big( \sum_{j=1}^{K} y_j^2 \Big) - L_y \Big( \sum_{j=1}^{K} y_j \Big) = L_y - L_y = 0,$$

so the total mass $\sum_j y_j$ is preserved. Moreover, if $y_j = 0$ then $\frac{dy_j}{d\tau} = y_j(y_j - L_y) = 0$, so the vector field is tangent to each face $\{y_j = 0\}$ and nonnegativity is preserved. Hence $\Delta^{K-1}$ is forward invariant.

For the potential, direct differentiation gives $\frac{\partial \Phi}{\partial y_j} = y_j^2 - L_y y_j$, i.e., $\nabla \Phi(y) = (y_j^2 - L_y y_j)_j$, which matches the right-hand side of $\frac{dy_j}{d\tau}$. The claimed monotonicity follows from the chain rule: $\frac{d}{d\tau} \Phi(y(\tau)) = \langle \nabla \Phi(y(\tau)), \frac{dy}{d\tau} \rangle = \|\nabla \Phi(y(\tau))\|_2^2 \geq 0$. $\qquad \square$

**Lemma B.4** (Monotone concentration and the collision identity). *Along* (28),

$$\frac{d}{d\tau} L_y = 2\big(S_3(y) - L_y^2\big) = 2 \sum_{j=1}^{K} y_j (y_j - L_y)^2 \geq 0.$$

*Equality holds iff $y$ is uniform on its support. In particular, $L_y(\tau) \in [1/K, 1]$ and it is strictly increasing away from the tie/saddle sets.*

*Proof.* Differentiate $L_y = \sum_j y_j^2$ and substitute $\frac{dy_j}{d\tau} = y_j(y_j - L_y)$. The variance form follows by expanding $\sum_j y_j(y_j - L_y)^2$. $\qquad \square$

**Proposition B.5** (Equilibria of the collision flow). *A point $y^\star \in \Delta^{K-1}$ is stationary for* (28) *iff each coordinate satisfies $y_j^\star \in \{0, L_y^\star\}$, where $L_y^\star = \|y^\star\|_2^2$. Equivalently, for any support size $m \in \{1, \ldots, K\}$, the vectors uniform on a support of size $m$ are precisely the equilibria:*

$$\mathcal{E}_m = \Big\{ y : \; y_{i_1} = \cdots = y_{i_m} = 1/m, \; y_j = 0 \text{ otherwise} \Big\}, \qquad \mathcal{E} = \bigcup_{m=1}^{K} \mathcal{E}_m.$$

### Order preservation, global convergence, and rates in internal time

**Lemma B.6** (Order preservation). *For $i \neq j$, let $\delta_{ij}(\tau) := y_i(\tau) - y_j(\tau)$. Then*

$$\delta_{ij}' = \delta_{ij}(y_i + y_j - L_y), \qquad \Rightarrow \qquad \operatorname{sign} \delta_{ij}(\tau) = \operatorname{sign} \delta_{ij}(0) \;\; \forall \tau.$$

*Hence the identity of the largest coordinate is preserved: if the maximizer is unique at $\tau = 0$, it remains the unique maximizer for all $\tau > 0$.*

*Proof.* From (28), $\frac{dy_k}{d\tau} = y_k(y_k - L_y)$. Thus

$$\frac{d}{d\tau} \delta_{ij} = \frac{dy_i}{d\tau} - \frac{dy_j}{d\tau} = y_i(y_i - L_y) - y_j(y_j - L_y) = (y_i - y_j)(y_i + y_j - L_y) = \delta_{ij}(y_i + y_j - L_y).$$

The sign claim follows by solving the scalar linear ODE $\delta_{ij}(\tau) = \delta_{ij}(0) \exp\big( \int_0^\tau (y_i + y_j - L_y) \, du \big)$. $\qquad \square$

**Theorem B.7** (Global limit for the good block). *Let $y(0)$ lie in the interior of $\Delta^{K-1}$ and assume no coordinate ties initially. Let $m = \arg\max_i y_i(0)$ (unique). Then the solution of (28) satisfies*

$$y(\tau) \to e_m \qquad (\tau \to \infty).$$

*Every non-vertex equilibrium (uniform on an $m$-subset with $m \geq 2$) is a saddle whose stable manifold is contained in the union of tie hyperplanes $\{y_i = y_j\}$.*

*Proof sketch.* $\Phi$ is a strict Lyapunov function off the equilibrium set (Lemma B.3). Order preservation (Lemma B.6) rules out convergence to a mixed-support equilibrium unless the trajectory lies in a tie set. Generic initial data avoids these sets, forcing convergence to the vertex corresponding to the initial winner. $\square$

**Proposition B.8** (Exponential polarization in $\tau$). *Under the assumptions of Theorem B.7, let $m$ be the winning index and write $\varepsilon_i(\tau) := y_i(\tau)$ for $i \neq m$. Then, as $\tau \to \infty$,*

$$\varepsilon_i(\tau) = c_i e^{-\tau}(1 + o(1)), \qquad 1 - y_m(\tau) = \Big(\sum_{i \neq m} c_i\Big) e^{-\tau}(1 + o(1)), \qquad 1 - L_y(\tau) = \Theta(e^{-\tau}),$$

*for constants $c_i > 0$ determined by the trajectory.*

*Proof sketch.* Linearize $y_i' = y_i(y_i - L_y)$ at $e_m$ on the simplex tangent space: transverse modes satisfy $\varepsilon_i' = -\varepsilon_i + O(\varepsilon^2)$. $\square$

### B.3.2. BAD-BLOCK DYNAMICS: SIGN REVERSAL

The within-bad ODE (26) is the time-reversal of the good collision flow. Equivalently, writing $\rho := -\tau$ gives $dz/d\rho = z \odot z - L_z z$ on $\Delta^{M-1}$.

**Corollary B.9** (Bad-block limits). *If $\tau \to +\infty$ corresponds to the forward direction of the coupled system (e.g. $\kappa \geq 0$), then (26) is a gradient descent flow for the same potential and satisfies:*

- $L_z(\tau)$ *is nonincreasing and converges to $1/M$ for interior initial data;*

- $z(\tau) \to \frac{1}{M}\mathbf{1}$ *(uniform on the full support).*

*If instead the effective direction is reversed (e.g. $\kappa \leq 0$), then $z$ follows the forward collision flow and generically polarizes to a vertex selected by the initial maximizer.*

**Coupling to the outer mass $p$ and logit envelopes**   In the multi-bad model, the total bad mass couples to the collision concentrations $L_y(\tau)$ and $L_z(\tau)$ via

$$\frac{dp}{d\tau} = -p(1-p)\big(L_y(\tau) + L_z(\tau)\big). \tag{29}$$

Equivalently, for the logit $L(\tau) := \log \frac{p(\tau)}{1-p(\tau)}$,

$$\frac{dL}{d\tau} = -(L_y(\tau) + L_z(\tau)), \qquad L(\tau) = L(0) - \int_0^\tau (L_y(u) + L_z(u))\, du. \tag{30}$$

**Corollary B.10** (Outer envelopes and internal hitting-time bracket). *Since $L_y \in [1/K, 1]$ and $L_z \in [1/M, 1]$, we have for all $\tau \geq 0$:*

$$L(0) - 2\tau \;\leq\; L(\tau) \;\leq\; L(0) - \Big(\frac{1}{K} + \frac{1}{M}\Big)\tau.$$

*Equivalently, writing $p_0 = p(0)$,*

$$\frac{1}{1 + \frac{1-p_0}{p_0} e^{2\tau}} \;\leq\; p(\tau) \;\leq\; \frac{1}{1 + \frac{1-p_0}{p_0} e^{(\frac{1}{K} + \frac{1}{M})\tau}}.$$

*Moreover, for any target $p_\star \in (0,1)$, the internal time $\tau_\star$ defined by $p(\tau_\star) = p_\star$ satisfies*

$$\frac{1}{2} \log \frac{p_0(1 - p_\star)}{(1 - p_0)p_\star} \;\leq\; \tau_\star \;\leq\; \frac{1}{\frac{1}{K} + \frac{1}{M}} \log \frac{p_0(1 - p_\star)}{(1 - p_0)p_\star}.$$

B.3.3. OPTIONAL: A ONE-SCALAR REPRESENTATION FOR THE GOOD COLLISION FLOW

The ODE (28) admits a convenient scalar parametrization. Let $q = y(0) \in \Delta^{K-1}$ and define moment sums

$$M_r(I) := \sum_{j=1}^{K} \frac{q_j^r}{(1 - Iq_j)^r}, \qquad I \in \left[0, \frac{1}{q_*}\right), \qquad q_* := \max_j q_j.$$

**Lemma B.11** (Scalar parametrization). *There exists a strictly increasing scalar $I(\tau)$ with $I(0) = 0$ such that*

$$y_j(\tau) = \frac{\frac{q_j}{1 - I(\tau)q_j}}{\sum_{\ell=1}^{K} \frac{q_\ell}{1 - I(\tau)q_\ell}} = \left. \frac{\frac{q_j}{1 - Iq_j}}{M_1(I)} \right|_{I=I(\tau)}.$$

*Moreover, $\tau$ and $I$ are related by*

$$\frac{d\tau}{dI} = M_1(I), \qquad \tau(I) = \int_0^I M_1(z)\, dz = -\sum_{j=1}^{K} \log(1 - Iq_j).$$

*Finally,*

$$L_y(\tau(I)) = \frac{M_2(I)}{M_1(I)^2}, \qquad S_3(\tau(I)) = \frac{M_3(I)}{M_1(I)^3}.$$

*Remark* B.12. The bad-block analogue replaces $(1 - Iq_j)$ by $(1 + Iq_m)$ and satisfies $z(\tau) \to \frac{1}{M}\mathbf{1}$ in forward $\tau$.

# C. Adding Granularity into the Advantage

A standard extension of GRPO adds a per-sample *bonus* term to the group-normalized advantage, for example to encourage diversity, enforce formatting constraints, or shape exploration. In a mean-field analysis, it is natural to model this bonus as a deterministic control signal $B(\mathbf{p})$ added to the pseudo-advantage. In the $(K+M)$ block model, such a control couples three quantities: the total bad mass $p$, the within-good distribution $y \in \Delta^{K-1}$, and the within-bad distribution $z \in \Delta^{M-1}$. We make this coupling explicit by deriving the bonus-to-drift map and isolating how *granularity* alters the resulting dynamics.

## C.1. GRPO bonus meechanism in the $K + M$ model

As in the earlier analysis, the expected GRPO/REINFORCE logit update takes the form

$$\mathbb{E}[\Delta\theta \mid \mathbf{p}] = \eta \, \mathbf{J}(\mathbf{p}) \, A, \qquad \mathbf{J}(\mathbf{p}) = \text{Diag}(\mathbf{p}) - \mathbf{p}\mathbf{p}^\top,$$

Introducing a pseudo-advantage (bonus) vector $B$ corresponds to replacing $A$ by $\tilde{A} = A + B$. The incremental contribution of the bonus to the probability drift is then

$$\mathbb{E}[\Delta\mathbf{p}]_{\text{bonus}} = \eta \, \mathbf{J}(\mathbf{p})^2 \, B. \tag{31}$$

**Lemma C.1** (Shift invariance). *For any scalar $c \in \mathbb{R}$, $\mathbf{J}(\mathbf{p})^2(B + c\mathbf{1}) = \mathbf{J}(\mathbf{p})^2 B$. In particular, global (all-arm) uniform bonuses are invisible to GRPO.*

*Proof.* Since $\mathbf{J}(\mathbf{p})\mathbf{1} = 0$, we also have $\mathbf{J}(\mathbf{p})^2\mathbf{1} = 0$, hence $\mathbf{J}(\mathbf{p})^2(B + c\mathbf{1}) = \mathbf{J}(\mathbf{p})^2 B + c\mathbf{J}(\mathbf{p})^2\mathbf{1} = \mathbf{J}(\mathbf{p})^2 B$. □

**Bonus structure and "granularity".** At full generality, the bonus may be arm-dependent on both blocks:

$$B = \big(B_1, \ldots, B_K, \; B_{b_1}, \ldots, B_{b_M}\big),$$

where $B_j$ and $B_{b_m}$ may depend on $(p, y, z)$. This *granular* form allows distinct auxiliary signals across different bad modes and therefore can reshape the within-bad distribution $z$ directly. We consider the most simple design for the bad block: we *do not attempt to distinguish incorrect modes*, and instead apply the *same scalar bonus* to every bad arm. Concretely, A common design choice for the bad block, however, is to *not* distinguish incorrect modes and instead apply the *same scalar bonus* to every bad arm. Concretely, this corresponds to the restricted structure

$$B = \big(B_1, \ldots, B_K, \underbrace{B_b, \ldots, B_b}_{M \text{ times}}\big), \tag{32}$$

where the within-good terms $B_j = B_j(y)$ may be mode-dependent (granular), while $B_b$ is a scalar that may depend on $(p, y, z)$.

*semantics* of the aggregated bad event: each incorrect rollout is a Monte Carlo sample from "bad", so the same per-sample scalar is applied to all incorrect responses within a group. Importantly, one does not divide $B_b$ by the number of bad samples; the expected contribution already scales with the bad mass through sampling.

This matches the intended *semantics* of the aggregated bad arm: each incorrect rollout is a Monte Carlo sample from the event "bad", so the same per-sample scalar should be applied to all incorrect responses within a group.

**Within-block centering.** To describe how a bonus reshapes a distribution inside a simplex, it is convenient to distinguish two moments.

**Definition C.2** (Mean- and second-moment centering). Fix $x \in \Delta^{n-1}$ and a bonus vector $u(x) \in \mathbb{R}^n$. We say $u$ is $x$-*centered* if $\langle x, u(x) \rangle = 0$. We say $u$ is $x^2$-*centered* if $\langle x \odot x, u(x) \rangle = 0$ (equivalently $\sum_i x_i^2 u_i(x) = 0$).

The first condition removes the uniform component under the *sampling* measure $x$, while the second removes the component that leaks into the block mass under the quadratic map $\mathbf{J}(\mathbf{p})^2$.

**Lemma C.3** (Centered within-good bonus: drift on $y$ and leak into $p$). *Assume the bonus is applied only to the good block and is $y$-centered, i.e., $B_b = 0$ and $\sum_{j=1}^{K} y_j B_j(y) = 0$. Then, to first order in the step size $\eta$,*

$$\mathbb{E}[\Delta y_j]_{\text{bonus}} = \eta\,(1-p)\,y_j\left(y_j B_j(y) - \sum_{\ell=1}^{K} y_\ell^2 B_\ell(y)\right), \qquad j = 1, \ldots, K, \tag{33}$$

$$\mathbb{E}[\Delta p]_{\text{bonus}} = -\,\eta\,p\,(1-p)^2 \sum_{\ell=1}^{K} y_\ell^2 B_\ell(y). \tag{34}$$

*Equivalently, in vector form, $\mathbb{E}[\Delta y]_{\text{bonus}} = \eta(1-p)\,\mathcal{C}_y(B_g)$, where $\mathcal{C}_y$ is the collision operator defined in (38).*

*Proof.* Let $\mathbf{p} \in \Delta^{K+M-1}$ denote the full probability vector, and let

$$p := \sum_{m=1}^{M} p_{K+m} \qquad \text{and} \qquad q := 1 - p = \sum_{j=1}^{K} p_j$$

be the total bad and good masses, respectively. Parameterize the good block by $p_j = q\,y_j$ with $y \in \Delta^{K-1}$. Assume the bonus has the block form $B = (B_g(y), 0)$, i.e. $B_{K+m} = 0$ for all bad arms.

Recall that, before adding any bonus term to the advantage, $\mathbb{E}[\Delta\boldsymbol{\theta} \mid \mathbf{p}] = \eta\,\mathbf{J}(\mathbf{p})\,A$. Thus upon replacing $A$ with $A + B$: $\mathbb{E}[\Delta\boldsymbol{\theta} \mid \mathbf{p}]_{\text{bonus}} = \eta\,\mathbf{J}(\mathbf{p})\,B$. The next step is to apply the first order approximation $\mathbb{E}[\Delta\mathbf{p} \mid \mathbf{p}] \approx \mathfrak{J}(\mathbf{p})\,\mathbb{E}[\Delta\boldsymbol{\theta} \mid \mathbf{p}]$ (we drop conditioning on $\mathbf{p}$ hereafter for the sake of brevity). This results in:

$$\mathbb{E}[\Delta\mathbf{p}]_{\text{bonus}} = \eta\,\mathbf{J}(\mathbf{p})^2\,B, \qquad \mathbf{J}(\mathbf{p}) = \text{Diag}(\mathbf{p}) - \mathbf{p}\mathbf{p}^\top. \tag{35}$$

The $y$-centering condition implies

$$\mathbf{p}^\top B = \sum_{j=1}^{K} p_j B_j(y) = q \sum_{j=1}^{K} y_j B_j(y) = 0.$$

Hence

$$\mathbf{J}(\mathbf{p})B = \left(\text{Diag}(\mathbf{p}) - \mathbf{p}\mathbf{p}^\top\right)B = \text{Diag}(\mathbf{p})B,$$

and therefore

$$\begin{aligned}
\mathbf{J}(\mathbf{p})^2 B &= \mathbf{J}(\mathbf{p})\,\text{Diag}(\mathbf{p})B \\
&= \left(\text{Diag}(\mathbf{p}) - \mathbf{p}\mathbf{p}^\top\right)\text{Diag}(\mathbf{p})B \\
&= \text{Diag}(\mathbf{p})^2 B - \mathbf{p}\,\mathbf{p}^\top\left(\text{Diag}(\mathbf{p})B\right) \\
&= \mathbf{p} \odot \mathbf{p} \odot B - \langle \mathbf{p}, \mathbf{p} \odot B\rangle\,\mathbf{p},
\end{aligned}$$

where $\langle \cdot, \cdot \rangle$ is the Euclidean inner product. Define the scalar

$$S(y) := \sum_{\ell=1}^{K} y_\ell^2 B_\ell(y).$$

Because $B$ is zero on the bad block,

$$\langle \mathbf{p}, \mathbf{p} \odot B\rangle = \sum_{\ell=1}^{K} p_\ell^2 B_\ell(y) = \sum_{\ell=1}^{K}(q^2 y_\ell^2)B_\ell(y) = q^2 S(y).$$

**Leak into $p$.** For a bad arm index $i > K$, we have $B_i = 0$, so the $i$th component of $\mathbf{p} \odot \mathbf{p} \odot B$ vanishes and

$$\mathbb{E}[\Delta p_i]_{\text{bonus}} = -\eta\,\langle \mathbf{p}, \mathbf{p} \odot B\rangle\,p_i = -\eta\,q^2 S(y)\,p_i.$$

Summing over all bad arms yields

$$\mathbb{E}[\Delta p]_{\text{bonus}} = \sum_{i>K} \mathbb{E}[\Delta p_i]_{\text{bonus}} = -\eta\,q^2 S(y) \sum_{i>K} p_i = -\eta\,p\,q^2 S(y),$$

which is exactly (34) since $q = 1 - p$.

**Drift on $y$.** For a good arm $j \leq K$,

$$
\mathbb{E}[\Delta p_j]_{\text{bonus}} = \eta \Big( p_j^2 B_j(y) - \langle \mathbf{p}, \mathbf{p} \odot B \rangle \, p_j \Big)
$$
$$
= \eta \Big( q^2 y_j^2 B_j(y) - q^2 S(y) \cdot q y_j \Big) = \eta \Big( q^2 y_j^2 B_j(y) - q^3 y_j S(y) \Big).
$$

Using $y_j = p_j/q$ and the first-order quotient rule,

$$
\Delta y_j = \Delta \Big( \frac{p_j}{q} \Big) \approx \frac{\Delta p_j}{q} + \frac{p_j}{q^2} \Delta p = \frac{\Delta p_j}{q} + \frac{y_j}{q} \Delta p,
$$

we obtain

$$
\mathbb{E}[\Delta y_j]_{\text{bonus}} = \frac{1}{q} \mathbb{E}[\Delta p_j]_{\text{bonus}} + \frac{y_j}{q} \mathbb{E}[\Delta p]_{\text{bonus}}
$$
$$
= \frac{1}{q} \eta \Big( q^2 y_j^2 B_j(y) - q^3 y_j S(y) \Big) + \frac{y_j}{q} \Big( -\eta \, p \, q^2 S(y) \Big)
$$
$$
= \eta \Big( q \, y_j^2 B_j(y) - q^2 y_j S(y) - pq \, y_j S(y) \Big)
$$
$$
= \eta \, q \, y_j \Big( y_j B_j(y) - S(y) \Big),
$$

since $q^2 + pq = q(q+p) = q$. Substituting $q = 1-p$ yields (33). The vector-form statement follows by identifying the componentwise action of $\mathcal{C}_y(B_g)$ as $(\mathcal{C}_y(B_g))_j = y_j \big( y_j B_j(y) - \sum_\ell y_\ell^2 B_\ell(y) \big)$. $\qquad \square$

**Corollary C.4** (Centered within-bad bonus: drift on $z$ and leak into $p$). *Assume the bonus is applied only to the bad block and is $z$-centered, i.e. $B_j = 0$ for $j \leq K$ and $\sum_{m=1}^{M} z_m C_m(z) = 0$ where $B_{b_m} = C_m(z)$ for $m = 1, \ldots, M$. Then, to first order in $\eta$,*

$$
\mathbb{E}[\Delta z_m]_{\text{bonus}} = \eta \, p \, z_m \Big( z_m C_m(z) - \sum_{r=1}^{M} z_r^2 C_r(z) \Big), \qquad m = 1, \ldots, M, \tag{36}
$$

$$
\mathbb{E}[\Delta p]_{\text{bonus}} = + \eta \, p^2 \, (1-p) \sum_{r=1}^{M} z_r^2 C_r(z). \tag{37}
$$

*Remark* C.5 (Neutrality conditions). Under Lemma C.3, a $y$-centered good bonus is *neutral in $p$* (i.e. $\mathbb{E}[\Delta p]_{\text{bonus}} = 0$) if and only if it is also $y^2$-centered: $\sum_j y_j^2 B_j(y) = 0$. The analogous neutrality condition for a $z$-centered bad bonus is $\sum_m z_m^2 C_m(z) = 0$ by (37). Most useful "rare-mode amplifiers" are *not* second-moment centered; the neutralization schemes later compensate the resulting leak in $p$ by introducing an additional scalar channel (a uniform bad bonus $B_b$ and/or a granular bad correction).

## C.2. Why inverse-probability bonuses are the natural choice

The role of the mode-dependent terms $B_j(y)$ and $C_m(z)$ is *not* to introduce a new direction in the within-block geometry, but to provide a *single scalar knob* that only rescales the existing collision fields $V(y) = y \odot y - \|y\|_2^2 y$ and $V(z) = z \odot z - \|z\|_2^2 z$. This is exactly what we need for baseline preservation: if a bonus injects any component *not collinear* with $V(y)$ or $V(z)$, then the resulting dynamics typically cannot be absorbed into a small number of scalar degrees of freedom, and we lose the ability to keep $\dot{p}$ and $\dot{z}$ baseline while shaping only $y$.

**The collision operator.** For a distribution $x \in \Delta^{n-1}$ and a within-block bonus vector $u(x) \in \mathbb{R}^n$, define the operator

$$
\mathcal{C}_x(u) := x \odot (x \odot u(x)) - \langle x, \, x \odot u(x) \rangle \, x, \qquad \sum_{i=1}^{n} \mathcal{C}_x(u)_i = 0, \tag{38}
$$

where $\langle a, b \rangle = \sum_i a_i b_i$ and $\odot$ is the Hadamard product. When $u$ is $x$-centered, this operator is precisely the within-block drift appearing in Lemma C.3 and Corollary C.4, up to the block-mass prefactor. For the centered bonuses considered above, the induced within-block drift takes the form

$$
\mathbb{E}[\Delta x]_{\text{bonus}} = \gamma \, \mathcal{C}_x(u),
$$

where $x \in \Delta^{n-1}$ denotes the within-block distribution, $u(x)$ is the corresponding within-block bonus vector, $\mathcal{C}_x$ is defined in (38), and $\gamma$ is the appropriate block-mass prefactor (e.g., $\gamma = \eta(1-p)$ for the good block and $\gamma = \eta p$ for the bad block).

**Sanity check: global vs. block-uniform constants.** A bonus that is uniform across *all* arms, $B = c\mathbf{1}$, induces *zero* drift by shift invariance: $\mathbf{J}(\mathbf{p})^2(B + c\mathbf{1}) = \mathbf{J}(\mathbf{p})^2 B$ and $\mathbf{J}(\mathbf{p})\mathbf{1} = 0$. The only nontrivial "uniform" case is therefore *block*-uniform with different constants on the good and bad blocks,

$$B = (\underbrace{c_g, \ldots, c_g}_{K}, \underbrace{c_b, \ldots, c_b}_{M}).$$

In this case the induced within-block dynamics are pure rescalings of the collision fields:

$$\mathbb{E}[\Delta y]_{\text{bonus}} = \eta\, p(1-p)(c_g - c_b)\, V(y), \qquad \mathbb{E}[\Delta z]_{\text{bonus}} = \eta\, p(1-p)(c_b - c_g)\, V(z), \tag{39}$$

while the block-mass channel generally drifts as

$$\mathbb{E}[\Delta p]_{\text{bonus}} = \eta\, [p(1-p)]^2 (\|y\|_2^2 + \|z\|_2^2)(c_b - c_g).$$

Thus block-uniform bonuses cannot target individual modes; they only rescale the existing collision geometry (and may shift mass between blocks).

**Lemma C.6** (Centered reciprocal bonuses are eigenvectors of $\mathcal{C}_x$). *Fix $n \geq 2$ and define the centered reciprocal bonus*

$$B_i(x) = a\left(\frac{1}{nx_i} - 1\right), \qquad i = 1, \ldots, n, \tag{40}$$

*for some scalar $a \in \mathbb{R}$. Then:*

1. **Centered under $x$:** $\langle x, B(x) \rangle = 0$ *for all $x \in \Delta^{n-1}$.*

2. **Pure $V$-alignment:** $\mathcal{C}_x(B) = -a\, V(x)$ *for all $x \in \Delta^{n-1}$.*

*Proof.* For centering,

$$\langle x, B(x) \rangle = a \sum_{i=1}^{n} x_i \left(\frac{1}{nx_i} - 1\right) = a \sum_{i=1}^{n} \left(\frac{1}{n} - x_i\right) = a(1 - 1) = 0.$$

For the operator identity, note that $x \odot B(x) = a\left(\frac{1}{n}\mathbf{1} - x\right)$, so

$$x \odot (x \odot B(x)) = a\left(\frac{1}{n}x - x \odot x\right), \qquad \langle x, x \odot B(x) \rangle = a\left(\frac{1}{n} - \|x\|_2^2\right).$$

Substituting into (38) gives

$$\mathcal{C}_x(B) = a\left(\frac{1}{n}x - x \odot x\right) - a\left(\frac{1}{n} - \|x\|_2^2\right)x = -a\left(x \odot x - \|x\|_2^2 x\right) = -a\, V(x).$$

$\square$

*Remark* C.7 (Interpretation). The reciprocal term $1/(nx_i)$ is the simplest "rare-mode amplifier": it allocates larger bonus to smaller coordinates and becomes strong near the boundary ($x_i \to 0$), which is exactly the regime where diversity needs active stabilization. The subtraction of 1 (equivalently, centering) removes the block-uniform component, helping prevent unintended interactions with the block-mass channel.

**Theorem C.8** (Essential uniqueness under permutation-equivariant scalar bonuses). *Fix $n \geq 3$. Consider any coordinate-wise, permutation-equivariant bonus family of the form $B_i(x) = f(x_i)$ (same scalar function $f$ for all coordinates). Suppose its induced within-block drift is always a* pure rescaling *of the collision field:*

$$\mathcal{C}_x(B) = \alpha(x)\, V(x) \qquad \text{for all } x \in \Delta^{n-1}, \tag{41}$$

*for some scalar functional $\alpha(x)$. Then necessarily*

$$f(s) = \frac{c}{s} + d \quad \text{for constants } c, d.$$

*If in addition we impose the normalization $B(\mathbf{1}/n) = 0$, then the family reduces to the centered reciprocal form (40) up to an overall scale.*

*Proof.* Write $\mathcal{C}_x(B)_i = x_i^2 f(x_i) - S(x)\, x_i$ where $S(x) = \sum_{j=1}^n x_j^2 f(x_j)$. The collinearity condition (41) implies, for each coordinate $i$,

$$x_i^2 f(x_i) - S(x)\, x_i = \alpha(x)\big(x_i^2 - \|x\|_2^2 x_i\big).$$

For any $i$ with $x_i > 0$, divide by $x_i$ to obtain

$$x_i f(x_i) = \alpha(x)\, x_i + \big(S(x) - \alpha(x)\|x\|_2^2\big). \tag{42}$$

Define $g(s) := s f(s)$. From (42), for any $x \in \Delta^{n-1}$ and any indices $i, k$ with $x_i, x_k > 0$,

$$g(x_i) - g(x_k) = \alpha(x)\,(x_i - x_k). \tag{43}$$

Thus, for each fixed $x$, the slope $\alpha(x)$ equals the divided difference of $g$ evaluated at any two positive coordinates of $x$.

Now pick any $a, b \in (0, 1)$ with $a + b < 1$ and consider the simplex point

$$x = (a, b, 1 - a - b, 0, \dots, 0) \in \Delta^{n-1}.$$

Applying (43) to the pairs $(a, b)$, $(a, 1 - a - b)$, and $(b, 1 - a - b)$ yields

$$\frac{g(a) - g(b)}{a - b} = \frac{g(a) - g(1 - a - b)}{a - (1 - a - b)} = \frac{g(b) - g(1 - a - b)}{b - (1 - a - b)}.$$

Hence the three points $\big(a, g(a)\big)$, $\big(b, g(b)\big)$, and $\big(1 - a - b, g(1 - a - b)\big)$ are collinear. Because $a, b$ can be chosen so that $1 - a - b$ ranges over an open interval, it follows that $g$ must be affine on $(0, 1)$, i.e. $g(s) = c + ds$. Therefore, $f(s) = g(s)/s = c/s + d$.

Finally, the normalization $B(\mathbf{1}/n) = 0$ is $f(1/n) = 0$, hence $0 = cn + d$ and $d = -cn$. Thus $f(s) = cn(\frac{1}{ns} - 1)$, which matches (40) up to an overall scale. $\qquad \square$

**Consequences for our choice.** Taking $x = y$ with $n = K$ gives $B_j(y) = \lambda(t)\big(\frac{1}{Ky_j} - 1\big)$, and taking $x = z$ with $n = M$ gives $C_m(z) = \mu(t)\big(\frac{1}{Mz_m} - 1\big)$. By Lemma C.6, these are exactly the (essentially unique, for $n \geq 3$) permutation-equivariant bonuses whose only effect is to rescale the collision fields $V(y)$ and $V(z)$. This is the structural reason later baseline-preserving constraints can be solved cleanly with a small number of scalars (e.g. $(\lambda, B_b, \mu)$), without injecting additional within-block geometry.

*Remark* C.9 (Implementation: clipping and re-centering). In finite-sample implementations one rarely uses a literal $1/x_i$ when $x_i$ can be very small. A standard safe variant is $1/(x_i + \varepsilon)$ or a clipped version $\min\{1/x_i, C\}$. This breaks exact centering and exact $V$-alignment, but the mean-centering can be restored by subtracting the empirical $x$-mean: $\widetilde{B}_i \leftarrow B_i - \sum_j x_j B_j$. Empirically this preserves the intended "rare-mode amplification" while preventing numerically unstable explosions near the boundary.

*Remark* C.10. In practice, the arm probabilities are estimated from finite rollouts via embedding and clustering, and are therefore bounded away from zero. If each prompt generates $G$ rollouts, then the empirical probabilities satisfy

$$y_j,\ z_m \in \left[\tfrac{1}{G}, 1\right].$$

Consequently, inverse-weight terms such as $1/y_j$ and $1/z_m$ are uniformly bounded by $G$ and cannot diverge. In typical training regimes, computational constraints limit $G$ to modest values (e.g., $G \in \{8, 16, 32, 64, 128\}$), ensuring that inverse-probability bonuses remain numerically stable in practice.

## D. A Minimal Bonus that Flips the Collision Drift

Recall that, in inner time, the baseline within-good dynamics follow the *collision* vector field $V(y) = y \odot y - \|y\|_2^2 \, y$, which amplifies large coordinates and tends to concentrate $y$ on a single good arm. Our goal is to design a bonus that can *cancel* this drift—or even *reverse* it—using a single scalar gain.

**Design desiderata.** We seek a bonus $B^\star$ added to the GRPO advantage with three properties: (i) it acts only on the good block (so it shapes $y$ directly), (ii) it is $y$-centered, $\sum_{j=1}^{K} y_j B_j^\star(y) = 0$ (so it does not create a common shift of the good logits), and (iii) it induces a drift on $y$ aligned with $-V(y)$, so that increasing the gain flips the sign of the collision flow.

**Construction (inverse-probability shaping).** A particularly simple choice meeting (i)–(iii) is the inverse-probability form $B_j^\star \propto 1/y_j$, which gives larger bonus to underrepresented good arms.

**Proposition D.1** (Anti-collision GRPO bonus). *Define, for some gain $\lambda(t) \geq 0$,*

$$B_j^\star(y) \ := \ \lambda(t) \left( \frac{1}{K \, y_j} - 1 \right), \qquad j = 1, \ldots, K, \qquad B_b^\star := 0. \tag{44}$$

*Then $\sum_{j=1}^{K} y_j B_j^\star(y) = 0$ and*

$$\mathbb{E}[\Delta y_j]_{\text{bonus}} \ = \ -\eta \, \lambda(t) \, (1 - p) \, y_j \left( y_j - \|y\|_2^2 \right), \tag{45}$$

$$\mathbb{E}[\Delta p]_{\text{bonus}} \ = \ +\eta \, \lambda(t) \, p \, (1 - p)^2 \left( \|y\|_2^2 - \tfrac{1}{K} \right). \tag{46}$$

*Consequently, with inner-time rescaling $d\tau/dt = \kappa\big(p(t)\big)$ and $\kappa$ as in (23),*

$$\frac{dy}{d\tau} = \left( 1 - \tilde{\lambda}(t) \right) \left( y \odot y - \|y\|_2^2 \, y \right), \qquad \tilde{\lambda}(t) := \frac{\sigma\big(p(t)\big)}{J} \frac{\lambda(t)}{p(t)}. \tag{47}$$

*Proof.* For the choice (44),

$$y_j B_j^\star(y) = \lambda(t) \left( \tfrac{1}{K} - y_j \right), \qquad \sum_{\ell=1}^{K} y_\ell^2 B_\ell^\star(y) = \lambda(t) \left( \tfrac{1}{K} - \|y\|_2^2 \right),$$

so

$$y_j B_j^\star(y) - \sum_{\ell=1}^{K} y_\ell^2 B_\ell^\star(y) = -\lambda(t) \left( y_j - \|y\|_2^2 \right).$$

Substituting this identity into (33) gives (45), and the same substitution in (34) yields (46).

For the inner-time form, recall that the baseline within-good drift in physical time has the form $\dot{y}_{\text{base}} = \kappa(p) \, V(y)$, while (45) corresponds to the bonus drift $\dot{y}_{\text{bonus}} = -\eta \, (1 - p) \lambda(t) \, V(y)$. Thus

$$\dot{y} = \Big[ \kappa(p) - \eta \, (1 - p) \lambda(t) \Big] V(y).$$

Dividing by $d\tau/dt = \kappa(p(t))$ and using $\kappa(p) = \eta \frac{J}{\sigma(p)} p(1 - p)$ gives (47). $\qquad \square$

**Interpretation (flip condition).** Notice that (47) shows that the bonus acts as a scalar controller on the collision field. In particular:

$$\tilde{\lambda}(t) = 1 \ \Rightarrow \ \text{the within-good collision drift is canceled,} \qquad \tilde{\lambda}(t) > 1 \ \Rightarrow \ \text{the drift flips and pushes } y \to \tfrac{1}{K} \mathbf{1}.$$

*Remark* D.2 (Practical stability). Because $B_j^\star \propto 1/y_j$, one may clip $y_j \leftarrow \max(y_j, \varepsilon)$ (or equivalently clip $B_j^\star$) to avoid numerical instabilities. With finite rollouts this is typically unnecessary in practice

**D.1. Full $(\dot{p}, \dot{y}, \dot{z})$ dynamics under the good-only anti-collision bonus**

We now combine Proposition D.1 with the baseline mean-field system to obtain the complete $(p, y, z)$ dynamics induced by the *good-only* anti-collision shaping. This serves two purposes: it confirms that the bonus only reshapes $y$ (and does not change the within-bad shape $z$), and it makes explicit the unintended interaction with the mass variable $p$ that motivates the neutralized constructions in Section E.

**Notation.** As before, let

$$V(y) = y \odot y - \|y\|_2^2 y, \qquad V(z) = z \odot z - \|z\|_2^2 z, \qquad L_y = \|y\|_2^2, \qquad L_z = \|z\|_2^2,$$

$$\kappa(p) = \eta \frac{J}{\sigma(p)} p(1 - p), \qquad \frac{d\tau}{dt} = \kappa(p(t)).$$

**Dynamics (baseline + good-only bonus).** Applying $B_j^\star(y) = \lambda(t)\left(\frac{1}{Ky_j} - 1\right)$ to the good block and $B_{b_m} \equiv 0$ to the bad block yields

$$\dot{y} = \left[\kappa(p) - \eta (1 - p)\lambda(t)\right] V(y),$$
$$\dot{z} = -\kappa(p) V(z), \tag{48}$$
$$\dot{p} = -\eta \frac{J}{\sigma(p)}[p(1 - p)]^2 (L_y + L_z) + \eta \lambda(t) p(1 - p)^2 \left(L_y - \frac{1}{K}\right).$$

**What the bonus does (and does not do).** *Within-good shaping.* The first line shows the intended effect: if $\eta (1 - p)\lambda(t) > \kappa(p)$, then the coefficient on $V(y)$ becomes negative and the within-good dynamics are *anti-collision*, pushing $y$ toward the uniform point.

*Within-bad shape.* The second line is unchanged from baseline: because the bonus is identical across all bad arms ($B_{b_m} \equiv 0$), it only induces a common shift of bad logits and therefore does not alter the within-bad shape $z$ to first order.

*Caveat: p-leak.* The third line shows the tradeoff: whenever $L_y > 1/K$ (i.e. the good block is non-uniform), the bonus contributes a *positive* drift to $p$, slowing down bad-mass decay and possibly increasing $p$ transiently in the midrun. A sufficient condition to keep $\dot{p} < 0$ in (48) is

$$\lambda(t)\left(L_y - \frac{1}{K}\right) < \frac{J}{\sigma(p)} p (L_y + L_z).$$

This is precisely why later sections add a bad-block correction: we want to preserve the desired shaping of $y$ while neutralizing the unintended $p$-drift.

**Special case $J = 1$ (perfect verifier).** Assume a perfect binary reward channel, i.e., $\delta_{\mathrm{FN}} = \delta_{\mathrm{FP}} = 0$, equivalently $J = 1 - \delta_{\mathrm{FN}} - \delta_{\mathrm{FP}} = 1$. Then $q(p) = \mathbb{E}[r] = 1 - p$, so

$$\sigma(p) = \sqrt{\mathrm{Var}(r)} = \sqrt{q(p)(1 - q(p))} = \sqrt{p(1 - p)}.$$

Recalling the definition $\kappa(p) := \eta \frac{J}{\sigma(p)} p(1 - p)$, this gives

$$\kappa(p) = \eta \frac{p(1 - p)}{\sigma(p)} = \eta \sqrt{p(1 - p)}, \qquad d\tau = \kappa(p) dt = \eta \sqrt{p(1 - p)} dt.$$

and the normalized gain becomes

$$\tilde{\lambda}(t) = \frac{\sigma(p(t))}{J} \frac{\lambda(t)}{p(t)} = \lambda(t) \sqrt{\frac{1 - p(t)}{p(t)}}.$$

In this case, (48) reduces to

$$\dot{y} = \eta \left[\sqrt{p(1 - p)} - (1 - p)\lambda(t)\right] V(y) = \eta (1 - p)\left(\sqrt{\frac{p}{1-p}} - \lambda(t)\right) V(y),$$
$$\dot{z} = -\eta \sqrt{p(1 - p)} V(z), \tag{49}$$
$$\dot{p} = -\eta [p(1 - p)]^{3/2} (L_y + L_z) + \eta \lambda(t) p(1 - p)^2 \left(L_y - \frac{1}{K}\right).$$

In inner time,

$$\frac{dy}{d\tau} = \left(1 - \tilde{\lambda}(t)\right) V(y) = \left(1 - \lambda(t)\sqrt{\tfrac{1-p}{p}}\right) V(y),$$

so the collision drift flips exactly when

$$\tilde{\lambda}(t) > 1 \quad \Longleftrightarrow \quad \lambda(t) > \sqrt{\tfrac{p(t)}{1-p(t)}}.$$

This dependence on $p(t)$ suggests normalizing $\lambda$ to keep the effective flip strength constant over training.

**Adaptive $\lambda(p)$ (constant flip strength in inner time).** Choose

$$\lambda(p) := \lambda_0 \sqrt{\frac{p}{1-p}}, \tag{50}$$

which makes $\tilde{\lambda}(t) \equiv \lambda_0$ for all $t$. Substituting (50) into (49) gives

$$\begin{aligned}
\dot{y} &= \eta \sqrt{p(1-p)} \left(1 - \lambda_0\right) V(y), \\
\dot{z} &= -\eta \sqrt{p(1-p)} \, V(z), \\
\dot{p} &= -\eta \left[p(1-p)\right]^{3/2} \left(L_z + (1-\lambda_0)L_y + \tfrac{\lambda_0}{K}\right).
\end{aligned} \tag{51}$$

Equivalently, in inner time,

$$\frac{dy}{d\tau} = \left(1 - \lambda_0\right) V(y),$$

so $\lambda_0$ is a single global knob: $\lambda_0 = 1$ cancels the collision drift, while $\lambda_0 > 1$ flips it into anti-collision.

**Asymptotic decay of bad mass.** Although (51) can exhibit transient $\dot{p} > 0$ when the good block is highly non-uniform, the bad-arm mass still vanishes asymptotically for $\lambda_0 > 1$.

**Proposition D.3** (Bad-mass elimination under adaptive gain). *In system* (51), *assume $\lambda_0 > 1$ and $0 \le p(0) < 1$ (so the good block is nonempty). Then $\lim_{t \to \infty} p(t) = 0$.*

*Proof.* The faces $p = 0$ and $p = 1$ are invariant, so it suffices to consider $0 < p(t) < 1$. Introduce inner time $\tau$ via

$$\frac{d\tau}{dt} = \eta \sqrt{p(1-p)}.$$

Dividing the $(y, z)$ equations in (51) by $d\tau/dt$ yields the decoupled inner-time dynamics

$$\frac{dy}{d\tau} = (1 - \lambda_0)V(y), \qquad \frac{dz}{d\tau} = -V(z).$$

For $\lambda_0 > 1$ these are anti-collision flows on the simplex interior, hence $y(\tau) \to \frac{1}{K}\mathbf{1}$ and $z(\tau) \to \frac{1}{M}\mathbf{1}$ as $\tau \to \infty$. In particular, $L_y(\tau) \to \frac{1}{K}$.

Next, rewrite the $p$-equation in inner time by dividing $\dot{p}$ in (51) by $d\tau/dt$:

$$\frac{dp}{d\tau} = -p(1-p)\left(L_z(\tau) + (1-\lambda_0)L_y(\tau) + \tfrac{\lambda_0}{K}\right).$$

Fix $\varepsilon := \frac{1}{2K(\lambda_0 - 1)}$. Since $L_y(\tau) \to \frac{1}{K}$, there exists $\tau_\star$ such that $L_y(\tau) \le \frac{1}{K} + \varepsilon$ for all $\tau \ge \tau_\star$. Using $L_z(\tau) \ge 0$, for $\tau \ge \tau_\star$ we obtain

$$L_z(\tau) + (1-\lambda_0)L_y(\tau) + \tfrac{\lambda_0}{K} \ge (1-\lambda_0)\left(\tfrac{1}{K} + \varepsilon\right) + \tfrac{\lambda_0}{K} = \tfrac{1}{K} - (\lambda_0 - 1)\varepsilon = \tfrac{1}{2K}.$$

Therefore, for $\tau \ge \tau_\star$,

$$\frac{dp}{d\tau} \le -\frac{1}{2K}p(1-p) \le -\frac{1}{2K}p,$$

so $p(\tau) \leq p(\tau_\star) \exp\left(-\frac{\tau - \tau_\star}{2K}\right) \to 0$ as $\tau \to \infty$.

Finally, since $p(\tau)$ decays exponentially, the physical time satisfies

$$t(\tau) - t(\tau_\star) = \int_{\tau_\star}^{\tau} \frac{ds}{\eta\sqrt{p(s)(1 - p(s))}} \geq \int_{\tau_\star}^{\tau} \frac{ds}{\eta\sqrt{p(s)}} = \infty \quad \text{as } \tau \to \infty,$$

hence $t(\tau) \to \infty$ and $\tau(t) \to \infty$ as $t \to \infty$. Thus $p(t) = p(\tau(t)) \to 0$. $\qquad \square$

# E. Flipping the collision drift while neutralizing bad mass

Our goal is to reshape the *within-good* distribution so that probability mass does not collapse onto a single good mode, without simultaneously accelerating or slowing the *inter-block* transfer of mass between good and bad. This separation is nontrivial because, under GRPO, any bonus enters the dynamics through the quadratic map $B \mapsto \mathbf{J}(\mathbf{p})^2 B$, which generically induces both (i) tangential drift inside each block (modifying $y$ and $z$) and (ii) a *leak* into the block-mass channel (modifying $p$). In this section we characterize the bonus-to-drift mapping in the $(K+M)$ model and derive neutrality conditions (and corresponding corrections) that keep $\dot{p}$ at its baseline value while flipping the sign of the within-good collision field.

## E.1. Bonus-induced drift of the total bad mass

We first compute the bonus-induced drift of the *total* bad mass

$$p := \sum_{m=1}^{M} p_{b_m},$$

under the $(K+M)$ block model when the bonus is *uniform within the bad block*:

$$B = \big(B_1, \dots, B_K, \underbrace{B_b, \dots, B_b}_{M \text{ times}}\big).$$

Let

$$s := (\underbrace{0, \dots, 0}_{K}, \underbrace{1, \dots, 1}_{M})$$

denote the indicator of the bad block. Then $p = s^\top \mathbf{p}$, hence $\Delta p = s^\top \Delta \mathbf{p}$. Under the bonus-only drift map $\mathbb{E}[\Delta \mathbf{p}]_{\text{bonus}} = \eta \, \mathbf{J}(\mathbf{p})^2 B$, we obtain

$$\mathbb{E}[\Delta p]_{\text{bonus}} = \eta \, s^\top \mathbf{J}(\mathbf{p})^2 B.$$

Since $\mathbf{J}(\mathbf{p})$ is symmetric, we may rewrite

$$s^\top \mathbf{J}(\mathbf{p})^2 B = (\mathbf{J}(\mathbf{p})s)^\top (\mathbf{J}(\mathbf{p})B).$$

Finally, we use the standard Jacobian action

$$\mathbf{J}(\mathbf{p})v = \mathbf{p} \odot \big(v - \bar{v}\,\mathbf{1}\big), \qquad \bar{v} := \mathbf{p}^\top v,$$

to obtain a closed-form expression in block coordinates, which makes the dependence on the within-block concentrations $L_y = \|y\|_2^2$ and $L_z = \|z\|_2^2$ explicit.

**Lemma E.1** (Bad-mass drift in the $(K+M)$ model). *Let $\mathbf{p} = ((1-p)y, pz)$ and $B = (B_1, \dots, B_K, B_b, \dots, B_b)$. Define the good-block moments*

$$m_1 := \sum_{j=1}^{K} y_j B_j, \qquad S := \sum_{j=1}^{K} y_j^2 B_j, \qquad L_y := \|y\|_2^2, \qquad L_z := \|z\|_2^2.$$

*Then, to first order in the step size $\eta$,*

$$\mathbb{E}[\Delta p]_{\text{bonus}} = \eta \, p(1-p)^2 \Big[ -S + m_1\big((1-p)L_y - pL_z\big) + p\,B_b\,(L_y + L_z) \Big]. \tag{52}$$

**Derivation sketch.** Let

$$\bar{B} := \langle \mathbf{p}, B \rangle = (1-p)m_1 + pB_b.$$

Using $\mathbf{J}(\mathbf{p})v = \mathbf{p} \odot \big(v - \langle \mathbf{p}, v \rangle \mathbf{1}\big)$, we have

$$\mathbf{J}(\mathbf{p})B = \mathbf{p} \odot (B - \bar{B}\mathbf{1}), \qquad \mathbf{J}(\mathbf{p})s = \mathbf{p} \odot \big(s - \langle \mathbf{p}, s \rangle \mathbf{1}\big) = \mathbf{p} \odot (s - p\mathbf{1}),$$

since $\langle \mathbf{p}, s \rangle = p$. By symmetry of $\mathbf{J}(\mathbf{p})$,

$$s^{\top}\mathbf{J}(\mathbf{p})^2 B = (\mathbf{J}(\mathbf{p})s)^{\top}(\mathbf{J}(\mathbf{p})B) = \sum_{i=1}^{K+M} p_i^2 (s_i - p)(B_i - \bar{B}).$$

This sum splits into a good-block part (where $s_i = 0$) and a bad-block part (where $s_i = 1$):

$$s^{\top}\mathbf{J}(\mathbf{p})^2 B = \underbrace{\sum_{j=1}^{K} p_j^2(-p)(B_j - \bar{B})}_{\text{good block}} + \underbrace{\sum_{m=1}^{M} p_{b_m}^2(1-p)(B_b - \bar{B})}_{\text{bad block}}.$$

Substituting $p_j = (1-p)y_j$ and $p_{b_m} = pz_m$, using $\sum_m z_m^2 = \|z\|_2^2$, and collecting terms in $\sum_j y_j B_j = m_1$ and $\sum_j y_j^2 B_j = S$ yields (52).

### E.2. A multi-bad-mass neutral bonus

We now choose $B_b$ so that the bonus does *not* perturb the total bad-mass dynamics, i.e. $\mathbb{E}[\Delta p]_{\text{bonus}} = 0$. Solving (52) for $B_b$ yields:

**Proposition E.2** (Multi-bad-mass neutralizer). *In the $(K+M)$-arm model with* $\mathbf{p} = ((1-p)y, pz)$ *and* $B = (B_1, \ldots, B_K, B_b, \ldots, B_b)$, *define* $m_1, S, L_y, L_z$ *as in Lemma E.1. The choice*

$$B_b^{\text{neu}}(y, z, p) := \frac{S - m_1\big((1-p)L_y - pL_z\big)}{p(L_y + L_z)} \tag{53}$$

*ensures*

$$\mathbb{E}[\Delta p]_{\text{bonus}} = 0 \qquad \Longrightarrow \qquad \dot{p} \text{ follows the baseline (no-bonus) ODE.}$$

**Centered simplification.** If the good-block bonus is exactly $y$-centered, i.e. $m_1 = \sum_j y_j B_j = 0$, then (53) simplifies to

$$B_b^{\text{neu}}(y, z, p) = \frac{S}{p(L_y + L_z)} = \frac{\sum_j y_j^2 B_j}{p(\|y\|_2^2 + \|z\|_2^2)}. \tag{54}$$

**Why $L_z$ matters (and why the aggregated neutralizer can under-compensate).** In the aggregated $(K+1)$-arm model we implicitly have $M = 1$ and hence $L_z = 1$. In practice, incorrect rollouts can be spread across many distinct bad modes, in which case $L_z < 1$. Since $L_y + L_z < L_y + 1$ when $L_z < 1$, the denominator in (54) is smaller, so the required neutralizer typically has *larger* magnitude than the aggregated formula (e.g. more negative when $S < 0$). Thus, the $(K+1)$ neutralizer can under-compensate when the bad block is diverse.

### E.3. Specialization to the anti-collision good bonus

Consider the anti-collision good-block bonus

$$B_j^{\star}(y) = \lambda(t)\Big(\frac{1}{Ky_j} - 1\Big), \qquad j = 1, \ldots, K,$$

which is $y$-centered for any $y \in \Delta^{K-1}$: $\sum_j y_j B_j^{\star}(y) = 0$. Moreover,

$$\sum_{j=1}^{K} y_j^2 B_j^{\star}(y) = \lambda(t)\Big(\frac{1}{K} - \|y\|_2^2\Big).$$

Plugging into (54) yields the refined neutralizer:

$$B_b^{\text{neu},\star}(y, z, p) = \lambda(t)\frac{\frac{1}{K} - \|y\|_2^2}{p\big(\|y\|_2^2 + \|z\|_2^2\big)}. \tag{55}$$

## E.4. Estimating $\|z\|_2^2$ from rollouts

To apply (55) in training, we need an estimate of the within-bad distribution $z$ (or at least its concentration $L_z = \|z\|_2^2$) for each prompt-wise group.

A direct empirical procedure is:

1. For a fixed prompt, separate rollouts into correct and incorrect sets.

2. Cluster the *correct* rollouts into $K$ semantic clusters; let $y_j$ be the resulting normalized cluster masses and compute $L_y = \sum_j y_j^2$.

3. Cluster the *incorrect* rollouts into $M$ semantic clusters; let $z_m$ be the normalized cluster masses and compute $L_z = \sum_m z_m^2$.

4. Compute $B_j^\star(y)$ for correct clusters and compute $B_b^{\mathrm{neu},\star}(y, z, p)$ via (55), with $p$ estimated as $\hat{p} = \#\text{incorrect}/\#\text{total}$.

5. Assign the per-sample bonus: correct rollouts in cluster $j$ receive $B_j^\star(y)$; all incorrect rollouts receive the same scalar $B_b^{\mathrm{neu},\star}(y, z, \hat{p})$.

### Remarks for stability.

- When there are very few incorrect samples in a group, the estimate of $z$ (hence $L_z$) is noisy; it is common to clip $L_z$ away from 0 and to clamp the magnitude of $B_b$.

- When $\hat{p}$ is extremely small, (55) scales like $1/\hat{p}$; in practice one typically disables the bad correction when $\hat{p}$ is below a threshold, or applies a smooth damping factor.

- One should *not* divide $B_b$ by the number of incorrect rollouts. The per-sample constant $B_b$ is the correct analogue of the single bad arm's bonus in the mean-field model; the averaging over samples already produces the $\hat{p}$ scaling.

**Interpretation via an effective number of bad modes.** Since $1/L_z$ is the effective number of bad clusters (analogous to $1/L_y$ for the good block), (55) shows that the neutralizer becomes stronger as the bad block becomes more diverse. Intuitively, when incorrect rollouts occupy many distinct modes, a uniform bad-block bonus has weaker leverage on the *total* bad mass $p$, so a larger magnitude is required to cancel the $p$-perturbation induced by the within-good diversity bonus.

## E.5. Final mean-field ODE under the multi-bad neutralized diversity bonus

We now combine the baseline GRPO mean-field dynamics (22) with the multi-bad anti-collision bonus design from Section. E.3. Let

$$V(y) := y \odot y - \|y\|_2^2\, y, \qquad V(z) := z \odot z - \|z\|_2^2\, z, \qquad L_y := \|y\|_2^2, \;\; L_z := \|z\|_2^2, \qquad \kappa(p) := \frac{J}{\sigma(p)}\, p(1-p).$$

We apply the good-block reciprocal (anti-collision) bonus

$$B_j^\star(y) = \lambda(t)\Big(\frac{1}{K y_j} - 1\Big), \qquad j = 1, \ldots, K,$$

together with a block-uniform bad correction (the same scalar on every bad arm $b_m$)

$$B_{b_m}^{\mathrm{neu}}(p, y, z) \equiv B_b^{\mathrm{neu}}(p, y, z), \qquad m = 1, \ldots, M,$$

chosen to neutralize the bonus-induced drift of the total bad mass $p$.

**Derivation sketch.** To derive the combined ODE to first order, we superpose two previously established drift identities. (1) For the good-only bonus $B^\star$ with $B_{b_m} \equiv 0$, Lemma C.3 applies because

$$y_j B_j^\star(y) = \lambda(t)\Big(\frac{1}{K} - y_j\Big), \qquad \sum_{j=1}^{K} y_j B_j^\star(y) = 0, \qquad \sum_{j=1}^{K} y_j^2 B_j^\star(y) = \lambda(t)\Big(\frac{1}{K} - L_y\Big),$$

yielding the bonus drifts

$$\dot{y}\big|_{\text{good-bonus}} = -(1-p)\lambda(t)\,V(y), \qquad \dot{p}\big|_{\text{good-bonus}} = \lambda(t)\,p(1-p)^2\Big(L_y - \frac{1}{K}\Big), \qquad \dot{z}\big|_{\text{good-bonus}} = 0.$$

(2) For the block-uniform bad correction we reuse the block-uniform $(c_g, c_b)$ formulas ((39)) with $c_g = 0$ and $c_b = B_b^{\text{neu}}(p, y, z)$:

$$\dot{y}\big|_{\text{bad-uniform}} = -p(1-p)\,B_b^{\text{neu}}(p, y, z)\,V(y), \qquad \dot{z}\big|_{\text{bad-uniform}} = p(1-p)\,B_b^{\text{neu}}(p, y, z)\,V(z),$$

$$\dot{p}\big|_{\text{bad-uniform}} = [p(1-p)]^2 (L_y + L_z)\,B_b^{\text{neu}}(p, y, z).$$

Imposing bad-mass neutrality $\dot{p}|_{\text{bonus}} = \dot{p}|_{\text{good-bonus}} + \dot{p}|_{\text{bad-uniform}} = 0$ gives

$$B_b^{\text{neu},\star}(p, y, z) = \lambda(t)\,\frac{\frac{1}{K} - L_y}{p(L_y + L_z)}.$$

Substituting $B_b^{\text{neu},\star}$ back into $\dot{y}|_{\text{bonus}}$ and $\dot{z}|_{\text{bonus}}$ yields the explicit prefactors in (56).

With $B_b^{\text{neu}} = B_b^{\text{neu},\star}$, the total bad-mass drift is unchanged by the bonus, while the within-block shape dynamics remain collinear with $V(y)$ and $V(z)$. Passing to the mean-field time scaling gives:

$$
\begin{aligned}
\dot{y} &= \underbrace{\kappa(p)\,V(y)}_{\text{baseline}} + \underbrace{\left[-(1-p)\frac{\lambda(t)\big(L_z + \frac{1}{K}\big)}{L_y + L_z}\right]V(y)}_{\text{bonus}}, \\[2mm]
\dot{z} &= \underbrace{-\kappa(p)\,V(z)}_{\text{baseline}} + \underbrace{\left[(1-p)\frac{\lambda(t)\big(\frac{1}{K} - L_y\big)}{L_y + L_z}\right]V(z)}_{\text{bonus}}, \\[2mm]
\dot{p} &= \underbrace{-\frac{J}{\sigma(p)}\,[p(1-p)]^2\big(L_y + L_z\big)}_{\text{baseline}} + \underbrace{0}_{\text{bonus (neutralized)}}.
\end{aligned}
\tag{56}
$$

In particular, since $L_y \geq 1/K$ and $\lambda(t) \geq 0$, we have $\frac{1}{K} - L_y \leq 0$, so the bonus term in $\dot{z}$ is non-positive and therefore *reinforces* the baseline anti-collision drift on the bad block.

### E.6. Adaptive diversity gains in the multi-bad neutralized scheme

With the neutralizer in place, $\dot{p}$ is *independent of* the diversity gain $\lambda(t)$ (cf. (56)). This cleanly separates roles:

- $p(t)$ continues to decay monotonically under the baseline ODE, and

- $\lambda(t)$ can be tuned purely to shape the *within-block* geometry (flatten $y$ via anti-collision, while typically further spreading $z$), without risking a sign flip in $\dot{p}$.

**Inner time and effective gains.** As in the aggregated analysis, introduce the "inner time" $\tau$ via

$$d\tau = \kappa(p(t))\,dt, \qquad \text{where} \quad \kappa(p) = \frac{J}{\sigma(p)}p(1-p).$$

Dividing the $y$ and $z$ equations in (56) by $\kappa(p)$ and using

$$\frac{1-p}{\kappa(p)} = \frac{\sigma(p)}{Jp},$$

we obtain

$$\frac{dy}{d\tau} = \left[1 - \underbrace{\left(\frac{\sigma(p)}{J}\frac{\lambda(t)}{p}\right)}_{=:\,\tilde{\lambda}(t)} \frac{L_z + \frac{1}{K}}{L_y + L_z}\right] V(y), \tag{57}$$

$$\frac{dz}{d\tau} = -\left[1 + \tilde{\lambda}(t)\frac{L_y - \frac{1}{K}}{L_y + L_z}\right] V(z), \tag{58}$$

where the inner-time gain is

$$\tilde{\lambda}(t) := \frac{\sigma(p(t))}{J}\frac{\lambda(t)}{p(t)}. \tag{59}$$

The key new feature is the attenuation factor

$$\rho(y,z) := \frac{L_z + \frac{1}{K}}{L_y + L_z} \in (0,1],$$

which reduces the *effective* strength of the within-good regularizer. The attenuation is strongest when the good block is collapsed ($L_y \approx 1$) and the bad block is very diffuse ($L_z \approx 1/M$).

**Condition for flipping the within-good drift.** In inner time, the baseline $y$-dynamics have coefficient $+1$ in front of $V(y)$ and hence are *collision*. The bonus rescales this coefficient. From (57),

$$\text{collision cancelled} \iff \tilde{\lambda}(t)\,\rho(y,z) = 1, \qquad \text{collision flipped (anti-collision)} \iff \tilde{\lambda}(t)\,\rho(y,z) > 1. \tag{60}$$

Equivalently, the state-dependent threshold is

$$\tilde{\lambda}(t) > \tilde{\lambda}_{\text{flip}}(y,z) := \frac{L_y + L_z}{L_z + \frac{1}{K}}. \tag{61}$$

Notably, $\tilde{\lambda}_{\text{flip}}$ increases as $L_z$ decreases: when incorrect rollouts occupy many distinct bad modes (diffuse $z$), the neutralizer must work harder, and the net anti-collision effect on $y$ becomes weaker unless $\lambda(t)$ is increased accordingly.

**Why adapt $\lambda(t)$? (neutralizer scaling).** The per-bad-arm neutralizer is

$$B_b^{\text{neu},\star}(p,y,z) = \lambda(t)\frac{\frac{1}{K} - L_y}{p(L_y + L_z)}.$$

If $\lambda(t)$ is held constant, then $B_b^{\text{neu},\star}$ scales like $1/p$ as $p \to 0$, producing very large per-sample bonuses late in training. A natural goal is therefore to choose $\lambda(t)$ so that:

- the *inner-time strength* is stable as $p(t)$ changes, and

- the neutralizer does not blow up as $1/p$ when $p$ becomes small.

Note that the apparent $1/p$ blow-up is not inevitable: as the good block becomes uniform, $L_y \to 1/K$ and hence $\frac{1}{K} - L_y \to 0$, which can partially (or fully) mitigate the growth when $p \to 0$.

**A $p$-adaptive schedule with constant $\tilde{\lambda}$.** To choose $\lambda(t)$ so that $\tilde{\lambda}(t)$ in (59) is constant:

$$\lambda(t) = \lambda_0 \frac{J\,p(t)}{\sigma(p(t))}, \qquad \lambda_0 > 0. \tag{62}$$

Then $\tilde{\lambda}(t) \equiv \lambda_0$ and the inner-time equations become

$$\frac{dy}{d\tau} = \left[1 - \lambda_0 \frac{L_z + \frac{1}{K}}{L_y + L_z}\right] V(y), \tag{63}$$

$$\frac{dz}{d\tau} = -\left[1 + \lambda_0 \frac{L_y - \frac{1}{K}}{L_y + L_z}\right] V(z), \tag{64}$$

and the bad neutralizer magnitude improves from $O(1/p)$ to $O(1/\sigma(p))$:

$$B_b^{\mathrm{neu},\star}(p, y, z) = \lambda_0 \frac{J}{\sigma(p)} \frac{\frac{1}{K} - L_y}{L_y + L_z}. \tag{65}$$

*When does this guarantee anti-collision on $y$?* By (61), flipping $y$ at the current state requires

$$\lambda_0 > \frac{L_y + L_z}{L_z + \frac{1}{K}}.$$

A conservative *global* sufficient condition (uniform over all $y \in \Delta^{K-1}$ and $z \in \Delta^{M-1}$) is obtained by taking the worst case $L_y = 1$ and $L_z = 1/M$:

$$\lambda_0 > \frac{1 + \frac{1}{M}}{\frac{1}{M} + \frac{1}{K}} = \frac{K(M+1)}{K+M}. \tag{66}$$

This can be large when $M$ is large (highly multi-modal errors), motivating a second schedule that normalizes away the attenuation factor.

**A $(p, y, z)$-adaptive schedule with constant within-good strength.** To keep the within-good dynamics at a constant coefficient in inner time, we can adapt $\lambda$ to cancel the attenuation:

$$\lambda(t) = \lambda_0 \frac{J\,p(t)}{\sigma(p(t))} \frac{L_y(t) + L_z(t)}{L_z(t) + \frac{1}{K}}, \qquad \lambda_0 > 0. \tag{67}$$

Substituting into (57) gives the decoupled inner-time good-block dynamics

$$\frac{dy}{d\tau} = (1 - \lambda_0)\, V(y), \tag{68}$$

so $\lambda_0 = 1$ cancels collision and any $\lambda_0 > 1$ flips the drift to anti-collision. Meanwhile, the bad-block dynamics remain anti-collision and are strengthened:

$$\frac{dz}{d\tau} = -\left[1 + \lambda_0 \frac{L_y - \frac{1}{K}}{L_z + \frac{1}{K}}\right] V(z), \tag{69}$$

since $L_y - \frac{1}{K} \geq 0$. Finally, the neutralizer simplifies to

$$B_b^{\mathrm{neu},\star}(p, y, z) = \lambda_0 \frac{J}{\sigma(p)} \frac{\frac{1}{K} - L_y}{L_z + \frac{1}{K}}, \tag{70}$$

removing both the explicit $1/p$ factor and the $(L_y + L_z)$ attenuation.

**Corollary E.3** (Constant-strength diversity shaping under multi-bad neutrality). *Under the neutralized bonus design and the adaptive gain* (67):

1. *the total bad mass still follows the baseline ODE and decays monotonically:* $\dot{p} = -(J/\sigma(p))[p(1-p)]^2(L_y + L_z) < 0$;

2. *in inner time, the within-good drift is exactly $(1 - \lambda_0)V(y)$, so any $\lambda_0 > 1$ flips the collision drift and drives $y$ toward the uniform point $\frac{1}{K}\mathbf{1}$;*

3. *the within-bad drift remains anti-collision and is amplified relative to baseline, cf.* (69).

**Specialization: the noise-free $J=1$ case.** When $J=1$ and $\sigma(p) = \sqrt{p(1-p)}$, the schedules simplify to

$$\lambda(t) = \begin{cases} \lambda_0 \sqrt{\dfrac{p(t)}{1 - p(t)}}, & p\text{-adaptive choice (62)}, \\[2em] \lambda_0 \sqrt{\dfrac{p(t)}{1 - p(t)}} \dfrac{L_y(t) + L_z(t)}{L_z(t) + \frac{1}{K}}, & (p, y, z)\text{-adaptive choice (67)}. \end{cases}$$

**Practical remarks (stability and estimation).**

- The schedules above require estimates of $p$, $L_y$, and $L_z$; these are available from the same clustering statistics used in Section. E.4.

- Even with $p$-adaptation, $B_b^{\mathrm{neu},\star}$ scales like $1/\sigma(p)$ and can become large when $p$ is very small; in practice one typically clips $|B_b|$, clips $\lambda(t)$, and/or smoothly turns off the correction below a minimum bad-mass threshold.

- When $L_z$ is noisy (few incorrect samples), the ratio $(L_y + L_z)/(L_z + 1/K)$ in (67) should be smoothed or clipped to avoid bursty gains.

**Quick reference (recommended schedules).**

> **QUICK REFERENCE: TWO PRACTICAL $\lambda(t)$ SCHEDULES**
>
> - **(A) $p$-adaptive (constant inner gain $\tilde{\lambda} \equiv \lambda_0$).** Set
>
> $$\lambda(t) = \lambda_0 \frac{J \, p(t)}{\sigma(p(t))}.$$
>
> Then $\tilde{\lambda}(t) = \frac{\sigma(p)}{J} \frac{\lambda(t)}{p(t)} \equiv \lambda_0$ and (63)–(64) apply. The per-bad-sample neutralizer scales as
>
> $$B_b^{\mathrm{neu},\star}(p, y, z) = \lambda_0 \frac{J}{\sigma(p)} \frac{\frac{1}{K} - L_y}{L_y + L_z}$$
>
> (cf. (65)).
>
> - **(B) $(p, y, z)$-adaptive (constant within-good strength).** Set
>
> $$\lambda(t) = \lambda_0 \frac{J \, p(t)}{\sigma(p(t))} \frac{L_y(t) + L_z(t)}{L_z(t) + \frac{1}{K}}.$$
>
> Then the within-good inner-time dynamics collapse to
>
> $$\frac{dy}{d\tau} = (1 - \lambda_0) \, V(y)$$
>
> (cf. (68)), and the neutralizer simplifies to
>
> $$B_b^{\mathrm{neu},\star}(p, y, z) = \lambda_0 \frac{J}{\sigma(p)} \frac{\frac{1}{K} - L_y}{L_z + \frac{1}{K}}$$
>
> (cf. (70)).

*Remark* E.4 (Large $\lambda_0$ can over-flatten the bad block under a uniform-bad neutralizer). Under the multi-bad neutralized scheme, increasing the within-good gain $\lambda_0$ also increases the magnitude of the *uniform* bad-block correction. This tends to push $z$ toward the center (uniform $z$) faster, decreasing $L_z$ toward $1/M$. Since the baseline bad-mass decay rate scales with $(L_y + L_z)$, this can *indirectly slow down* the mid-training decay of $p$ even while $y$ becomes diverse.

## F. Microstate Probabilities vs. Macrostate Masses

In our bandit abstraction, each arm (or *reasoning mode*) is a *macrostate*: a cluster of full sequences $C_j \subset \mathcal{H}$ that share some semantic structure (e.g., the same final answer, the same proof template, or the same style of reasoning). The policy $\pi$ is a distribution on *microstates* $h \in \mathcal{H}$, while the bandit model tracks only *coarse-grained* masses obtained by summing $\pi$ over clusters.

**Good/bad partition and conditional masses.** Let $V(h) \in \{0, 1\}$ be the verifier and define the good event $G := \{V(h) = 1\}$ and bad event $G^c$. We partition the good set into $K$ clusters $\{C_j^+\}_{j=1}^K$ and the bad set into $M$ clusters $\{C_m^-\}_{m=1}^M$ (as in the $(K+M)$-block model). The policy induces *unconditional* macrostate masses

$$\pi(h) = \Pr(\text{sequence } h), \qquad q_j^+ = \Pr(h \in C_j^+) = \sum_{h \in C_j^+} \pi(h), \qquad q_m^- = \Pr(h \in C_m^-) = \sum_{h \in C_m^-} \pi(h).$$

The total bad mass and good mass are

$$p := \Pr(G^c) = \sum_{m=1}^M q_m^-, \qquad 1 - p = \Pr(G) = \sum_{j=1}^K q_j^+.$$

Finally, the within-block (conditional) distributions are

$$y_j := \Pr(C_j^+ \mid G) = \frac{q_j^+}{1-p}, \qquad z_m := \Pr(C_m^- \mid G^c) = \frac{q_m^-}{p},$$

so $\sum_{j=1}^K y_j = 1$ and $\sum_{m=1}^M z_m = 1$. (If a cluster were allowed to mix good and bad sequences, then the numerator would be $\Pr(C_j \cap G)$ rather than $\Pr(C_j)$; here we avoid this ambiguity by clustering $G$ and $G^c$ separately.)

**Microstates vs. macrostates.** A key conceptual point is that $\pi(h)$ lives at the *microstate* level, whereas $q_j^\pm$, $y$, and $z$ live at the *macrostate* level. The mapping $\pi \mapsto \{q_j^\pm\}$ is a coarse-graining: it aggregates exponentially many sequences inside each cluster and therefore discards essentially all micro-level detail. Consequently, observing a few microstate probabilities conveys little about macrostate masses. For example, knowing that two sampled sequences satisfy

$$\pi(h_1) \approx 10^{-40}, \qquad \pi(h_2) \approx 10^{-20},$$

tells us that $h_2$ is more likely than $h_1$ *as an individual sequence*, but it does *not* determine which *arm* has larger total probability. The arm containing $h_1$ might aggregate a huge number of similarly rare sequences and end up with $q_{\mathrm{arm}(h_1)} \gg q_{\mathrm{arm}(h_2)}$, or vice versa.

**Why sequence probabilities look "astronomically small."** A second (common) source of confusion is the scale of sequence-level probabilities. For a length-$T$ sequence $h = (x_1, \ldots, x_T)$,

$$\pi(h) = \prod_{t=1}^T \pi(x_t \mid x_{<t}).$$

In practice we include the EOS/stop token in this product; the exact stopping rule is immaterial for the point below. Even if per-token probabilities are "reasonable" (say 0.05–0.5), multiplying over $T \approx 50$–$200$ tokens typically yields extremely small values (often spanning many tens of orders of magnitude). Thus, a gap of 20 orders of magnitude between two sequences,

$$\frac{\pi(h_2)}{\pi(h_1)} \approx 10^{20},$$

sounds dramatic, but spread over $T$ tokens it corresponds to only

$$\frac{1}{T}\Big(\log_{10} \pi(h_2) - \log_{10} \pi(h_1)\Big) = \mathcal{O}\Big(\frac{20}{T}\Big)$$

*Table 12.* Empirical distribution of $\log_{10} \pi(h)$ for $N = 20{,}000$ sequences in the toy model ($T = 80$).

|  | $\min \log_{10} \pi(h)$ | $\max \log_{10} \pi(h)$ | $\mathbb{E}[\log_{10} \pi(h)]$ | $\mathrm{Std}[\log_{10} \pi(h)]$ |
|---|---|---|---|---|
| Empirical | $-44.06$ | $-10.76$ | $-25.81$ | $4.16$ |

*Table 13.* Distribution of within-batch ranges of $\log_{10} \pi(h)$ for 500 batches of size $G = 10$.

|  | min range | max range | mean range | std. range |
|---|---|---|---|---|
| Empirical | 4.12 | 22.20 | 12.70 | 3.25 |

orders of magnitude *per token*, i.e. an average per-token factor of about $10^{20/T}$ (roughly $1.3\times$–$2.5\times$ when $T \in [50, 200]$). In a high-entropy model, variation of this magnitude is entirely normal from one rollout to another, even in small groups (e.g., $G = 10$ samples).

To make this concrete, we now show a simple synthetic experiment where a toy "LLM" already exhibits a broad distribution over sequence probabilities and large within-batch gaps, mirroring what we observe in practice.

**Toy product model.** Consider a vocabulary $\{A, B, C, D, E\}$ with fixed token probabilities

$$P(A) = 0.8, \quad P(B) = 0.1, \quad P(C) = 0.05, \quad P(D) = 0.03, \quad P(E) = 0.02,$$

and sequences of length $T = 80$. Tokens are sampled i.i.d. across positions, so for a sequence $h = (x_1, \ldots, x_{80})$ we have

$$\pi(h) = \prod_{t=1}^{80} P(x_t), \qquad \log_{10} \pi(h) = \sum_{t=1}^{80} \log_{10} P(x_t).$$

Since $\log_{10} \pi(h)$ is a sum of i.i.d. terms, it concentrates around $T\,\mathbb{E}[\log_{10} P(X)]$ with standard deviation $\sqrt{T \mathrm{Var}(\log_{10} P(X))}$, so even this simple product model produces a broad (approximately Gaussian) spread in sequence log-probabilities.

We sampled $N = 20{,}000$ sequences from this toy model and recorded $\log_{10} \pi(h)$ for each. Table 12 summarizes the empirical distribution of sequence log-probabilities.

Even in this extremely simple product model, typical log-probabilities cluster around $-25.8$, while the sampled support spans $> 30$ orders of magnitude. This already suggests that a handful of samples can easily land at very different points along the log-probability axis.

**Within-batch variability.** To mimic "$G$ rollouts for a fixed prompt," we repeatedly drew batches of size $G = 10$ and, for each batch $b$, computed the range

$$\mathrm{range}_b = \max_{1 \leq i \leq G} \log_{10} \pi\big(h^{(i)}\big) - \min_{1 \leq i \leq G} \log_{10} \pi\big(h^{(i)}\big).$$

Table 13 shows summary statistics over 500 such batches.

A typical batch of only 10 samples already spans $\approx$ 10–15 orders of magnitude in sequence probability, and some batches exhibit gaps exceeding $10^{22}$ between their most and least likely sequences.

**One concrete batch.** To illustrate the effect more explicitly, Table 14 shows one representative batch of $G = 10$ sequences. Instead of listing all 80 tokens, we record the counts $(n_A, n_B, n_C, n_D, n_E)$ for each sequence $h^{(i)}$, along with its log-probability and probability. Sequences are sorted from most to least probable.

Within this single batch, the most likely sequence (row 1) has probability

$$\pi(h^{(1)}) \approx 2.1 \times 10^{-20},$$

while the least likely sequence (row 10) has

$$\pi(h^{(10)}) \approx 2.2 \times 10^{-31},$$

*Table 14.* Example batch of $G = 10$ sequences from the toy model, sorted by probability. For each $h^{(i)}$ we show token counts and the resulting sequence probability.

| $i$ | $n_A$ | $n_B$ | $n_C$ | $n_D$ | $n_E$ | $\log_{10} \pi(h^{(i)})$ | $\pi(h^{(i)})$ |
|---|---|---|---|---|---|---|---|
| 1 | 69 | 7 | 2 | 0 | 2 | $-19.69$ | $2.06 \times 10^{-20}$ |
| 2 | 68 | 7 | 4 | 1 | 0 | $-20.32$ | $4.82 \times 10^{-21}$ |
| 3 | 67 | 7 | 2 | 1 | 3 | $-22.71$ | $1.93 \times 10^{-23}$ |
| 4 | 66 | 7 | 4 | 2 | 1 | $-23.34$ | $4.52 \times 10^{-24}$ |
| 5 | 65 | 5 | 5 | 3 | 2 | $-25.77$ | $1.69 \times 10^{-26}$ |
| 6 | 62 | 10 | 4 | 3 | 1 | $-27.48$ | $3.31 \times 10^{-28}$ |
| 7 | 60 | 11 | 5 | 2 | 2 | $-29.76$ | $1.72 \times 10^{-30}$ |
| 8 | 60 | 9 | 6 | 4 | 1 | $-30.41$ | $3.88 \times 10^{-31}$ |
| 9 | 60 | 7 | 10 | 2 | 1 | $-30.57$ | $2.69 \times 10^{-31}$ |
| 10 | 60 | 8 | 8 | 2 | 2 | $-30.67$ | $2.16 \times 10^{-31}$ |

a gap of roughly 10.98 orders of magnitude:

$$\frac{\pi(h^{(1)})}{\pi(h^{(10)})} \approx 10^{10.98}.$$

Yet the underlying per-token difference is mild: dividing by $T = 80$,

$$\frac{1}{T} \left( \log_{10} \pi(h^{(1)}) - \log_{10} \pi(h^{(10)}) \right) \approx 0.14,$$

which corresponds to only a factor of $10^{0.14} \approx 1.38$ in average token probability. Intuitively, $h^{(1)}$ simply chose slightly more "typical" tokens (more $A$'s, fewer rare tokens) than $h^{(10)}$; this small per-step advantage compounds over 80 positions into an $\approx 10^{11}$ gap at the sequence level.

**Implications.** This toy example explains why, in real LLM rollouts, it is perfectly plausible to observe within a single group:

- one sequence with $\pi(h) \approx 10^{-40}$,

- another with $\pi(h) \approx 10^{-20}$,

and why that observation does *not* imply that the $10^{-40}$ sequence came from a "vanishingly small" region of the policy. In high-dimensional sequence spaces, absolute sequence probabilities are generically tiny, and modest token-level differences translate into huge multiplicative gaps. Consequently:

1. Large differences in $\pi(h)$ across a handful of samples are expected and consistent with a broad, smooth distribution over log-probabilities.

2. These microstate-level differences provide only limited information about macrostate masses $q_j^{\pm}$ (arms / reasoning modes), which depend on aggregating over many unseen sequences.

3. In our bandit analysis, $y$ should be understood as a latent property of the policy—"how the good mass is distributed across reasoning modes"—and must be estimated via coarse-grained statistics (e.g., clustering and frequencies), not inferred from raw $\pi(h)$ values alone.

This perspective clarifies why sequence-level log-probabilities, though crucial for gradient updates, cannot by themselves resolve the macro-level structure of good arms in the bandit model.

## G. Reasoning-Mode Clusters for Sampled Model Outputs

To make our mode-based diversity metrics concrete, we present a qualitative case study on a single prompt (AIME 2025 #6). For each checkpoint (base, GRPO, and G$^2$RPO), we draw 32 independent rollouts, embed and cluster the resulting solutions into *reasoning modes*, and then inspect representative outputs. We report cluster frequencies and include abridged solution prints for each mode to show that the clusters correspond to genuinely distinct proof strategies (not superficial paraphrases), providing a sanity check that our clustering-and the improvements under G$^2$RPO-are not a self-fulfilling artifact of the analysis pipeline.

---

**Math Reasoning Sample: Tangential Isosceles Trapezoid**

SOURCE: AIME 2025 #6     MODEL: `deepseek-ai/DeepSeek-R1-Distill-Qwen-7B`

SAMPLES: 32 rollouts

**Problem.** An isosceles trapezoid has an incircle tangent to all four sides. The incircle has radius 3 and the trapezoid's area is 72. Let the bases have lengths $r \neq s$. Find $r^2 + s^2$.

**Clustering goal.** We cluster rollouts into *reasoning modes* based on the proof strategy (core lemmas and derivation path), rather than surface phrasing. Each mode is shown using a distinct color; within a mode, multiple rollouts can be near-duplicates (same reasoning with minor stylistic variation).

---

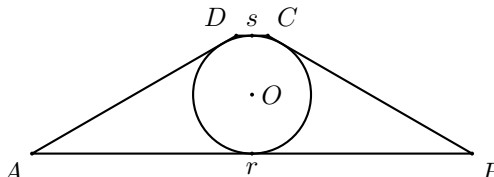

*Figure 7.* **Example prompt for mode clustering (AIME 2025 #6).** Geometry diagram of the tangential isosceles trapezoid used in Appendix F; the incircle (radius 3) and bases (lengths $r, s$) define a multi-solution setting with several distinct valid reasoning strategies.

| Mode | Proof strategy (signature) | Count | Prob. |
|---|---|---|---|
| ■ A | Height+Area $\to r+s$; Pitot; Pythagoras (altitude drop) | 19/32 | $\approx 66\%$ |
| ■ B | $A = \rho \cdot s$ (semiperimeter); equal tangents bookkeeping | 6/32 | $\approx 16\%$ |
| ■ C | Cyclic + Brahmagupta (cyclic quadrilateral area identity) | 4/32 | $\approx 9\%$ |
| ■ D | Coordinate tangency (circle + tangent lines) | 2/32 | $\approx 6\%$ |
| ■ E | Trig parameterization ($\tan\theta, \cot\theta$) | 1/32 | $\approx 3\%$ |

*Table 15.* Reasoning-mode clusters and *estimated* frequencies among 32 sampled rollouts.

### G.1. Mode A (blue): Height + Area $\to r + s$; Pitot; Pythagoras

---

**Mode A (blue): canonical synthetic derivation**                                      **(est. 19/32)**

CLUSTER: Mode A     FINAL: 504

**Model output (abridged; ... denotes omitted details).** Let $\rho = 3$ be the inradius and $r \neq s$ the bases. Since the incircle is tangent to both bases, $h = 2\rho = 6$.

$$72 = \frac{(r+s)h}{2} = 3(r+s) \implies r+s = 24.$$

Tangential quadrilateral property ($\dots$) gives $r + s = $ (sum of legs) $= 2\ell$, hence $\ell = 12$. Dropping perpendiculars ($\dots$),

$$\ell^2 = \left(\frac{s-r}{2}\right)^2 + h^2 \implies 12^2 = \left(\frac{s-r}{2}\right)^2 + 6^2 \implies s - r = 12\sqrt{3}.$$

---

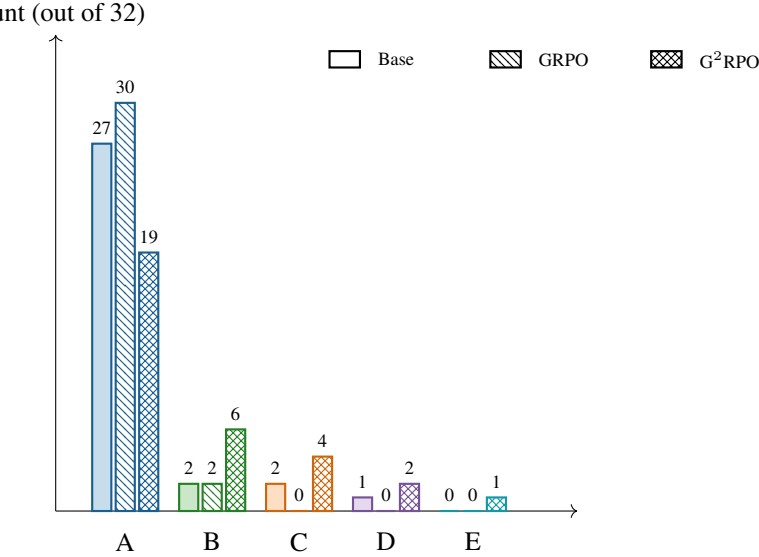

*Figure 8.* **Empirical reasoning-mode counts across checkpoints (one prompt).** For the fixed prompt in Fig. 7, we sample 32 rollouts from the base model, GRPO, and G²RPO, cluster solutions into modes (A–E), and plot per-mode counts in a single grouped histogram.

---

Thus
$$r = 12 - 6\sqrt{3}, \qquad s = 12 + 6\sqrt{3}, \qquad r^2 + s^2 = 504.$$

---

**Mode A (blue): compute $(r - s)^2$ then $rs$** (est. 19/32)

CLUSTER: Mode A     FINAL: 504

**Model output (abridged; ... denotes omitted details).** The incircle has radius 3, so the height is $h = 6$. From area,

$$72 = \frac{1}{2}(r + s)h = \frac{1}{2}(r + s) \cdot 6 \Rightarrow r + s = 24.$$

Tangency gives $r + s = 2\ell$ (...), hence $\ell = 12$. Dropping an altitude (...) yields

$$\left(\frac{r - s}{2}\right)^2 + 6^2 = 12^2 \Rightarrow (r - s)^2 = 432.$$

Now

$$(r + s)^2 - (r - s)^2 = 4rs \Rightarrow 576 - 432 = 4rs \Rightarrow rs = 36,$$

so

$$r^2 + s^2 = (r + s)^2 - 2rs = 24^2 - 72 = 504.$$

---

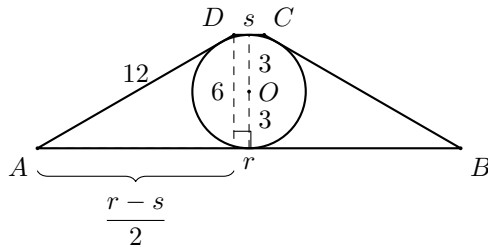

*Figure 9.* Mode A sketch: dropping an altitude yields a right triangle with legs 6 and $\frac{r-s}{2}$, hypotenuse 12.

## G.2. Mode B (green): $A = \rho \cdot$ (**semiperimeter**) **and tangency bookkeeping**

---

**Mode B (green): semiperimeter first** (est. 6/32)

CLUSTER: Mode B    FINAL: 504

**Model output (abridged; ... denotes omitted details).** Let the bases be $r \neq s$ and the equal legs be $\ell$. Since the trapezoid is tangential,

$$\text{Area} = \rho \cdot (\text{semiperimeter}).$$

With $\rho = 3$ and area 72,

$$72 = 3 \cdot \text{sp} \;\Rightarrow\; \text{sp} = 24 \;\Rightarrow\; r + s + 2\ell = 48.$$

Tangential trapezoid property gives $r + s = 2\ell$ (...), hence

$$2(r + s) = 48 \Rightarrow r + s = 24, \qquad \ell = 12.$$

Also $h = 2\rho = 6$. Dropping perpendiculars (...) gives $s - r = 12\sqrt{3}$, so $r^2 + s^2 = 504$.

---

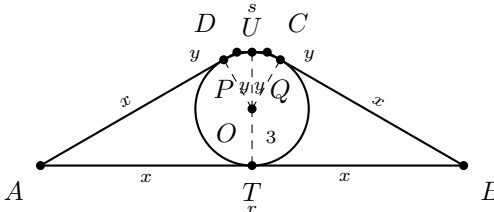

*Figure 10.* Mode B sketch: tangency implies equal tangent segments from each vertex (e.g., $AT = AP$), a common bookkeeping route.

## G.3. Mode C (orange): Cyclic trapezoid + Brahmagupta

---

**Mode C (orange): Brahmagupta shortcut** (est. 4/32)

CLUSTER: Mode C    FINAL: 504

**Model output (abridged; ... denotes omitted details).** Let the bases be $r \neq s$ and the equal legs be $\ell$. Since the trapezoid is tangential,

$$A = \rho\, p \Rightarrow 72 = 3p \Rightarrow p = 24,$$

so the perimeter is $2p = 48$. Tangency implies $r + s = 2\ell$, hence $\ell = 12$ and $r + s = 24$. An isosceles trapezoid is cyclic (...), so apply Brahmagupta with sides $(r, s, 12, 12)$:

$$72 = \sqrt{(24 - r)(24 - s)(24 - 12)(24 - 12)} = 12\sqrt{(24 - r)(24 - s)}.$$

Thus $(24 - r)(24 - s) = 36$. Using $r + s = 24$ gives $rs = 36$, so

$$r^2 + s^2 = (r + s)^2 - 2rs = 24^2 - 72 = 504.$$

---

## G.4. Mode D (purple): Coordinate tangency (circle at origin)

---

**Mode D (purple): analytic geometry with tangents** (est. 2/32)

CLUSTER: Mode D    FINAL: 504

**Model output (abridged; ... denotes omitted details).** Let the incircle have radius $\rho = 3$. Then $h = 2\rho = 6$ and

$$72 = \tfrac{1}{2}(r + s)h = \tfrac{1}{2}(r + s) \cdot 6 \Rightarrow r + s = 24.$$

---

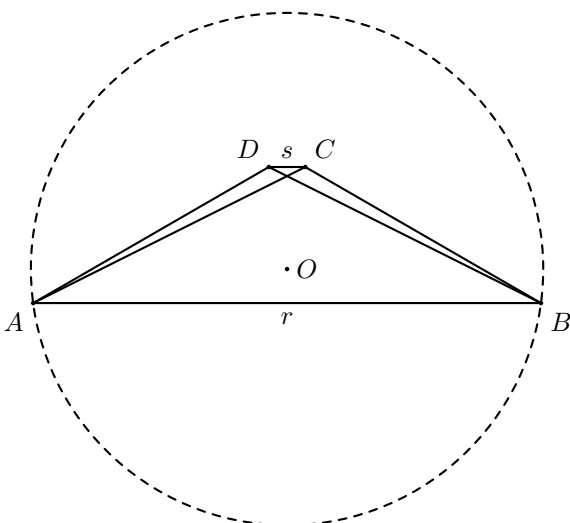

*Figure 11.* Mode C sketch: an isosceles trapezoid is cyclic, enabling Brahmagupta on the cyclic quadrilateral with sides $(r, s, 12, 12)$ and $p = 24$.

Place the circle at the origin: $x^2 + y^2 = 9$. The bases are horizontal tangents $y = \pm 3$. Let the legs be tangents $y = mx + b$ and $y = -mx + b$. Tangency gives

$$\frac{b}{\sqrt{1 + m^2}} = 3 \Rightarrow b = 3\sqrt{1 + m^2} \quad (b > 0).$$

Intersecting with $y = \pm 3$ yields base lengths (. . . )

$$s = \frac{2(b - 3)}{m}, \qquad r = \frac{2(b + 3)}{m}.$$

Hence

$$r + s = \frac{4b}{m} = 24 \Rightarrow b = 6m.$$

Combine with $b = 3\sqrt{1 + m^2}$:

$$6m = 3\sqrt{1 + m^2} \Rightarrow m = \frac{1}{\sqrt{3}}.$$

Thus $r, s = 12 \pm 6\sqrt{3}$ and $r^2 + s^2 = 504$.

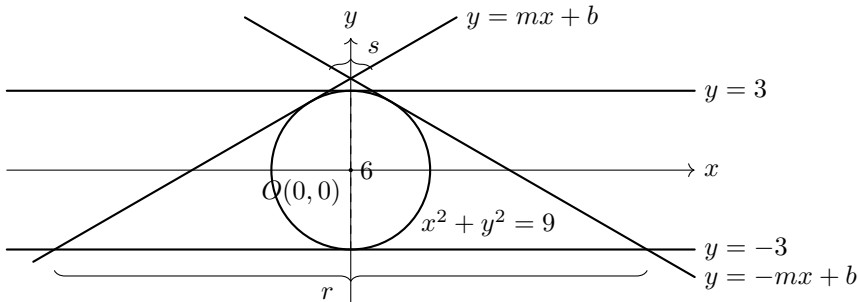

*Figure 12.* Mode D sketch: set the incircle as $x^2 + y^2 = 9$ and represent sides as tangent lines.

### G.5. Mode E (teal): Trigonometric parameterization

---

**Mode E (teal): trig route** (est. 1/32)

CLUSTER: Mode E

**Abridged solution (... = omitted details).** Tangency implies the sum of the bases equals the sum of the legs. Since the trapezoid is isosceles,

$$r + s = 2\ell.$$

With inradius 3, the height is $h = 2 \cdot 3 = 6$, and the area condition gives

$$72 = \tfrac{1}{2}(r+s)h = \tfrac{1}{2}(r+s)\cdot 6 \ \Rightarrow\ r+s = 24 \ \Rightarrow\ \ell = 12.$$

Using the standard tangent-length/trig parametrization for a tangential isosceles trapezoid (...),

$$r = 6\cot\left(\frac{\theta}{2}\right), \qquad s = 6\tan\left(\frac{\theta}{2}\right).$$

Moreover $r + s = 24$ forces $\theta = 30°$ (...), so $u = \theta/2 = 15°$. Using

$$\cot 15° = 2 + \sqrt{3}, \qquad \tan 15° = 2 - \sqrt{3},$$

we obtain

$$r = 6(2+\sqrt{3}) = 12 + 6\sqrt{3}, \qquad s = 6(2-\sqrt{3}) = 12 - 6\sqrt{3}.$$

Therefore

$$r^2 + s^2 = (12+6\sqrt{3})^2 + (12-6\sqrt{3})^2 = 2\big(12^2 + (6\sqrt{3})^2\big) = 504.$$

---

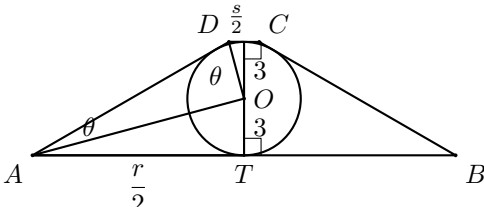

*Figure 13.* Mode E sketch: right triangles at tangency points can yield relations like $\frac{r}{2} = 3\cot\theta$ and $\frac{s}{2} = 3\tan\theta$.

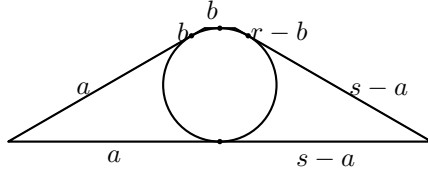

*Figure 14.* Mode B-style tangency bookkeeping: equal tangent segments from a vertex yield repeated segment labels.

## H. Hyperparameters and Training Details

In this section, we describe the training setup used in our experiments, including the evaluation metrics employed to measure diversity and accuracy, the reward function design, and the hyperparameters required to ensure reproducibility.

## H.1. Setup

- **Models.** We study both a *reasoning* model and a *base* (non-RL post-trained) model: `deepseek-ai/DeepSeek-R1-Distill-Qwen-7B` (7B) and `qwen3-14B-base` (14B).

- **Evaluation.** We report downstream math reasoning accuracy on AIME 2024 and AIME 2025.

- **Training data.** We train on `DAPO-17K`. We intentionally include an older-generation reasoning model (DeepSeek-R1 distill) and a base model with lower benchmark capability to reduce the risk that our evaluation is dominated by highly-optimized recent baselines.

- **Compute + schedule.** Global batch size 256, trained for 8 epochs on $4\times8$ H200 GPUs.

- **Rollout budget.** We use $G=16$ rollouts per prompt. This modest budget was sufficient to observe and shape diversity, while remaining practical for throughput.

- **Clustering pipeline.** We embed rollouts with `sentence-transformers/all-MiniLM-L6-v2` and cluster in embedding space using DBSCAN (scikit-learn) to estimate reasoning-mode masses.

- **Verification.** We use a rule-based verifier that scores rollouts by exact match on the extracted final answer (GSM-style `\boxed {...}`).

- **Baselines and implementation.** The GRPO baseline uses DAPO-inspired implementation choices (asymmetric clipping and `loss_agg_mode=token-mean`). We disable KL regularization for all methods to isolate the effect of the diversity bonus.

- **Reproducibility.** Full hyperparameters and implementation details are reported in the following subsections.

## H.2. Metrics.

We report both *task performance* and *mode-dynamics* during training. For performance, we evaluate AIME 2024/2025 accuracy as pass@1, averaged over 30 independent evaluation runs (i.e., an empirical $\mathbb{E}[\text{pass@1}]$). For dynamics, we log prompt-wise estimates $(\hat{p}, \hat{y}, \hat{z})$ and aggregate them over each training batch $\mathcal{D}$ (here $|\mathcal{D}| = 256$ prompts) using the batch mean

$$\langle s \rangle_{\mathcal{D}} \; := \; \frac{1}{|\mathcal{D}|} \sum_{i \in \mathcal{D}} s_i.$$

This provides a low-variance view of the same trends observed at the single-prompt level. Concretely, we track:

- **Bad mass.** `rarity/avg_p_hat` $= \langle \hat{p} \rangle_{\mathcal{D}}$.

- **Mode counts.** `rarity/avg_K` $= \langle K \rangle_{\mathcal{D}}$ (good clusters), `rarity/avg_Mbad` $= \langle M \rangle_{\mathcal{D}}$ (bad clusters).

- **Concentration (geometry).** For each prompt $i$, define $L_{y,i} = \|\hat{y}_i\|_2^2$ and $L_{z,i} = \|\hat{z}_i\|_2^2$. We log `rarity/avg_L2y` $= \langle L_y \rangle_{\mathcal{D}}$ and `rarity/avg_L2z` $= \langle L_z \rangle_{\mathcal{D}}$ (smaller $\Rightarrow$ more diverse).

- **Effective mode count.** `rarity/avg_effK` $= \langle 1/L_y \rangle_{\mathcal{D}}$, which is the standard effective-number proxy $K_{\text{eff}} \approx 1/\|\hat{y}\|_2^2$.

- **Entropy-style diversity.** For a prompt with $K$ good clusters, we compute $H(\hat{y}) = -\sum_{j=1}^{K} \hat{y}_j \log \hat{y}_j$ and the normalized entropy $\widetilde{H}(\hat{y}) = H(\hat{y})/\log K \in [0,1]$. We log `rarity/avg_normH` $= \langle \widetilde{H} \rangle_{\mathcal{D}}$ (values near 1 indicate an approximately uniform spread over observed good modes).

- **Entropy-style diversity metrics.** For the estimated good-mode distribution $\hat{y} \in \Delta^{K-1}$ (from clustering), we report the Shannon entropy

$$H(\hat{y}) \; = \; -\sum_{j=1}^{K} \hat{y}_j \log \hat{y}_j, \qquad \widetilde{H}(\hat{y}) \; = \; \frac{H(\hat{y})}{\log K} \in [0, 1],$$

where $\widetilde{H} \approx 1$ indicates an (approximately) uniform spread over observed good modes. We also log model uncertainty at two granularities. The *token-level entropy* of a rollout $h = (a_{1:T})$ is the average entropy of the next-token distribution along the generated trajectory,

$$H_{\text{tok}}(h) \;=\; \frac{1}{T}\sum_{t=1}^{T} H\big(\pi_\theta(\cdot \mid x, a_{<t})\big), \qquad H(\pi_t) \;=\; -\sum_{v\in\mathcal{V}} \pi_t(v)\log\pi_t(v),$$

(optionally normalized by $\log|\mathcal{V}|$). rollouts.

### H.3. Hyperparameters.

Table 16 summarizes the training and evaluation configuration. we fine-tune `DeepSeek-R1-Distill-Qwen-7B` and `Qwen3-14B-Bsse` with a global prompt batch size of 256 for 10 epochs. For rollout generation, we sample $G{=}16$ responses per prompt at temperature 1.0, truncating prompts/responses at 2048 and 16384 tokens. For actor optimization, we use GRPO with asymmetric PPO clipping $(\varepsilon_{\text{low}}, \varepsilon_{\text{high}}) = (0.20, 0.28)$ and Adam with learning rate $10^{-6}$, while disabling KL regularization (all KL coefficients set to 0). Validation is performed with sampling at temperature 0.6 using $n{=}30$ rollouts per prompt.

### H.4. Reward Script and Granular Rarity Bonus

**Rarity bonus from reasoning-mode clusters.** To complete the end-to-end picture of G$^2$RPO algorithm spells out the reward-side implementation that augments the base accuracy signal with a diversity-aware `rarity_bonus`. We first compute a binary task score $\text{score}_i \in \{0, 1\}$ via exact match. On the training split (and when embeddings are available), we group rollouts by problem id, embed all solutions, and cluster the *correct* rollouts (DBSCAN) to obtain $K$ reasoning-mode clusters. Let $y$ denote the normalized histogram over these correct clusters (after a minimum-probability clip $\rho$ and renormalization), and define the concentration proxy $L_y = \sum_j y_j^2$. Each correct cluster $j$ receives a centered and clipped bonus

$$B_j \propto \lambda\Big(\frac{1}{K\,y_j} - 1\Big),$$

so underrepresented correct modes ($y_j$ small) are upweighted while overly dominant modes are downweighted, encouraging exploration across distinct correct strategies.

**Scheduling and bad-mass neutralization.** The overall scale $\lambda$ is scheduled using the empirical incorrect mass $\hat{p}$ (fraction of wrong rollouts within a prompt group) and can be attenuated using concentration statistics from both correct and incorrect clusters (via $L_y$ and $L_z$) to improve stability when wrong solutions are highly clustered. Optionally (`bad_mass_neutral=True`), we assign a constant bonus $B_{\text{bad}}$ to incorrect rollouts chosen to cancel the net drift induced by the cluster bonuses when $\hat{p} > 0$, preventing unintended shifts in the mean update direction. The reward function finally returns per-sample dictionaries {`score`, `rarity_bonus`}, where `rarity_bonus` is later broadcast across response tokens and added to token-level advantages

*Listing 1.* Batch reward computation with a cluster-based `rarity_bonus` (G$^2$RPO) written in the format of VeRL RL training library. For each prompt, we compute the base task score (exact match), embed rollouts, cluster correct (and optionally incorrect) solutions into reasoning-mode groups, and return a per-sample bonus that upweights underrepresented correct modes while optionally neutralizing drift from incorrect mass.

```
def compute_score_batch(solutions, ground_truths, extra_info,
            lambda0, rho, bad_mass_neutral=True,
            use_lambda_attenuation=True):
  # 1) base task reward (accuracy)
  score = [0.0] * B
  for i in range(B):
    score[i] = exact_match(extract_final(solutions[i]),
            extract_final(ground_truths[i]))

  # 2) only compute rarity on train split
  bonus = [0.0] * B
```

```
if split != "train" or not embedder_ok:
    return [{"score": score[i], "rarity_bonus": 0.0} for i in range(B)]

# 3) group by prompt/problem id
groups = group_by_problem(extra_info)   # pid −> list of indices
E = embed_all(solutions)                # embeddings for all rollouts

for pid, I in groups.items():
    C = [i for i in I if score[i] == 1.0]   # correct
    W = [i for i in I if score[i] == 0.0]   # wrong
    if len(C) < 2:
        continue

    # (a) estimate incorrect mass and base schedule
    p_hat = float(len(W)) / float(len(I))
    if 0.0 < p_hat < 1.0:
        lam_p = lambda0 * sqrt(p_hat / max(1e−8, 1.0 − p_hat))
    else:
        lam_p = 0.0

    # (b) cluster correct solutions
    labels = dbscan([E[i] for i in C])      # cluster id per correct sample
    K = num_clusters(labels)
    y = cluster_hist(labels, K)             # counts per cluster
    y = [v / float(len(C)) for v in y]      # normalize
    y = clip_min_and_renorm(y, rho)         # min−prob clip + renorm
    Ly = sum(v*v for v in y)
    invK = 1.0 / float(K)

    # (c) bad concentration proxy
    if len(W) == 0:
        Lz = 0.0
    elif len(W) == 1:
        Lz = 1.0
    else:
        labels_bad = dbscan([E[i] for i in W])
        M = num_clusters(labels_bad)
        z = cluster_hist(labels_bad, M)
        z = [v / float(len(W)) for v in z]
        Lz = sum(v*v for v in z)

    # (d) optional attenuation−cancel schedule
    if use_lambda_attenuation and len(W) > 0:
        lam = lam_p * (Ly + Lz) / (Lz + invK)
    else:
        lam = lam_p

    # (e) rarity bonus for good clusters
    Bc = [lam * (1.0 / (float(K) * yj) − 1.0) for yj in y]
    meanB = sum(yj * Bj for (yj, Bj) in zip(y, Bc))
    Bc = [Bj − meanB for Bj in Bc]
    Bc = [clip(Bj, −B_max, B_max) for Bj in Bc]

    # (f) optional constant bonus for incorrect rollouts
```

```
if bad_mass_neutral and p_hat > 0.0 and len(W) > 0:
    m1 = sum(yj * Bj for (yj, Bj) in zip(y, Bc))
    S  = sum((yj*yj) * Bj for (yj, Bj) in zip(y, Bc))
    B_bad = (S − m1 * ((1.0 − p_hat)*Ly − p_hat*Lz)) / (p_hat * (Ly + Lz))
    B_bad = clip(B_bad, −B_max, B_max)
else:
    B_bad = 0.0

# (g) assign per−sample rarity_bonus
for idx, cid in zip(C, labels):
    bonus[idx] = Bc[cid]
for idx in W:
    bonus[idx] = B_bad

return [{"score": score[i], "rarity_bonus": bonus[i]} for i in range(B)]
```

*Table 16.* Training configuration.

| System & Data | |
| --- | --- |
| Base model | `DeepSeek-R1-Distill-Qwen-7B` or `Qwen3-14B-Base` |
| Nodes $\times$ GPUs/node | $4 \times 8$ (32 GPUs total) |
| Train batch size (prompts) | 256 |
| Total rollouts / train step | $256 \times 16 = 4096$ |
| Total epochs | 8 |
| Optimization steps | 560 |
| Train file(s) | DAPO-17k |
| Validation/Test file(s) | AIME24 + AIME25 |
| **Rollout generation (vLLM)** | |
| Rollouts per prompt ($G$) | 16 |
| Temperature | 1.0 |
| Top-$p$/Top-$k$ | default (1,-1) |
| Max prompt length | 2048 |
| Max response length | 16384 |
| dtype | bfloat16 |
| Max batched tokens | 36864 |
| GPU memory utilization | 0.9 |
| Tensor model parallel size | 1 |
| **Actor optimization (GRPO)** | |
| Advantage estimator | GRPO |
| PPO epochs | 1 |
| PPO mini-batch size | 64 |
| PPO micro-batch / GPU | 32 |
| Clipping $\varepsilon_{\text{low}}$ | 0.20 |
| Clipping $\varepsilon_{\text{high}}$ | 0.28 |
| Loss aggregation | `token-mean` |
| Optimizer | Adam (lr $10^{-6}$; other settings default) |
| KL in reward / KL in loss | disabled (`use_kl_in_reward`=False; coefficients 0) |
| **Reward / diversity controls** | |
| Diversity weight | {0, 1.5, 3.0} |
| Reward manager | batch |
| **Evaluation & logging** | |
| Validation sampling | `do_sample`=true, $T = 0.6$, $n = 30$ |

