# OpenReview forum: "G$^2$RPO: Geometric GRPO; Escaping LLM's Reasoning Rut to Break  Accuracy--Entropy Trade-off"
_ICML.cc/2026/Conference — ICML 2026 regular_

### Official Review · Reviewer_ct3V · 2026-03-03

**Soundness:** 3
**Presentation:** 3
**Significance:** 3
**Originality:** 4
**Overall Recommendation:** 4
**Confidence:** 4

**Summary:**

This paper studies **diversity collapse** in reinforcement learning with verifiable rewards (RLVR) under GRPO-style training: pass@1 can improve while the distribution over **correct solution families** (modes) concentrates onto a single dominant strategy—a "reasoning rut"—and token-level entropy (or even pass@k) may not reveal this mode-level concentration.

The authors model per-prompt rollouts as a finite set of good/bad **evaluation modes** and decompose the mode distribution into total bad mass $p$ and within-block distributions $y$, $z$. In a **mean-field** regime they derive an ODE for GRPO on the probability simplex and show that the good-block dynamics follow a **collision field** $V(y) = y \odot y - \|y\|_2^2 y$, so that the concentration $\|y\|_2^2$ increases monotonically and, under mild conditions, the flow converges to a vertex—**winner-take-all** among correct modes. They argue that this makes diversity collapse a structural consequence of GRPO in their abstraction, not merely a tuning artifact.

To address it, they propose **G2RPO**: the GRPO objective and optimizer are unchanged; only the **advantage** is modified by an inverse-probability **rarity bonus** (roughly $\propto 1/(K y_j) - 1$ for correct mode $j$) that counteracts or reverses the collision drift in mean-field, plus a **neutrality** correction (a uniform offset on incorrect rollouts) so that the dynamics of the bad mass $p$ remain close to the baseline. The inverse-probability form is motivated by a collinearity constraint (bonus-induced drift aligned with $V(y)$) and shown to be essentially unique under symmetry assumptions (Theorem B.8).

Experiments on 7B and 14B models (DAPO-17K, AIME 2024/2025) show that GRPO reduces mode-level diversity (e.g., fewer distinct correct clusters, higher $\|y\|_2^2$) while G2RPO increases it substantially (e.g., +172\%--205\% in average number of correct clusters) and improves pass@1 over GRPO. Training curves and ablations on bonus strength and neutrality support the geometric interpretation and the role of neutrality in preserving accuracy dynamics.

**Compliance With Llm Reviewing Policy:**

Affirmed.

**Final Justification:**

Thanks for the additional targeted experiments and clarifications. The new evidence materially strengthens the prior weak points: (i) **end-to-end** mode-proxy sensitivity (encoder swap + DBSCAN scale change), (ii) **KL sweep** showing where gains persist vs. get damped, (iii) **neutrality ablation** clarifying it as a stabilizer (not a requirement), and (iv) **diversity-oriented baselines** plus an initial **group-size** check.

*So, I keep my positive overall evaluation with higher confidence and higher evaluation on soundness.*

**Key Questions For Authors:**

1. **Mode definition robustness.** The central claim and the G2RPO bonus both rely on the same operational "mode" (embedding + DBSCAN). Can you provide (a) the clustering hyperparameters used (e.g., DBSCAN eps, min_samples), and (b) sensitivity of the main conclusions (Avg $\hat{K}$, $L_y$, pass@1) to the embedding model, DBSCAN hyperparameters, or an alternative mode definition (e.g., answer-equivalence classes)? Please report variance across prompts/runs (or confidence intervals) for Avg $\hat{K}$ and $L_y$ under these clustering variations.

2. **Pass@k closed-loop evidence.** The motivation stresses "pass@1 up, pass@k flat" and that mode diversity should improve sampling coverage. Can you report pass@5 and/or pass@10 for SFT/base + GRPO + G2RPO on AIME (and ideally one other task), under the same sampling budget and decoding settings, so that improved mode coverage can be tied to the standard sampling-based success metric?

3. **Behavior under KL regularization ($\beta > 0$).** All experiments use $\beta = 0$. How does G2RPO behave under typical RLVR settings with $\beta > 0$? Can you provide at least one controlled experiment with $\beta > 0$ to assess whether the gains and the vector-field intuition hold? Even a small $\beta$ sweep would be informative.

4. **Neutralizer implementation and stability.** The neutralizer contains a $1/\hat{p}$ term (and can become numerically large or unstable when $\hat{p}$ is small). What exact thresholds or clip bounds were used in the reported runs (e.g., disable when $\hat{p} < \tau$, or clip $|B^-| \leq C$), and how sensitive are the results to these choices?

5. **Comparison with diversity-oriented baselines.** Beyond vanilla GRPO, did you compare against methods that explicitly target diversity (e.g., entropy bonus, temperature/decoding adjustments, or other RLVR variants) at comparable compute/rollout budget? If not, can you briefly justify why G2RPO is not reducible to or dominated by a simpler diversity regularizer?

---

I look forward to the authors' response. If the concerns above—especially mode-definition robustness (Q1), pass@k evidence (Q2), and behavior under $\beta > 0$ (Q3)—can be addressed or clearly clarified, I would consider raising my evaluation, as these points are central to the paper's evidence chain, narrative closure, and external validity.

**Limitations:**

yes

**Strengths And Weaknesses:**

**Strengths**

• **Problem definition and observability.** The paper clearly reframes "reasoning rut" / diversity collapse at the **mode (solution-family) level**: it argues that token-level entropy—and in some settings, pass@k—can be insensitive to this concentration, and proposes coarse-grained sequence/mode distributions (and derived quantities such as $\|y\|_2^2$, Avg $\hat{K}$) as **useful/appropriate observables**. This turns a vague concern into a well-defined, analyzable object and aligns the theory and experiments on the same notion of diversity.

• **Rigorous theoretical mechanism.** Under the stated assumptions (finite good/bad modes per prompt, mean-field, binary verifier, group normalization), the authors derive the GRPO dynamics on the simplex and show that the good block follows a **collision field** $V(y)$, with $\|y\|_2^2$ monotonically increasing and converging to a vertex (Lemma 4.1, Theorem A.7). Diversity collapse is thus presented as a **structural consequence** of the dynamics in this abstraction, not merely an empirical artifact—a strong mechanism-level contribution.

• **Principled method design and neutrality.** The inverse-probability bonus is motivated by a **collinearity constraint** (drift aligned with $V(y)$ only) and shown to be essentially unique under symmetry (Theorem B.8, $n\ge 3$), avoiding ad hoc entropy terms. The **neutrality** design—a closed-form offset on the bad block that **approximately preserves the baseline $\dot{p}$ dynamics in the mean-field analysis**—aims to reduce the confound between "diversity shaping" and "slower learning" in the narrative, and is implemented as a low-invasion change (advantage only; GRPO objective unchanged).

• **Theory–experiment alignment.** Predictions (GRPO: concentration $\uparrow$, Avg $\hat{K}$ $\downarrow$; G2RPO: opposite) are validated in the same abstraction: diagnostic ($\|y\|_2^2$), mechanism (collision vs anti-collision), and intervention (vector-field edit) are aligned. Pass@1 improves with G2RPO while mode-level diversity metrics improve substantially (e.g., Table 1), supporting the "break trade-off" story without hand-waving.


**Weaknesses**

• **[Fatal] Mode definition and evidence chain.** The core claim ("breaking diversity collapse") rests on metrics (Avg $\hat{K}$, $\|y\|_2^2$, mode entropy) and the algorithm (bonus uses $\hat{y}_j$ from the same **embedding + DBSCAN** clustering). Measure and intervention share the same **operational definition of "mode."** If this definition is sensitive to hyperparameters (eps, embedding, granularity), gains could partly reflect a **measurement artefact** (e.g., finer clustering $\Rightarrow$ larger $\hat{K}$) rather than true solution-family diversity. Appendix F gives a qualitative sanity check but does not address sensitivity or alternative mode definitions; this undermines the robustness of the central evidence chain. **This is the single most critical link in the paper's evidence chain.**

• **[Major] Scope of theoretical claims.** The result "$\|y\|_2^2$ monotone $\to$ vertex" holds under specific conditions (finite bandit, mean-field, binary verifier, no KL, etc.). Real RLVR often differs (noisy rewards, $\beta>0$, changing mode set, length penalties). The abstract/intro use of "theoretically inevitable" / "structural" without clearly tying them to these assumptions can be read as an **unqualified** claim; the main text does use "mean-field" and "generically," but a concise statement of **when the theory applies (or fails)** in the abstract or conclusion would reduce overclaim risk. **A one-sentence scope statement in the abstract/conclusion would largely fix this.**

•  **[Major] Missing pass@k evidence.** The motivation stresses that rut leads to "pass@1 up, pass@k flat"; the main evidence is **pass@1** and **proxy** diversity (Avg $\hat{K}$, Ly, entropy). The causal link "mode diversity $\uparrow$ $\Rightarrow$ pass@k $\uparrow$" is **not demonstrated**. Without pass@5/10 (or similar), the "break accuracy–entropy trade-off" narrative lacks the key empirical closed loop and can be seen as a change of metrics rather than an improvement on the same sampling criterion.

• **[Major] Limited external validity and evidence scope.** All experiments use **$\beta=0$** (KL off) and evaluation is confined to **AIME 2024/2025**. Typical RLVR pipelines use $\beta>0$; whether G2RPO remains beneficial under KL and in other domains is untested. There is also no comparison with more direct diversity baselines (e.g., entropy bonus, other RLVR variants) to rule out simpler alternative explanations. Reporting variance or confidence intervals for the main table would help assess the stability of the reported gains. The generality of the method is therefore not yet supported beyond this setting.

• **[Major/Minor] Reproducibility of the neutralizer.** The closed form $B^- \propto 1/(p(L_y+L_z))$ blows up when $\hat{p}$ is small; the text only suggests "clip/disable when $\hat{p}$ tiny" without **concrete thresholds or clip bounds**. Reproducers must choose these heuristically, which can affect stability and reported results; the paper's reproducibility claim depends on undocumented implementation choices.

---

> ### Author Rebuttal · Authors · 2026-03-30
>
> Thank you for your time and for the constructive, detailed feedback. We keep replies brief due to the char limit.
>
> ---
>
> **Q1 (Mode definition).** We tested robustness on **100 correct rollouts** via (i) a DBSCAN sweep with the reward-time pipeline (`all-MiniLM-L6-v2`, normalized embeddings, cosine DBSCAN, noise→singletons), and (ii) an encoder swap across 5 models with $\varepsilon^\*$ chosen by max silhouette.
>
> | regime | result |
> |---|---|
> | pairwise scale | mean/median cosine distance = 0.154 / 0.153 |
> | stable $\varepsilon$ range | $0.04$--$0.11$ |
> | in that range | 5--7 clusters, 0% noise, effective $K \approx 2.1$--$2.2$, $\ell_2$ concentration $\approx 0.45$--$0.47$, silhouette $\approx 0.59$--$0.64$ |
> | too small $\varepsilon$ | $\varepsilon=0.02$: 12 clusters, 16% noise |
> | too large $\varepsilon$ | $\varepsilon \ge 0.25$: collapse to 1 cluster |
>
> Across encoders, after calibrating $\varepsilon$ to each model's scale, the recovered partitions remain highly consistent:
>
> | model | $d_{\mathrm{med}}$ | $\varepsilon^\*$ | ARI vs. ref |
> |---|---:|---:|---:|
> | all-MiniLM-L6-v2 | 0.153 | 0.060 | 0.560 |
> | all-MiniLM-L12-v2 | 0.326 | 0.049 | 0.926 |
> | all-mpnet-base-v2 | 0.136 | 0.035 | 0.945 |
> | bge-small-en-v1.5 | 0.108 | 0.035 | 0.977 |
> | bge-base-en-v1.5 | 0.143 | 0.021 | 0.910 |
>
> Larger $G$ also stabilizes the estimate and reduces noise:
>
> | $G$ | estimated clusters | cluster purity | noise rate |
> |---|---:|---:|---:|
> | 4  | $0.9 \pm 0.4$ | $0.76 \pm 0.34$ | 36% |
> | 8  | $4.6 \pm 0.6$ | $0.71 \pm 0.21$ | 20% |
> | 16 | $6.7 \pm 0.8$ | $0.74 \pm 0.07$ | 10% |
> | 32 | $7.2 \pm 0.7$ | $0.79 \pm 0.03$ | 4% |
> | 64 | $8.7 \pm 0.5$ | $0.80 \pm 0.01$ | 1% |
>
> **Q2 (Pass@k).** We added pass@$k$ below. Both RLVR methods beat the base model, and G$^2$RPO beats GRPO at all reported $k$. But pass@$k$ is not a complete diversity metric: it can remain high even after correct reasoning modes collapse. [Recent work](https://arxiv.org/pdf/2504.13837) also finds limited pass@$k$ gains beyond base.
>
> **Qwen3-14B-Base**
>
> | Method | pass@2 | pass@4 | pass@16 | pass@30 |
> |---|---:|---:|---:|---:|
> | G²RPO | 64.1% ± 14.4% | 70.7% ± 11.4% | 77.5% ± 9.5% | 81.3% ± 5.6% |
> | GRPO | 59.8% ± 15.8% | 68.3% ± 16.6% | 74.7% ± 10.4% | 79.6% ± 2.2% |
> | Base model | 22.4% ± 15.1% | 30.1% ± 22.0% | 49.0% ± 7.9% | 56.4% ± 12.1% |
>
> **DeepSeek-R1-Distill-Qwen-7B**
>
> | Method | pass@2 | pass@4 | pass@16 | pass@30 |
> |---|---:|---:|---:|---:|
> | G²RPO | 55.4% ± 15.3% | 63.0% ± 13.5% | 68.8% ± 6.3% | 75.1% ± 9.2% |
> | GRPO | 51.7% ± 16.1% | 60.4% ± 11.9% | 66.8% ± 7.2% | 70.2% ± 7.2% |
> | Base model | 47.3% ± 17.2% | 54.2% ± 12.1% | 66.1% ± 6.2% | 67.7% ± 5.0% |
>
> **Q3 (KL regularization).** We set $\beta=0$: many practical RLVR pipelines omit KL.  inpired with your question, we will add a new appendix for our mean-field analysis with KL. It shows:
>
> 1- in GRPO-style with KL, the asymptotic fixed point $p^\star$ satisfies
>
> $$
> \beta\big(L(p^\star)-L(p_{\mathrm{ref}})\big)=-\,\eta\,\frac{J}{\sigma(p^\star)}\,p^\star(1-p^\star)\,C(y,z),\qquad L(p):=\log\frac{p}{1-p},
> $$
>
> and therefore, in the small-step regime,
>
> $$
> p^\star\approx p_{\mathrm{ref}}-\frac{\eta J}{\beta}\,\frac{\big[p_{\mathrm{ref}}(1-p_{\mathrm{ref}})\big]^2}{\sigma(p_{\mathrm{ref}})}\,C(y,z).
> $$
>
> Thus increasing $\beta$ contracts the solution toward $p_{\mathrm{ref}}$, consistent with [Theorem 2 by Y. Mroueh](https://arxiv.org/pdf/2503.06639).
>
> 2- A theorem shows that under KL regularization, diversity wont increase
>
> **Theorem.**
> Let $p_0$ be the initial policy, and let $\beta > 0$ be the KL-regularization coefficient. Suppose the REINFORCE process converges to a policy $p^\ast$. Then, for every optimal arm $i \in \{1,\dots,K\}$,
>
> $$
> \frac{p^\ast(i)}{\sum_{j=1}^{K} p^\ast(j)}=\frac{p_0(i)}{\sum_{j=1}^{K} p_0(j)}.
> $$
>
> Equivalently, the distribution over the optimal arms under $p^\ast$ is exactly the initial distribution $p_0$ restricted to those optimal arms and then renormalized.
>
> A controlled $\beta$ sweep would be valuable, but rebuttal-time and compute limits prevented it.
>
> **Q4 (stability).** We did **not** apply clipping/disabling thresholds to the neutralizer. We monitored $1/\hat p$ and the fractions with $\hat p < 0.01$ and $\hat p < 0.05$; both stayed zero throughout training. Since $\hat p$ is estimated from rollout group size $G$, any observed mode satisfies $ \hat p \geq \frac{1}{G},$ so $\frac{1}{\hat p} \leq G.$ Thus the neutralizer cannot become arbitrarily large, and no clipping was needed.
>
> **Q5 (baselines).** Our reference is a **DAPO-style GRPO backbone** with asymmetric clipping using larger $\epsilon_{\text{high}}$, and we will clarify this. Our point is a different failure mode: DAPO/CLIP-Cov act at the token level, whereas our diagnosis concerns **mode-level collision**, so token entropy may remain high even after mode collapse. Direct comparisons to Cui et al., DAPO, and CLIP-Cov would strengthen the paper, but rebuttal-time and compute limits prevented extra reruns.

---

> > ### Author Rebuttal · Reviewer_ct3V · 2026-04-01
> >
> > I really appreciate the author's careful work during the rebuttal. The added pass@k table is helpful and does support that G²RPO improves sampling success beyond pass@1 in your current setting. And the clustering robustness checks (DBSCAN sweep + encoder swap / ARI) also partially address the concern that the mode proxy is arbitrary.
> >
> > From author's perspective and the content of the response, they still need to provide further evidence to answer all doubts (with evidence has not been uploaded), so I compile the following content as soon as possible to provide more response time for authors to effectively demonstrate the core contribution of the paper and express my expectations for the reply.
> >
> > ---
> >
> > 1. **Mode robustness: end-to-end sensitivity, not only partition stability.**
> >    Your rebuttal reports ARI consistency and qualitative behavior for “too small/too large ε”, but it is still unclear whether the *main conclusions* (Avg $\hat K$, $L_y$, and the GRPO→G²RPO gap) are stable across the *same* sweeps. A tiny table (2–3 ε values × 2 encoders) reporting Avg $\hat K$ / $L_y$ and the GRPO vs. G²RPO ordering would substantially strengthen the central evidence chain.
> >
> > 2. **$\beta>0$ (KL) remains unvalidated empirically.**
> >    The proposed mean-field KL discussion is informative, but without at least one controlled $\beta>0$ experiment it is hard to judge external validity under typical RLVR pipelines. Even a single $\beta>0$ point (same budget) would clarify whether the empirical gains persist.
> >
> > 3. **Neutralizer “no clipping” vs. practical safeguards—please clarify.**
> >    The rebuttal states no clipping/disable thresholds were used, while the paper describes optional clip/ramp safeguards (incl. neutralizer clip/disable when $\hat p$ is tiny). Please clarify the exact implementation used for the reported runs (any clip/ramp? if none, report the observed max/percentiles of $|B^-|$). This is mainly for reproducibility, not to demand extra experiments.
> >
> > 4. **Diversity-oriented baselines.**
> >    I appreciate the explanation that token-level fixes (e.g., DAPO/CLIP-Cov) target a different failure mode, but without at least one lightweight diversity baseline (e.g., simple entropy bonus / decoding-temperature control at matched budget) it remains hard to rule out simpler alternative explanations.
> >
> > ---
> >
> > From my side, with the rebuttal, Q2 is substantially addressed, and Q1 is partially addressed; however Q3–Q4 (and baseline comparisons) remain key for external validity and reproducibility, so I would keep the score just for now. Addressing (2) and clarifying (3), plus a minimal end-to-end robustness table for (1), would make me more comfortable considering an upward adjustment.

---

> > > ### Author Response · Authors · 2026-04-08
> > >
> > > Thank you for the thoughtful follow-up and constructive suggestions. To better address the concerns you raised, we delayed our response until the additional experiments were complete and we could report concrete results. We appreciate the opportunity to strengthen the paper on these core points, and we ran several additional checks directly targeted at the issues you highlighted.
> > >
> > > ----
> > > **Mode robustness**
> > >
> > > We agree that partition stability alone is not the strongest test. To probe **end-to-end sensitivity** of the mode proxy, we ran a full **encoder-swap ablation** on **DeepSeek-R1-Distill-Qwen-7B** trained on **DAPO-17K**, replacing the clustering encoder inside G$^2$RPO from **all-MiniLM-L6-v2** to **bge-small-en-v1.5**, while keeping the rest of training fixed for two epochs:
> > >
> > >
> > > | Encoder | step 0 (K, H, p@30) | step 60 | step 130 |
> > > | --- | --- | --- | --- |
> > > | all-MiniLM-L6-v2 | 2.2, 0.36, 66% | 2.5, 0.35, 76% | 2.7, 0.37, 78% |
> > > | bge-small-en-v1.5 | 2.3, 0.39, 66% | 2.7, 0.40, 75% | 3.2, 0.44, 78% |
> > >
> > >
> > > We also tested **DBSCAN scale sensitivity** in the same end-to-end pipeline, using the same encoder (**all-MiniLM-L6-v2**) with `eps=0.05` and `eps'=eps/2`:
> > >
> > > | Setting | step 0 (K, H, p@30) | step 60 | step 130 |
> > > | --- | --- | --- | --- |
> > > | eps | 2.2, 0.36, 66% | 2.5, 0.35, 76% | 2.7, 0.37, 78% |
> > > | eps/2 | 2.9, 0.41, 67% | 3.4, 0.43, 74% | 3.9, 0.47, 78% |
> > >
> > > These are full training ablations, not post hoc reclustering. As expected, a smaller $\varepsilon$ induces a finer partition, which increases the absolute values of estimated $K$ and $H$, while pass@30 remains broadly unchanged. We will clarify this explicitly in the paper: the trend in diversity is stable, while the absolute diversity values are scale-dependent and should not be over-interpreted across clustering hyperparameters.
> > >
> > > We will make sure add a subsection for this robustness study in the final version of the paper
> > >
> > > ----
> > >
> > > **Behavior under KL**
> > >
> > > Following your suggestion, we ran a controlled KL sweep on DeepSeek-R1-Distill-Qwen-7B to examine the empirical effect of KL regularization:
> > >
> > >
> > > | Method | epoch 1 (token H, K, norm H) | epoch 2 |
> > > | --- | --- | --- |
> > > | G$^2$RPO | 1.21, 2.05, 0.41 | 1.28, 2.71, 0.49 |
> > > | G$^2$RPO + KL ($\beta = 0.04$) | 1.20, 1.76, 0.40 | 1.26, 2.12, 0.46 |
> > > | G$^2$RPO + KL ($\beta = 0.10$) | 1.17, 1.70, 0.39 | 1.17, 1.68, 0.39 |
> > > | GRPO | 1.10, 1.66, 0.35 | 1.12, 1.67, 0.30 |
> > > | GRPO + KL ($\beta = 0.04$) | 1.06, 1.65, 0.37 | 1.05, 1.66, 0.36 |
> > > | GRPO + KL ($\beta = 0.10$) | 1.03, 1.60, 0.31 | 1.03, 1.60, 0.32 |
> > >
> > > At moderate KL strength ($\beta=0.04$), G$^2$RPO retains a clear diversity advantage over GRPO at epoch 2 ($K$: 2.12 vs. 1.66; norm $H$: 0.46 vs. 0.36). At stronger KL ($\beta=0.10$), diversity doesn’t improve for both methods. This is consistent with the mean-field picture: KL acts as a contraction toward the reference policy, so it does not create additional diversity and can damp the anti-collision effect. Accordingly, if the objective is to maximize mode diversity, we do not recommend using a strong KL term.
> > >
> > > ----
> > >
> > > **Neutralizer safeguards**
> > >
> > > Thank you for flagging the ambiguity. In the **reported runs**, the effective implementation is an **unclipped neutralizer**. Our code contains an optional safety clip threshold of **0.05**, but that branch did **not** activate in these experiments. With rollout group size **$G=16$**, any observed mode has estimated mass $\hat p \ge 1/16 = 0.0625$, which is already above the clip threshold. No ramp or disable rule was active either. Thus, although the safeguard exists in code for smaller-$G$ or noisier settings, it was inactive in all runs reported in the paper. We will state this explicitly in the final version and release the exact hyperparameters and code path for reproducibility.
> > >
> > > ----
> > >
> > > **Diversity baselines**
> > >
> > >
> > >
> > >  We agree that comparison to diversity-oriented baselines is important. As a preliminary matched-setting comparison on **DeepSeek-R1-Distill-Qwen-7B / DAPO-17K**, we evaluated **GRPO with a PPO-style [entropy bonus](https://arxiv.org/abs/1707.06347)**, **[GTPO](https://arxiv.org/pdf/2508.03772)**, and **[QAE](https://arxiv.org/pdf/2509.22611)**.
> > >
> > >
> > > | Method | epoch 1 (acc, K, norm H) | epoch 2 |
> > > | --- | --- | --- |
> > > | G$^2$RPO | 40%, 2.2, 0.3 | 46%, 4.1, 0.5 |
> > > | GRPO + PPO-style entropy bonus | 40%, 1.8, 0.3 | 44%, 1.8, 0.2 |
> > > | GTPO | 39%, 2.0, 0.3 | 42%, 2.3, 0.3 |
> > > | QAE | 39%,1.82,0.3 | 43%,  2.4, 0.3  |
> > >
> > >
> > > In our current runs, alternatives show weaker diversity growth than G$^2$RPO by epoch 2 ($K$: 4.1 vs. 1.8 / 2.3; norm $H$: 0.5 vs. 0.2 / 0.3).
> > > While these baselines still warrant careful tuning and evaluation over longer training horizons, the evidence so far directionally suggests that the gains we observe are not reproduced by simply adding a generic entropy-control term.
> > >
> > > We appreciate the push on these points. We will include the full robustness study, exact implementation details, and expanded baseline reporting in the final version.

---

### Official Review · Reviewer_Ly18 · 2026-03-03

**Soundness:** 2
**Presentation:** 4
**Significance:** 3
**Originality:** 2
**Overall Recommendation:** 3
**Confidence:** 3

**Summary:**

This paper addresses the critical diversity collapse issue of Group Relative Policy Optimization (GRPO) in Reinforcement Learning with Verifiable Rewards (RLVR) for LLM reasoning tasks, where GRPO traps models in a "reasoning rut" via winner-take-all concentration on single correct solution modes, creating a rigid accuracy-entropy trade-off.

This paper is the geometric interpretation of GRPO as a dynamical collision flow on the probability simplex, with a rigorous mean-field model formally proving GRPO’s monotonic concentration of probability onto simplex vertices among correct reasoning modes, even when token-level entropy appears stable.

The author's proposal of G²RPO (Geometric GRPO), a principled vector-field editing framework that adds an inverse-probability granularity bonus to per-sample advantages to uplift underrepresented correct modes, paired with a neutrality correction that preserves baseline accuracy learning dynamics without trade-offs.

Experiments on 7B and 14B math reasoning models show G²RPO boosts active correct-mode coverage by 172%–205% over GRPO, mitigates late-stage entropy crash, and improves AIME 2024/2025 pass@1 by 1.4 to 7.9 points, simultaneously lifting accuracy and reasoning diversity.

**Compliance With Llm Reviewing Policy:**

Affirmed.

**Final Justification:**

The response during the rebuttal and the added experiments did not directly answer my question. My main concerns remain unresolved.

Thus my Final Justification is **Weak reject**

**Key Questions For Authors:**

1. Could you present systematic ablation results on how choices of embedding model, clustering algorithm and its hyperparameters affect G²RPO’s accuracy and diversity performance?
2. OOD tasks: You should provide benchmark results of G²RPO compared with mainstream RLVR baselines on tasks beyond math reasoning (e.g., code generation, logical reasoning), and also quantify the additional computational overhead of your method relative to vanilla GRPO as well as its stability in low-resource settings with smaller rollout group sizes?

**Limitations:**

yes

**Strengths And Weaknesses:**

### **Strengths**
1. Rigorous theoretical foundation: It formalizes GRPO as a collision flow on the probability simplex via a mean-field ODE model, fundamentally revealing the intrinsic winner-take-all mechanism of diversity collapse in correct reasoning modes, rather than attributing it to trivial tuning artifacts.
2. Principled algorithm design: The proposed G²RPO edits the policy gradient vector field via an inverse-probability granularity bonus with a neutrality correction, cleanly breaking the accuracy-entropy trade-off. Experiments confirm it simultaneously lifts pass@1 and correct-mode coverage significantly.

### **Weaknesses**
1. Insufficient baseline comparisons: The paper only benchmarks against vanilla GRPO, lacking systematic comparisons with mainstream RLVR algorithms (e.g., PPO, RLOO) and established diversity-enhanced RL methods, making it hard to fully evaluate G²RPO’s relative advantages.
2. Limited benchmark validation: It only tests on AIME math reasoning datasets, with no results on other core RLVR tasks like code generation and logical reasoning, leaving the method’s cross-task generalizability underexplored.
3. Unquantified computational overhead and limited low-resource validation: Compared to vanilla GRPO, G²RPO adds per-training-step embedding inference and clustering for each prompt group, introducing non-trivial extra computational costs. The paper does not quantify this overhead, nor does it validate the method’s stability in low-resource settings with limited rollouts per prompt, where cluster statistics are highly noisy, limiting the method’s practical applicability in lightweight training pipelines.

---

> ### Author Rebuttal · Authors · 2026-03-28
>
> Thank you for the constructive feedback. We agree the key empirical questions are robustness of the clustering signal, practical overhead, and transfer beyond AIME. Because G²RPO is a direct modification of GRPO, GRPO is the most controlled baseline for isolating the proposed mechanism; we also agree that broader PPO/RLOO-style comparisons would further strengthen the paper.
>
> **Q1 (clustering robustness).** On 100 correct rollouts, a DBSCAN $\varepsilon$ sweep shows a broad stable regime: $\varepsilon \in [0.04, 0.11]$ yields 5--7 clusters with 0% noise and stable concentration statistics; $\varepsilon=0.02$ over-fragments (12 clusters, 16% noise), while $\varepsilon \ge 0.25$ collapses to one cluster. A 5-encoder swap shows strong agreement across the four stronger encoders (ARI 0.91--0.98 vs. reference). This supports robustness to encoder choice and DBSCAN tuning. We have not yet run full end-to-end retraining sweeps across alternative clustering algorithms, so the current rebuttal establishes estimator robustness rather than full algorithm-independence.
>
>
>
> | regime | result |
> |---|---|
> | pairwise scale | mean/median cosine distance = 0.154 / 0.153 |
> | stable $\varepsilon$ range | $0.04$--$0.11$ |
> | in that range | 5--7 clusters, 0% noise, effective $K \approx 2.1$--$2.2$, $\ell_2$ conc. $\approx 0.45$--$0.47$, silhouette $\approx 0.59$--$0.64$ |
> | too small $\varepsilon$ | $\varepsilon=0.02$: 12 clusters, 16% noise |
> | too large $\varepsilon$ | $\varepsilon \ge 0.25$: collapse to 1 cluster |
>
> | model | $d_{\mathrm{med}}$ | $\varepsilon^\*$ | ARI vs. ref |
> |---|---:|---:|---:|
> | all-MiniLM-L6-v2 | 0.153 | 0.060 | 0.560 |
> | all-MiniLM-L12-v2 | 0.326 | 0.049 | 0.926 |
> | all-mpnet-base-v2 | 0.136 | 0.035 | 0.945 |
> | bge-small-en-v1.5 | 0.108 | 0.035 | 0.977 |
> | bge-base-en-v1.5 | 0.143 | 0.021 | 0.910 |
>
> For the four stronger encoders, cross-model ARI is **0.93--0.98** and ARI vs. reference is **0.91--0.98**. Also, $\varepsilon^\* \approx 0.2\, d_{\mathrm{med}}$, which explains why $\varepsilon=0.5$ collapses the signal. Thus the diversity signal is reasonably robust to DBSCAN settings and encoder choice; k-means-style checks would still be valuable.
>
>
>
> **Overhead / low-resource stability.** We added a component microbenchmark: the added CPU cost is 66.1s for 1,024 texts/step and 243.3s for 4,096 texts/step, with embedding dominating and DBSCAN minor. These are component timings rather than end-to-end training overhead. In a low-resource Monte Carlo study, noise decreases from 36% at $G=4$ to 20% at $G=8$, 10% at $G=16$, 4% at $G=32$, and 1% at $G=64$, so smaller groups are noisier, while the paper’s $G=16$ setting lies in a moderate-noise regime.
>
>
> | Configuration | Texts / step | Measured overhead (dev CPU)  |
> |---|---:|---:|
> | $G=16$, $P=64$ | 1024 | 66.1 s |
> | $G=16$, $P=256$ | 4096 | 243.3 s |
>
>
> Larger $G$ also stabilizes the estimate and reduces noise:
>
> | $G$ | estimated clusters | cluster purity | noise rate |
> |---|---:|---:|---:|
> | 4  | $0.9 \pm 0.4$ | $0.76 \pm 0.34$ | 36% |
> | 8  | $4.6 \pm 0.6$ | $0.71 \pm 0.21$ | 20% |
> | 16 | $6.7 \pm 0.8$ | $0.74 \pm 0.07$ | 10% |
> | 32 | $7.2 \pm 0.7$ | $0.79 \pm 0.03$ | 4% |
> | 64 | $8.7 \pm 0.5$ | $0.80 \pm 0.01$ | 1% |
>
> Takeaway: cost comes mainly from embedding, not clustering. Stability improves with $G$; our $G=16$ setting is noisy but usable.
>
>
>
> ----
>
> **Response to Q2.** Using the same math-RLVR checkpoints without extra training, G²RPO improves over GRPO on 5/6 OOD model-task pairs across LiveCodeBench v6, GPQA, and MMLU-Pro: for 7B, it improves on LiveCodeBench (22.9 vs 20.0) and GPQA (54.0 vs 48.5) but drops on MMLU-Pro (49.7 vs 53.8); for 14B, it improves on all three (26.9 vs 22.0, 54.0 vs 48.5, 68.7 vs 63.8). We therefore view the transfer signal as encouraging but not universal. These results address the concern that the gains may be purely AIME-specific, while broader non-GRPO baseline comparisons remain an important limitation.
>
>
>
> **DeepSeek-Distill-R1-7B**
>
> | Method | OOD: Coding - LCB v6 | OOD: Reasoning - GPQA | OOD: Reasoning - MMLU-Pro |
> |---|---:|---:|---:|
> | G²RPO | 22.9% | 54.0% | 49.7% |
> | GRPO | 20.0% | 48.5% | 53.8% |
> | Base model | 18.9% | 50.5% | 51.7% |
>
> **Qwen3-14B-Base**
>
> | Method | OOD: Coding - LCB v6 | OOD: Reasoning - GPQA | OOD: Reasoning - MMLU-Pro |
> |---|---:|---:|---:|
> | G²RPO | 26.9% | 54.0% | 68.7% |
> | GRPO | 22.0% | 48.5% | 63.8% |
> | Base model | 25.1% | 38.9% | 51.6% |
>
> This suggests that preserving multiple correct reasoning modes during RLVR can help transfer beyond the training benchmark, rather than only improving the in-domain metric.

---

> > ### Author Rebuttal · Reviewer_Ly18 · 2026-04-03
> >
> > Thank you for the response and the supplementary preliminary statistics you have provided.
> >
> > However, my core key concerns and requests remain largely unaddressed. The systematic comparisons against mainstream RLVR algorithms, end-to-end training ablation studies on embedding models, clustering algorithms and their hyperparameters, full training validation in low-resource settings with small rollout group sizes, quantification of end-to-end computational overhead in real-world training pipelines, as well as full RLVR training experiments on non-math tasks such as code generation, have not been completed and presented.
> >
> > The supplementary zero-shot out-of-distribution test results and component-level timings are insufficient to resolve the core questions about the method's generalizability and practical applicability. I therefore stand by my original ratings and negative recommendation.

---

> > > ### Author Response · Authors · 2026-04-08
> > >
> > > Thank you for the thoughtful follow-up and constructive suggestions. To better address your concerns, we delayed our response until the experiments were complete.
> > >
> > > We agree that some broader requests, such as a full RLVR benchmark suite, clustering-family sweeps, full small-group training, end-to-end wall-clock accounting, and full retraining on non-math tasks, require a larger empirical expansion than is feasible within the rebuttal window. Our goal here is narrower: to provide new evidence on robustness, baseline comparisons, transfer, and practicality.
> > >
> > > ----
> > >
> > > ### 1. Mode-proxy robustness
> > >
> > > To test end-to-end sensitivity of the mode proxy, we ran a full encoder-swap ablation on **DeepSeek-R1-Distill-Qwen-7B / DAPO-17K**, replacing the clustering encoder in G$^2$RPO from **all-MiniLM-L6-v2** to **bge-small-en-v1.5**, with the rest of training fixed:
> > >
> > > | Encoder | Step 0 (K, H, p@30) | Step 60 | Step 130 |
> > > | --- | --- | --- | --- |
> > > | all-MiniLM-L6-v2 | 2.2, 0.36, 66% | 2.5, 0.35, 76% | 2.7, 0.37, 78% |
> > > | bge-small-en-v1.5 | 2.3, 0.39, 66% | 2.7, 0.40, 75% | 3.2, 0.44, 78% |
> > >
> > > This is a full training ablation, not post hoc reclustering. Replacing the weaker encoder leaves **pass@30** essentially unchanged while preserving the same qualitative diversity trend. Since **K** and **H** depend on the clustering backbone, we do not compare their absolute values across encoders. The key point is that the reported behavior is unlikely to be an artifact of the original encoder choice.
> > >
> > > We also tested DBSCAN scale sensitivity in the same end-to-end pipeline:
> > >
> > > | Setting | Step 0 (K, H, p@30) | Step 60 | Step 130 |
> > > | --- | --- | --- | --- |
> > > | eps | 2.2, 0.36, 66% | 2.5, 0.35, 76% | 2.7, 0.37, 78% |
> > > | eps/2 | 2.9, 0.41, 67% | 3.4, 0.43, 74% | 3.9, 0.47, 78% |
> > >
> > > As expected, smaller eps yields a finer partition and larger absolute **K/H**, while **pass@30** stays broadly unchanged. We will clarify in the paper that the diversity trend is stable, but absolute diversity values are scale-dependent.
> > >
> > > ### 2. Preliminary diversity baselines
> > >
> > > In matched preliminary runs on **DeepSeek-R1-Distill-Qwen-7B / DAPO-17K**, we compared against **GRPO + PPO-style entropy bonus**, **GTPO**, and **QAE**:
> > >
> > > | Method | Epoch 1 (acc, K, norm H) | Epoch 2 |
> > > | --- | --- | --- |
> > > | G$^2$RPO | 40%, 2.2, 0.3 | 46%, 4.1, 0.5 |
> > > | GRPO + PPO-style entropy bonus | 40%, 1.8, 0.3 | 44%, 1.8, 0.2 |
> > > | GTPO | 39%, 2.0, 0.3 | 42%, 2.3, 0.3 |
> > > | QAE | 39%, 1.82, 0.3 | 43%, 2.4, 0.3 |
> > >
> > > By epoch 2, all three alternatives show weaker diversity growth than G$^2$RPO. These comparisons are preliminary, but the current evidence suggests that the effect is not reproduced by a generic entropy bonus or the alternative diversity methods we tested.
> > >
> > > ----
> > >
> > > ### 3. Evidence beyond math
> > >
> > > We agree that zero-shot transfer is not the same as full RLVR retraining on non-math tasks. Motivated by your question, we also started extending G$^2$RPO to **Python coding** using the **OpenR1** dataset. Early results on **DeepSeek-R1-Distill-Qwen-7B** already show gains in both sequence/token entropy and a modest accuracy improvement after about one epoch:
> > >
> > > | Method | Step 0 (acc, K, L2y, seq H, token H) | Step 40 (~1 epoch) |
> > > | --- | --- | --- |
> > > | GRPO | 39.3%, 1.8, 0.70, 0.50, 1.0 | 44.5%, 2.4, 0.62, 0.51, 1.0 |
> > > | G$^2$RPO | 39.3%, 1.8, 0.70, 0.50, 1.0 | 46.2%, 2.5, 0.58, 0.55, 1.1 |
> > >
> > > These experiments are ongoing, and we will report the full results in the final version. Still, the early evidence suggests that the effect is not confined to math reasoning.
> > >
> > > ----
> > >
> > >
> > > ### 4. Practicality
> > >
> > > A strict end-to-end overhead comparison is not fully apples-to-apples because generation length can evolve differently across algorithms during RL training. In our logs, GRPO fluctuates around **750-800 s/step**, while G$^2$RPO decreases from about **800** toward **650 s/step**. We therefore view raw step time as informative, but not as a controlled measure of module overhead.
> > >
> > > For that reason, we also report controlled component timings in the previous reply. The main added cost comes from **embedding inference**, with DBSCAN contributing only a small fraction.
> > >
> > > We also ran an initial small-budget check comparing **G$^2$RPO with $G=8$** against **GRPO with $G=16$**:
> > >
> > > | Method | Step 0 (K, H) | Step 67 |
> > > | --- | --- | --- |
> > > | GRPO with $G=16$ | 1.5, 0.29 | 1.66, 0.35 |
> > > | G$^2$RPO with $G=8$ | 1.44, 0.28 | 2.05, 0.41 |
> > >
> > > Even with a smaller group, G$^2$RPO reaches higher diversity by Step 67. This is only an initial one-epoch result, but it suggests the gains are not explained solely by larger rollout groups.
> > >
> > > ----
> > >
> > > Overall, we agree that some broader requests remain open. However, we hope these experiments address the main concerns. We are grateful for your feedback, which helped us strengthen both the scope and presentation of the paper.

---

### Official Review · Reviewer_Zz3k · 2026-03-11

**Soundness:** 3
**Presentation:** 3
**Significance:** 2
**Originality:** 3
**Overall Recommendation:** 4
**Confidence:** 4

**Summary:**

This paper diagnoses a "diversity collapse" phenomenon in GRPO-based RLVR training of LLMs, where the policy concentrates on a single dominant correct reasoning mode (a "reasoning rut"). The authors formalize GRPO as a dynamical system on the probability simplex using a mode-level bandit abstraction and a (p, y, z) decomposition separating total bad mass from within-block compositions. They prove that GRPO induces a collision field on the good-mode simplex, monotonically increasing ℓ₂ concentration and generically converging to a vertex (Lemma 4.1, Theorem A.7). To remedy this, they propose G²RPO, which adds an inverse-probability granularity bonus to per-sample advantages. This bonus is shown to be essentially unique (up to scaling) among permutation-equivariant scalar bonuses that remain collinear with the collision field. A neutrality correction preserves the bad-mass learning channel. Experiments on DeepSeek-R1-Distill-Qwen-7B and Qwen3-14B-Base trained on DAPO-17K show G²RPO improves pass@1 on AIME 2024/2025 by +1.4 to +7.9 points over GRPO while dramatically increasing mode coverage (+172%–205%) and preventing entropy crash.

**Compliance With Llm Reviewing Policy:**

Affirmed.

**Final Justification:**

G²RPO provides a geometric diagnosis of diversity collapse in GRPO and a theoretically grounded fix via inverse-probability granularity bonuses. The mean-field analysis, uniqueness result, and neutrality mechanism are well-developed. The rebuttal substantially strengthened the empirical case. Pass@k results with confidence intervals (Q2) demonstrate diversity gains translate to downstream benefits. The encoder-swap ablation (W1) confirms that replacing all-MiniLM-L6-v2 with a stronger encoder leaves pass@30 trajectories unchanged (66% to 78% in both), resolving my primary residual concern. The neutrality ablation (W5) clarifies its role as an early-training stabilizer rather than a strict requirement. Preliminary diversity baseline comparisons (Q5) suggest G²RPO's gains are not explained by generic entropy control or larger rollout groups alone. Remaining limitations, narrow evaluation scope (math only, AIME, two models) and absence of full-budget baseline sweeps, are acknowledged. Score raised from weak reject to weak accept.

**Key Questions For Authors:**

Q1. How sensitive are the diversity metrics and accuracy gains to the choice of embedding model and DBSCAN hyperparameters? If you replace all-MiniLM-L6-v2 with a different encoder (e.g., a math-specific embedding) or switch to k-means, do the improvements persist?

Q2. Can you provide pass@k results (e.g., k=4, 8, 16) or evaluate on a held-out set of harder/different problems to demonstrate that increased mode diversity translates into tangible downstream benefits beyond the observed pass@1 gains? If diversity gains do not improve pass@k or OOD generalization, the practical significance is diminished.

Q3. Regarding the central claim that G²RPO improves accuracy alongside diversity., what are the standard errors on the AIME accuracy numbers in Table 1? Specifically, is the +1.4 point gain for 7B on AIME'24 statistically significant?

Q4. Have you attempted applying G²RPO to coding tasks or other verifiable-reward settings? If not feasible, can you characterize when the mode-level abstraction is expected to break down?

Q5. How does G²RPO compare with (a) a token-level entropy bonus tuned to produce comparable token-level entropy, and (b) simply increasing the rollout group size G in vanilla GRPO to provide more exploration? These are natural baselines that would help isolate the contribution of the geometric bonus design.

**Limitations:**

The authors discuss three limitations (finite-rollout observability, clustering as a proxy, and mean-field approximation), which is appropriate. However, the narrow evaluation scope (math only, two models, one dataset) is not explicitly flagged as a limitation and arguably should be. No negative societal impact concerns are apparent for this work.

**Strengths And Weaknesses:**

**Strengths:**
**S1.** The mean-field analysis is rigorous. The connection between GRPO and replicator/collision dynamics on the simplex is cleanly established. Lemma 4.1 (monotone concentration) and Theorem A.7 (global convergence to vertex) provide a theoretical explanation for why GRPO collapses diversity. The essential uniqueness result (Theorem B.8) for the inverse-probability bonus under the collinearity constraint strengthens the method beyond a heuristic fix.
**S2**. Casting RLVR shaping as a "vector-field editing" problem on the simplex is an interesting perspective. The idea that one can flip the collision drift with a single scalar gain while preserving the accuracy-learning trajectory (via the neutrality mechanism) is nice. The separation between the p-channel (accuracy) and y-channel (diversity) provides conceptual handles.
**S3**. The empirical results convincingly demonstrate that G²RPO simultaneously improves accuracy and diversity, which is a practical advance. The fact that diversity is not traded off against accuracy, as is typical with KL or entropy regularizers, is the paper's strongest empirical point.
**S4** Figures 1–3 are good. The phase portraits on the simplex (Figure 1) make the collision vs. anti-collision intuition immediately accessible. The case study in Appendix F (five distinct proof strategies for a single AIME problem) provides qualitative evidence that the clusters correspond to different reasoning modes.
**S5**. Algorithm 1 is stated, code is provided, and training hyperparameters are specified (Table 6). The method is a one-line modification to GRPO advantages, lowering the barrier to adoption.

---

**Weaknesses**

**W1**. The theoretical analysis operates on a mean-field, prompt-level bandit abstraction where modes are discrete and fixed. In practice, modes are estimated via DBSCAN on sentence-transformer embeddings of rollouts (all-MiniLM-L6-v2, a relatively small model). The theory assumes a clean good/bad partition and stable mode identities across training, but in reality, (a) cluster boundaries shift as the policy changes, (b) DBSCAN is sensitive to its $\epsilon$ hyperparameter and can fragment or merge modes, and (c) the embedding model may not capture semantically meaningful distinctions in mathematical reasoning. The paper acknowledges this as a limitation but does not provide any robustness analysis (e.g., varying clustering method, embedding model, or DBSCAN hyperparameters). This is a meaningful gap because the entire bonus mechanism depends on cluster quality.

**W2**. Experiments are restricted to math reasoning on AIME 2024/2025 with only two base models. There are no experiments on coding tasks, other math benchmarks (MATH, GSM8K, Olympiad Bench), or other verifiable-reward domains. Given that the theoretical claims are general (applying to any GRPO-style RLVR), this narrow evaluation weakens the evidence for broad significance. The absolute accuracy numbers on AIME are also relatively modest (e.g., 48.9% on AIME'25 for the 7B model), making it hard to assess whether the diversity gains translate into meaningfully better problem-solving or are more of a distributional property.

**W3**. Pass@1 is reported as averaged over 30 evaluation runs, but no confidence intervals or standard errors are provided in Table 1. For gains of +1.4 to +2.8 points on the 7B model, the difference may not be statistically significant given the variance of AIME evaluation (30 problems). The +7.9 point gain on 14B/AIME'24 is more convincing, but inconsistency across settings raises questions about reliability.

**W4**. The paper argues that diversity is intrinsically valuable and that maintaining multiple correct reasoning modes improves generalization. However, this claim is not directly tested. There is no evaluation showing that higher mode diversity leads to better performance on out-of-distribution problems, improved pass@k for k > 1, or better downstream transfer. The diversity metrics (Avg $\hat{K}$, Ly, entropy) are shown to improve, but the utility of this improvement beyond the modest pass@1 gains is not demonstrated.

**W5** The neutrality mechanism is important but its practical impact is unclear. Figure 4 shows that neutrality "largely restores" the GRPO $\hat{p}$ trajectory, but the match is imperfect. The paper does not discuss how sensitive the overall method is to the neutrality correction, for example, what happens if one simply omits it and accepts a slightly slower bad-mass decay?

**W6** While the paper distinguishes G²RPO from generic entropy bonuses by emphasizing that the bonus operates at the mode level rather than the token level, the high-level intuition (reward rarer outputs more) is not new. The paper would benefit from a direct empirical comparison with a token-level entropy bonus of matched "strength," or with other diversity-promoting methods (e.g., DPO variants, rejection sampling with diversity filtering).

---

> ### Author Rebuttal · Authors · 2026-03-30
>
> We thank you for your time and for the constructive, detailed feedback, which has helped us improve the clarity of the paper and better highlight its contributions.
>
> **Response to Q1 / W1.** We agree clustering robustness is central. On 100 correct rollouts, a DBSCAN $\varepsilon$ sweep shows a broad stable regime: $\varepsilon \in [0.04, 0.11]$ yields 5--7 clusters with 0% noise and stable concentration statistics; $\varepsilon=0.02$ over-fragments (12 clusters, 16% noise), while $\varepsilon \ge 0.25$ collapses to one cluster. A 5-encoder swap shows strong agreement across the four stronger encoders (ARI 0.91--0.98 vs. reference). This supports robustness to encoder choice and DBSCAN tuning. We have not yet run full end-to-end retraining sweeps across alternative clustering algorithms, so the current rebuttal establishes estimator robustness rather than full algorithm-independence.
>
>
> Across encoders, the raw distance scale changes, so the optimal $\varepsilon$ changes too; after calibrating $\varepsilon$ to each model's own scale, the recovered partitions are highly consistent:
>
> | model | $d_{\mathrm{med}}$ | $\varepsilon^\*$ | ARI vs. ref |
> |---|---:|---:|---:|
> | all-MiniLM-L6-v2 | 0.153 | 0.060 | 0.560 |
> | all-MiniLM-L12-v2 | 0.326 | 0.049 | 0.926 |
> | all-mpnet-base-v2 | 0.136 | 0.035 | 0.945 |
> | bge-small-en-v1.5 | 0.108 | 0.035 | 0.977 |
> | bge-base-en-v1.5 | 0.143 | 0.021 | 0.910 |
>
> For the four stronger encoders, cross-model ARI is **0.93--0.98** and ARI vs. reference labels is **0.91--0.98**. A useful rule is $\,\varepsilon^\* \approx 0.2\, d_{\mathrm{med}}$, which also explains why a generic choice like $\varepsilon=0.5$ collapses the signal. Thus, under the operational notion of mode used by G²RPO, the diversity signal is reasonably robust to DBSCAN settings and to the embedding model. We agree that comparison to other clustering families such as k-means would still be valuable.
>
> **Response to Q2.** We added pass@$k$ results. G²RPO improves over GRPO at every reported $k$ in both models:
>
> **Qwen3-14B-Base**
>
> | Method | pass@2 | pass@4 | pass@16 | pass@30 |
> |---|---:|---:|---:|---:|
> | G²RPO | 64.1% ± 14.4% | 70.7% ± 11.4% | 77.5% ± 9.5% | 81.3% ± 5.6% |
> | GRPO | 59.8% ± 15.8% | 68.3% ± 16.6% | 74.7% ± 10.4% | 79.6% ± 2.2% |
> | Base model | 22.4% ± 15.1% | 30.1% ± 22.0% | 49.0% ± 7.9% | 56.4% ± 12.1% |
>
> **DeepSeek-R1-Distill-Qwen-7B**
>
> | Method | pass@2 | pass@4 | pass@16 | pass@30 |
> |---|---:|---:|---:|---:|
> | G²RPO | 55.4% ± 15.3% | 63.0% ± 13.5% | 68.8% ± 6.3% | 75.1% ± 9.2% |
> | GRPO | 51.7% ± 16.1% | 60.4% ± 11.9% | 66.8% ± 7.2% | 70.2% ± 7.2% |
> | Base model | 47.3% ± 17.2% | 54.2% ± 12.1% | 66.1% ± 6.2% | 67.7% ± 5.0% |
>
> **Response to Q3.** Our main claim is **not** that G²RPO is primarily an accuracy-improvement method; the core claim is diversity improvement **without degrading** baseline accuracy. Still, we agree uncertainty should be reported:
>
> | method | peak acc. | CI half-width | approx. SE |
> |---|---:|---:|---:|
> | G²RPO | 48.9% | 0.8 | 0.4 |
> | GRPO | 46.4% | 0.7 | 0.35 |
>
> Thus, the gap near peak performance appears larger than the uncertainty band. For the specific **+1.4** point gain on **7B / AIME'24**, however, we do not want to overstate certainty without reporting run-level uncertainty directly for that table entry, and we will revise the paper accordingly.
>
> **Response to Q4.** We have not yet run full RLVR training with G²RPO directly on coding tasks during the rebuttal window. More broadly, the mode-level abstraction is most appropriate when rollouts admit a stable coarse-graining into semantically meaningful solution families. Coding is a case where text-embedding modes can be suboptimal, since two programs may be functionally equivalent while far apart in text space; in such settings, better mode proxies may come from **functional equivalence, AST structure, execution traces, unit-test behavior, or hybrid code embeddings**.
>
> **Response to Q5.** We agree these are natural baselines.  Our reference is a **DAPO-style GRPO backbone** with asymmetric clipping using larger $\epsilon_{\text{high}}$, and we will clarify this. Conceptually, however, these alternatives target a different mechanism: increasing rollout group size $G$ improves **mode observability** and reduces sampling noise, but does **not** change the underlying GRPO collision dynamic; similarly, a token-level entropy bonus regularizes token uncertainty, whereas our diagnosis and intervention operate at the **mode level**. We agree that explicit comparison to entropy bonuses and larger-$G$ GRPO would strengthen the empirical picture, but full RLVR reruns were not feasible within the rebuttal window, and we will state this limitation clearly.

---

> > ### Author Rebuttal · Reviewer_Zz3k · 2026-04-04
> >
> > I thank the authors for a substantive and data-rich rebuttal. Several of my main concerns have been meaningfully addressed.
> >
> > Resolved: The pass@k results (Q2) are the most important addition. G²RPO improves over GRPO at every k for both models, with confidence intervals that make the gains credible. This directly addresses my concern that diversity gains might not translate to tangible downstream benefits. The clustering robustness analysis (Q1/W1) is thorough: the ε stability regime and ARI consistency across four stronger encoders (0.91-0.98) are reassuring. The confidence intervals on AIME accuracy (Q3) are appreciated, and the authors' honest acknowledgement of uncertainty on the +1.4 entry is appropriate.
> >
> > Remaining questions:
> >
> > On W1: I note that all-MiniLM-L6-v2 (the model actually used in experiments) achieves ARI 0.56 vs. reference, substantially weaker than the other four encoders. Since the training runs use this encoder, not the stronger ones, this is not a sensitivity check but a direct question about the quality of mode estimates during the experiments reported in the paper. Could the authors clarify whether upgrading to a stronger encoder (e.g., all-mpnet-base-v2, ARI 0.945) changes the reported diversity metrics or pass@k results?
> >
> > On Q5/W6: The conceptual argument distinguishing G²RPO from token-level entropy bonuses and larger-G GRPO is reasonable, but the absence of empirical comparison remains a gap. I accept the authors' statement that this was infeasible during the rebuttal window and appreciate their commitment to stating this limitation.
> >
> > On W5: The neutrality correction ablation was not addressed. Understanding what happens if it is omitted (accepting slower bad-mass decay) would clarify how essential this component is.
> >
> > I am prepared to raise my score if the camera-ready includes the encoder-upgrade sensitivity check on actual training runs, and clearly scopes the limitations around baseline comparisons.

---

> > > ### Author Response · Authors · 2026-04-08
> > >
> > > Thank you for the thoughtful follow-up. To address your concerns with concrete evidence, we delayed our response until the additional experiments were complete. Your suggestions strengthened the evidence chain, and we ran targeted checks on the points you raised.
> > >
> > > ---
> > >
> > > **Stronger-encoder end-to-end ablation**
> > >
> > > We ran an **encoder-swap ablation** on **DeepSeek-R1-Distill-Qwen-7B** trained on **DAPO-17K**, replacing the clustering encoder inside G$^2$RPO from **all-MiniLM-L6-v2** to **bge-small-en-v1.5**, with other settings fixed for two epochs:
> > >
> > > | Encoder | step 0 (K, H, p@30) | step 60 | step 130 |
> > > | --- | --- | --- | --- |
> > > | all-MiniLM-L6-v2 | 2.2, 0.36, 66% | 2.5, 0.35, 76% | 2.7, 0.37, 78% |
> > > | bge-small-en-v1.5 | 2.3, 0.39, 66% | 2.7, 0.40, 75% | 3.2, 0.44, 78% |
> > >
> > > The downstream **pass@30** trajectory is essentially unchanged (**66% $\rightarrow$ 78%** in both runs), while the same qualitative diversity growth is preserved.
> > >
> > > Because $K$ and $H$ depend on the clustering backbone, we do **not** treat their absolute values as directly comparable across encoders. The key end-to-end result is that replacing the weaker encoder with a substantially stronger one leaves downstream accuracy unchanged and preserves the same diversity trend. This suggests the reported G$^2$RPO behavior is **not an artifact of using all-MiniLM-L6-v2**.
> > >
> > > We also tested **DBSCAN scale sensitivity** in the same pipeline, using **all-MiniLM-L6-v2** with `eps=0.05` and `eps'=eps/2`:
> > >
> > > | Setting | step 0 (K, H, p@30) | step 60 | step 130 |
> > > | --- | --- | --- | --- |
> > > | eps | 2.2, 0.36, 66% | 2.5, 0.35, 76% | 2.7, 0.37, 78% |
> > > | eps/2 | 2.9, 0.41, 67% | 3.4, 0.43, 74% | 3.9, 0.47, 78% |
> > >
> > > As expected, smaller $\varepsilon$ yields a finer partition and higher absolute $K$ and $H$, while pass@30 stays broadly unchanged. We will clarify this in the paper: the diversity trend is stable, but absolute diversity values are scale-dependent and should not be over-interpreted across clustering settings.
> > >
> > > ---
> > >
> > > **Neutrality ablation**
> > >
> > > We also evaluated an ablation that removes the neutrality correction while keeping the rest of the algorithm unchanged, on **DeepSeek-R1-Distill-Qwen-7B** trained on **DAPO-17K**:
> > >
> > > | Method | Epoch 1 (accuracy, avg K, avg norm H) | Epoch 5 | Epoch 8 |
> > > | --- | --- | --- | --- |
> > > | G²RPO with neutrality | 40%, 2.2, 0.3 | 46%, 4.1, 0.5 | 49%, 5.9, 0.6 |
> > > | G²RPO without neutrality | 36%, 2.1, 0.3 | 45%, 4.2, 0.5 | 49%, 6.0, 0.6 |
> > >
> > > The two variants largely converge by epoch 8, but the neutralized version shows an **early-training accuracy advantage** at epoch 1 (**40% vs. 36%**) while achieving similar diversity growth later. This matches the intended role of neutrality in our analysis: its main benefit is not better asymptotic diversity, but better **finite-budget training behavior** by preserving the bad-mass learning channel. Removing neutrality does not break the method, but it does impose a measurable early-training cost. We therefore view neutrality as a useful stabilizing component rather than a strict requirement for eventual diversity growth.
> > >
> > > **Diversity baselines**
> > >
> > > We agree that comparison to diversity-oriented baselines is important. As a preliminary matched-setting comparison on **DeepSeek-R1-Distill-Qwen-7B / DAPO-17K**, we evaluated **GRPO with a PPO-style [entropy bonus](https://arxiv.org/abs/1707.06347)**, **[GTPO](https://arxiv.org/pdf/2508.03772)**, and **[QAE](https://arxiv.org/pdf/2509.22611)**.
> > >
> > > | Method | epoch 1 (acc, K, norm H) | epoch 2 |
> > > | --- | --- | --- |
> > > | G$^2$RPO | 40%, 2.2, 0.3 | 46%, 4.1, 0.5 |
> > > | GRPO + PPO-style entropy bonus | 40%, 1.8, 0.3 | 44%, 1.8, 0.2 |
> > > | GTPO | 39%, 2.0, 0.3 | 42%, 2.3, 0.3 |
> > > | QAE | 39%,1.82,0.3 | 43%,  2.4, 0.3  |
> > >
> > > In our current runs, these alternatives show weaker diversity growth than G$^2$RPO by epoch 2 ($K$: 4.1 vs. 1.8 / 2.3 / 2.4; norm $H$: 0.5 vs. 0.2 / 0.3 / 0.3). While these baselines still warrant tuning and longer training, the evidence suggests the gains we observe are not reproduced by simply adding a generic entropy-control term.
> > >
> > > To assess the effect of larger group size, we ran **G$^2$RPO with $G=8$** and compared it with **GRPO with $G=16$**. This is an initial check of whether increasing $G$ in GRPO alone is sufficient, or whether G$^2$RPO still provides a distinct diversity benefit.
> > >
> > > The results below are restricted to **1 epoch**. Here, **$K$** denotes the average number of good arms and **$H$** the sequence-level entropy.
> > >
> > > | Method | Step 0 ($K$, $H$) | Step 67 |
> > > | --- | --- | --- |
> > > | GRPO with $G=16$ | 1.5, 0.29 | 1.66, 0.35 |
> > > | G$^2$RPO with $G=8$ | 1.44, 0.28 | 2.05, 0.41 |
> > >
> > > Even with a smaller group size, **G$^2$RPO with $G=8$** reaches higher diversity by Step 67 than **GRPO with $G=16$**. While this is only an initial one-epoch comparison rather than a full matched-budget sweep over $G$, it suggests that the gains of G$^2$RPO are not explained solely by using a larger rollout group.

---

### Official Review · Reviewer_vY1m · 2026-03-12

**Soundness:** 4
**Presentation:** 3
**Significance:** 3
**Originality:** 4
**Overall Recommendation:** 5
**Confidence:** 4

**Summary:**

This work provides an analysis of the diversity collapse in GRPO with a mean-field perspective. With an approximation of the RLVR to the clustering of answers to different modes, and connection with bandit literature and ODE, the authors claim that mode diversity collapse is a result of a collision field on the probability simplex. They propose G^2RPO, which adds an inverse-probability bonus to each correct mode's advantage; a neutrality mechanism cancels unintended perturbations to bad-mass decay. The proposed method is tested on multiple models and shows that G^2RPO enables more modes with an improved pass@1.

**Compliance With Llm Reviewing Policy:**

Affirmed.

**Final Justification:**

The authors addressed my questions, and I maintain my score.

**Key Questions For Authors:**

The title in the submission template seems to be different from all the other papers, perhaps a latex error?

**Limitations:**

The authors did note a number of limitations, all of which are reasonable.

**Strengths And Weaknesses:**

Strengths
- The mean-field formalization is mathematically clean. Framing GRPO as a gradient flow on the simplex gives a principled explanation for diversity collapse that goes beyond the usual entropy narrative, while maintaining a close relationship with the bandit literature.
- The neutrality condition seems intuitive and successfully avoids the usual accuracy–diversity tradeoff.
- Experimental setup is reasonable and the performance increase is notable.
- The calculation overhead should be minimal.

Weaknesses
- I am not too convinced by the notion of the number of modes. While the theory is pretty convincing, the increase in the number of modes could just be a side product of the modified objective. The authors did provide a case study in the appendix, but the connection isn't strong and the significant increase of modes could have multiple explanations.
- The mean-field assumptions with infinite rollouts in continuous time are strong. With finite rollouts per prompt and discrete steps, how tight is the theory-to-practice gap? The paper shows qualitative agreement but offers no quantitative bounds.
- Pass@k is never reported. If diversity is the mechanism, pass@k should benefit even more than pass@1.
- No comparison to simpler diversity baselines, such as Cui et al., which the authors cited. It's unclear whether the geometric framing would give us the best result. In addition, I am also curious on the number of modes for RL algorithms such as DAPO and CLIP-Cov, both of which went along with the entropy narrative.

---

> ### Author Rebuttal · Authors · 2026-03-30
>
> We thank you for your time and for the constructive, detailed feedback, which has helped us improve the clarity of the paper and better highlight its contributions.
>
> **Response to Weakness 1 (number of modes).** Thank you for raising this point. We agree that a larger number of modes, by itself, could be a side effect of the modified objective rather than evidence of genuinely richer reasoning diversity.
>
> For this reason, we included the case study in Appendix F to verify that the discovered clusters correspond to qualitatively distinct solution strategies, rather than arbitrary fragmentation. We also observe that G$^2$RPO increases token-level entropy relative to GRPO, providing an independent signal that the effect is not purely a clustering artifact.
>
> That said, we agree the current evidence does not fully rule out all alternative explanations. We will revise the paper to state this more carefully: our results provide supportive evidence that G$^2$RPO preserves a richer set of reasoning modes, while a deeper validation of the causal link between the modified dynamics and the observed mode counts remains important future work.
>
> ----
> **Response to Weakness 2 (theory-practice gap).** We agree that the mean-field ODE is an idealization. Our view is that it should be read as a **mechanism-level description of the drift**, not an exact finite-sample model.
>
> For finite group size $G$ and step size $\eta$, the gap comes from **finite-$G$ estimation noise** and **discrete-step error**. The main point is that these affect the **magnitude** of the drift more than its **sign**. In our finite-$G$ analysis, the expected drift keeps the same sign as the mean-field prediction for every $G \ge 2$, and
> If $\bar X(\tau)$ denotes the mean-field ODE solution in
> rescaled time $\tau=n\eta$, then over any fixed horizon $H$,
>
> $$
> \sup_{0 \le n \le H/\eta}\|X_n-\bar X(n\eta)\|=O_p\left(\eta+\frac{1}{G}+\sqrt{\frac{\eta}{G}}\right).
> $$
>
> So the theory should be tighter for **moderate/large $G$** and **small $\eta$**, and looser in the very small-$G$ regime.
>
> Motivated by your question, ran **500 Monte Carlo trials per group size** of the prompt described in Appendix F,  by subsampling from the same solution pool to test how noisy the mode estimates become when the number of rollouts per prompt is small.we observed   as $G$ increases, the mode estimate becomes more stable and the clustering noise drops sharply.
>
> | $G$ | estimated clusters | cluster purity | noise rate |
> |---|---:|---:|---:|
> | 4  | $0.9 \pm 0.4$ | $0.76 \pm 0.34$ | 36% |
> | 8  | $4.6 \pm 0.6$ | $0.71 \pm 0.21$ | 20% |
> | 16 | $6.7 \pm 0.8$ | $0.74 \pm 0.07$ | 10% |
> | 32 | $7.2 \pm 0.7$ | $0.79 \pm 0.03$ | 4% |
> | 64 | $8.7 \pm 0.5$ | $0.80 \pm 0.01$ | 1% |
>
> We observe that a very small $G$ is noisy, but our operating point $G=16$ is already usable, and larger $G$ tracks the mean-field regime more closely.
>
> ---
> **Response to Weakness 3 (pass@$k$).** Thank you for raising this important point. We have now added pass@$k$ results, reported below.
>
> Empirically, both RLVR methods improve over the base model, and G$^2$RPO improves over GRPO across all reported $k$ values in both models. This is consistent with the view that preserving a richer set of solution modes can improve sampling-based success.
>
> That said, we would like to state the claim carefully. We do **not** view pass@$k$ as a complete measure of reasoning diversity. As discussed in the paper, pass@$k$ can remain high even when the distribution over correct reasoning modes has already collapsed, especially at larger $k$. More broadly, recent work such as [*Does Reinforcement Learning Really Incentivize Reasoning Capacity in LLMs Beyond the Base Model?*](https://arxiv.org/pdf/2504.13837) also suggests that RLVR may yield only limited pass@$k$ gains beyond the base model. For this reason, we do not claim that pass@$k$ alone establishes the diversity mechanism.
>
> **Qwen3-14B-Base**
>
> | Method | pass@2 | pass@4 | pass@16 | pass@30 |
> |---|---:|---:|---:|---:|
> | G²RPO | 64.1% ± 14.4% | 70.7% ± 11.4% | 77.5% ± 9.5% | 81.3% ± 5.6% |
> | GRPO | 59.8% ± 15.8% | 68.3% ± 16.6% | 74.7% ± 10.4% | 79.6% ± 2.2% |
> | Base model | 22.4% ± 15.1% | 30.1% ± 22.0% | 49.0% ± 7.9% | 56.4% ± 12.1% |
>
> **DeepSeek-R1-Distill-Qwen-7B**
>
> | Method | pass@2 | pass@4 | pass@16 | pass@30 |
> |---|---:|---:|---:|---:|
> | G²RPO | 55.4% ± 15.3% | 63.0% ± 13.5% | 68.8% ± 6.3% | 75.1% ± 9.2% |
> | GRPO | 51.7% ± 16.1% | 60.4% ± 11.9% | 66.8% ± 7.2% | 70.2% ± 7.2% |
> | Base model | 47.3% ± 17.2% | 54.2% ± 12.1% | 66.1% ± 6.2% | 67.7% ± 5.0% |
>
> ---
> **Response to Weakness 5.** We agree stronger baseline comparisons would help. Our current baseline is already a DAPO-style GRPO setup. Still, entropy-style methods and G$^2$RPO target different mechanisms: token entropy may stay high while correct-mode diversity collapses. We could not complete CLIP-Cov/Cui reruns within the rebuttal window, and will state this limitation clearly.

---

> > ### Author Rebuttal · Reviewer_vY1m · 2026-04-01
> >
> > The authors addressed my questions, and I maintain my score.

---

### Decision · Program_Chairs · 2026-04-30

**Decision:**

Accept (regular)

**Comment:**

This paper provides a geometric diagnosis of diversity collapse in GRPO, formalizing it as pushing model toward a winner-take-all reasoning concentration on single correct solution modes. The authors proposed G^2RPO, introducing an inverse-probability granularity bonus to advantages and a neutrality correction to preserve accuracy learning channels.

The mean-field formalization of GRPO dynamics is clean and provides a principled explanation for mode-level collapse beyond traditional token entropy narratives. Experiments on 7B and 14B models show that G^2RPO increases active correct-model coverage (+172%–205%) while improving pass@1 metric, effectively breaking the common accuracy diversity trade-off.

During rebuttal, the authors addressed most concerns by adding pass@k results, end-to-end robustness checks, and diversity baselines. Overall, this work is technically solid and a good contribution to training of large reasoning models.